# A Unifying Perspective on Multicalibration: Game Dynamics for Multi-Objective Learning[*]

Nika Haghtalab, Michael I. Jordan, and Eric Zhao

University of California, Berkeley
{nika,jordan,eric.zh}@berkeley.edu

## Abstract

We provide a unifying framework for the design and analysis of multicalibrated predictors. By placing the multicalibration problem in the general setting of multi-objective learning—where learning guarantees must hold simultaneously over a set of distributions and loss functions—we exploit connections to game dynamics to achieve state-of-the-art guarantees for a diverse set of multicalibration learning problems. In addition to shedding light on existing multicalibration guarantees and greatly simplifying their analysis, our approach also yields improved guarantees, such as error tolerances that scale with the square-root of group size versus the constant tolerances guaranteed by prior works, and improving the complexity of $k$-class multicalibration by an exponential factor of $k$ versus Gopalan et al. [17]. Beyond multicalibration, we use these game dynamics to address emerging considerations in the study of group fairness and multi-distribution learning.

## 1 Introduction

Multicalibration has emerged as a powerful tool for addressing fairness considerations in machine learning. Based on calibrated forecasting [6, 12]—which requires among instances $x$ a predictor $h$ predicts the probability $h(x) = v$, a fraction $v$ have a positive outcome—multicalibration yields more fine-grained guarantees by seeking calibration across large and possibly overlapping collections of sub-populations [21]. Multicalibration has been studied in numerous settings, including those with rich label sets (multi-class multicalibration [17]), adversarial rather than stochastic data (online multicalibration [18]), and problems where no Bayes classifier exists (agnostic multicalibration [38]). The concept of multicalibration has also been applied to estimate other quantities, such as higher moments [22], and been strengthened in various ways, such as providing conditional guarantees [2].

Multicalibration's versatility has led to the development of numerous specialized algorithms, each tailored to a unique multicalibration problem and requiring its own individualized analysis. Promising attempts to provide an overarching conceptual framework for multicalibration, such as outcome indistinguishability [7], have had limited success in unifying these various algorithms. In this paper, we tackle this challenge, developing *a general-purpose algorithmic framework to guide the design of multicalibrated learning algorithms for a wide range of settings and considerations.*

Our approach is a dynamical systems and game-theoretic approach [see, e.g., 13, 9]. We demonstrate that many multicalibration algorithms can be formulated as particular instances of two-player zero-sum games where players independently either run no-regret algorithms or best-response algorithms. A wide range of multicalibration algorithms that exhibit varying trade-offs can be obtained by plugging in different no-regret and best-response algorithms. This unified framework both recovers existing guarantees and in many cases improves upon them.

---

[*]Authors are ordered alphabetically. Correspondence to eric.zh@berkeley.edu.

37th Conference on Neural Information Processing Systems (NeurIPS 2023).

| Problem | Complexity | Dynamic | Previous Results | Our Results | Reference |
|---|---|---|---|---|---|
| MC (Det) | Oracle | NRBR | $O(k\varepsilon^{-2})$ [17] | $O(\ln(k)\varepsilon^{-2})$ | Thm 4.2 |
| MC (Sqrt Err, Det) | Sample | NRBR | $\widetilde{O}(\varepsilon^{-6}\sqrt{k}\ln(k\,|\mathcal{S}|))$ | $\widetilde{O}(\varepsilon^{-4}\ln(k\,|\mathcal{S}|))$ | Thm 4.4 |
| Agnostic MC (Det) | Oracle | NRBR | $O(|\mathcal{X}|\,\varepsilon^{-2})$ [38] | $O(\varepsilon^{-2})$ | Thm E.9 |
| Agnostic MC | Sample | NRNR | $O(|\mathcal{X}|\,\varepsilon^{-2})$ [38] | $O((d+k)\varepsilon^{-2})$ | Thm E.10 |
| Cond. MC (Det) | Oracle | NRBR | $O(|\mathcal{S}|^2\,\varepsilon^{-2})$ | $O(|\mathcal{S}|\,\varepsilon^{-2})$ | Thm E.3 |
| Cond. MC | Sample | NRNR | $O(|\mathcal{S}|^2\,(d+k)\varepsilon^{-2})$ | $O(|\mathcal{S}|\,(d+k)\varepsilon^{-2})$ | Thm E.4 |
| MC (Succinct) | Sample | NRNR | $\widetilde{O}((d+k)\varepsilon^{-3})$ [17] | $O((d+k)\varepsilon^{-2})$ | Thm 4.1 |
| Online MC | Regret | BRNR | $O(\sqrt{\log(|\mathcal{S}|)T})$ [18] | (Matching) | Thm 4.3 |
| $r$th Moment MC | Oracle | NRBR | $O(r\varepsilon^{-4})$ [22] | (Matching) | Thm E.13 |

Table 1: This table summarizes the sample complexity and agnostic learning oracle complexity rates we obtain for multicalibration (MC), compared with the previous state of the art. $\mathcal{S}$ denotes the set of groups one wants multicalibration on, $d$ the VC dimension of $\mathcal{S}$, $k$ the number of label classes, and $\varepsilon$ the error tolerance. (Det) and (Succinct) respectively denote when only deterministic or only succinct predictors are acceptable. "Sqrt Err" refers to requiring an error tolerance of $\varepsilon\sqrt{\Pr(x \in S)}$ for each group rather than $\varepsilon$.

Although approaching multicalibration—a min-max optimization problem—with game dynamics seems like an obvious approach, no prior work has succeeded in using a game dynamics framing to unify multicalibration algorithms. The primary challenge is not reducing multicalibration to min-max optimization, which is straightforward (Facts 2.5,2.6), but rather solving the resulting equilibrium computation problem in a way that connects to practical algorithms. The needs of multicalibration (such as determinism, large and complex predictor spaces, etc.) differ significantly in this regard from earlier applications of general-purpose no-regret algorithms and game dynamics.

Our primary contributions can be categorized into three areas.

**1) Unifying framework.** In Section 3, we give an overview of game dynamics as a concise but general framework for obtaining multi-objective learning guarantees. To use these dynamics as a generic solution to various multicalibration problems, we introduce a powerful general-purpose no-regret algorithm (Theorem 3.7) and a distribution-free best-response algorithm (Theorem 3.8), for calibration-like objectives. This approach allows us to unify the diverse—and often, seemingly unrelated—algorithms that have been studied in the multicalibration literature, offering a general template for their derivation and analysis.

**2) New guarantees.** In Sections 4, we use our framework to improve guarantees for various multicalibration settings (see Table 1) by simply plugging in different no-regret and best-response algorithms. These improvements include an exponential (in $k$) reduction in the complexity of $k$-class multicalibration over Gopalan et al. [17], a polynomial (in $1/\varepsilon$) reduction in the complexity of learning a succinct multicalibrated predictor over Dwork et al. [7], the first conditional multicalibration results for the batch setting, the first agnostic multicalibration guarantees that beats uniform convergence, and a multicalibration algorithm that efficiently guarantees an error tolerance that scales with *square-root* of each group's probability mass. In Section 5, we demonstrate that our framework can be extended to analyze problems beyond multicalibration, such as multi-group learning [39].

**3) Simplified analyses.** We also demonstrate that our framework can recover the guarantees of various existing multicalibration algorithms, including online multicalibration [18] and moment multicalibration [22], while avoiding intricate and problem-specific arguments.

## 1.1 Related Work

The study of calibration originated in online (adversarial) forecasting [6, 20], with classical literature having also studied calibration across multiple sub-populations [11]. Multicalibration, on the other hand, is classically studied in the stochastic setting; i.e., where $(x_i, y_i) \sim D$, in which case calibration is trivially satisfied by the predictor $h(x) = \mathbb{E}[y]$. Due to this difference in formulation in the literature on calibration and that on multicalibration, the specific technical tools that have been developed in these areas have largely remained distinct. Our work can be viewed as bridging this gap by showing

that game-theoretic dynamics provides a unified foundation for studying multicalibration, just as no-regret learning underpins the study of calibration.

Motivated by fairness considerations, a formal definition of multicalibration was presented by [21], and has found a wide range of applications and conceptual connections to Bayes optimality, conformal predictions, and computational indistinguishability [21, 22, 16, 18, 7, 23]. Algorithms for multicalibration have largely developed along two lines: one studying oracle-efficient boosting-like algorithms [see, e.g., 21, 25, 17, 7] and another studying algorithms with flavors of online optimization [see, e.g., 18, 35]. Our work establishes that the contrast between these lines of work and the algorithms they develop are entirely attributable to different choices of game dynamics.

Multi-objective and multi-distribution learning are concepts that have found broad applications in addressing fairness, collaboration, and robustness challenges. Blum et al. [4] initiated the study of learning predictors with near-optimal accuracy across multiple populations (distributions), with later works attaining tight bounds [19, 34, 31]. Recently, multi-objective learning has been applied to settings where sub-populations are mutually compatible [37, 39, 3]. In Section 5, we use our framework to match and improve guarantees relative to this literature.

## 2 Preliminaries

We use $\mathcal{X}$ to denote a feature space and $\mathcal{Y}$ a label space, where $\mathcal{Y} = [k]$ in $k$-class classification. A data distribution $D$ is a probability distribution supported on labeled datapoints $\mathcal{X} \times \mathcal{Y}$. We use $\mathcal{H}$ to denote a set of hypotheses and $\mathcal{G}$ a set of objectives, where an objective—or equivalently, a loss—is a function $\ell : \mathcal{H} \times (\mathcal{X} \times \mathcal{Y}) \to [0, 1]$ that takes a hypothesis and datapoint and returns a penalty value. We denote expected objective values by $\mathcal{L}_{D,\ell}(h) := \mathbb{E}_{(x,y) \sim D} [\ell(h, (x, y))]$. For non-deterministic $p \in \Delta(\mathcal{H})$ and $q \in \Delta(\mathcal{D} \times \mathcal{G})$, we overload notation to write $\mathcal{L}_q(p) := \mathbb{E}_{h \sim p, (D, \ell) \sim q} [\mathcal{L}_{D,\ell}(h)]$. We often use the shorthands $x^{(1:T)} := x^{(1)}, \ldots, x^{(T)}$ and $\{f(x^{(t)})\}^{(1:T)} = f(x^{(1)}), \ldots, f(x^{(T)})$. We also write $f(a, \cdot)$ to denote the function $x \mapsto f(a, x)$ or $f_{(\cdot)}(a)$ to denote $x \mapsto f_x(a)$. For $y \in [k]$, $\delta_y \in \{0, 1\}^k$ denotes its one-hot delta function, while for $y \in [k], j \in [k]$ we write $\delta_{y,j} = 1[y = j]$.

### 2.1 Multicalibration

We use $\mathcal{P} = (\Delta \mathcal{Y})^{\mathcal{X}}$ to denote the set of all $k$-class *predictors* which are maps from features to label distributions. To differentiate between predictors and distributions over predictors, we refer to $h \in \mathcal{P}$ as a *deterministic* predictor, and $p \in \Delta(\mathcal{P})$, as a *non-deterministic* predictor.[2] Calibration is a property of predictors $h \in \mathcal{P}$ requiring, for example in binary classification, that among instances $x$ assigned prediction probability $h(x) = [1 - v, v]$ a fraction $v$ are truly labeled 1. *Multicalibration* [21] is the finer-grained notion that requires calibration on subgroups of one's domain. This set of subgroups is typically finite or of finite VC dimension. In practice, we work with approximate notions of calibration/multicalibration and discretize the range of probability assignment. In $k$-class prediction, we partition the $k$-dimensional hypercube into $\lambda^k$ equal cubes $V_\lambda^k$, where $V_\lambda := \{[0, 1/\lambda), [1/\lambda, 2/\lambda), \ldots\}$. For any interval $v \in V_\lambda^k$, we use $h(x) \in v$ to denote that prediction $h(x)$ falls pointwise in the buckets of $v$, i.e., $h(x)_j \in v_j$ for all $j \in [k]$. We next formally define multicalibration.[3]

**Definition 2.1.** *Fix $\varepsilon > 0$, $\lambda \in \mathbb{Z}_+$, and a set of groups $\mathcal{S} \subseteq 2^{\mathcal{X}}$. A (possibly non-deterministic) $k$-class predictor $p \in \Delta(\mathcal{P})$ is $(\mathcal{S}, \varepsilon, \lambda)$-multicalibrated for some data distribution $D$ if*

$$\forall S \in \mathcal{S}, v \in V_\lambda^k, j \in [k] : \left| \mathbb{E}_{(x,y) \sim D, h \sim p} [(h(x)_j - \delta_{y,j}) \cdot 1[h(x) \in v, x \in S]] \right| \leq \varepsilon.$$

*That is, $p$ is calibrated on every level set $v \in V_\lambda^k$ of every group $S \in \mathcal{S}$ for every class $j \in [k]$. We are often specifically interested in a deterministic solution; that is, where $p \in \mathcal{P}$.*

---

[2]Importantly, determinism of a predictor $h$ does not imply that $h \in \mathcal{Y}^{\mathcal{X}}$ as opposed to $\Delta(\mathcal{Y})^{\mathcal{X}}$.

[3]This discretized (binned) definition of multicalibration follows convention and can be readily translated into others [21]. We also follow convention in measuring in the $\ell_\infty$ norm, the $\ell_2$ norm is also commonly used in calibration. Some definitions of multicalibration [e.g., 21] appear as conditional expectations, but for constant probability domain subgroups, which makes them equivalent to our definition.

The batch setting, where we want to find a multicalibrated predictor for some fixed data distribution $D$, is the most commonly studied. Multicalibration can also be defined for online settings where the data distribution changes adversarially over time.

**Definition 2.2** (Online multicalibration). *Fix $\varepsilon > 0$, $\lambda \in \mathbb{Z}_+$, and a set of groups $\mathcal{S} \subseteq 2^{\mathcal{X}}$. In online multicalibration, at every timestep $t \in [T]$, a learner chooses a $k$-class predictor $p^{(t)} \in \Delta(\mathcal{P})$. Nature, which observes $p^{(1:t)}$, responds with any choice of data distribution $D^{(t)}$. The learner's predictors $p^{(1:T)}$ are $(\mathcal{S}, \varepsilon, \lambda)$-online multicalibrated on $D^{(1:T)}$ if*

$$\forall S \in \mathcal{S}, v \in V_\lambda^k, j \in [k] : \left| \frac{1}{T} \sum_{t=1}^{T} \mathop{\mathbb{E}}_{\substack{h \sim p^{(t)} \\ (x,y) \sim D^{(t)}}} [1[x \in S] \cdot 1[h(x) \in v] \cdot (h(x)_j - \delta_{y,j})] \right| \leq \varepsilon.$$

## 2.2 Multi-Objective Learning

We use multi-objective learning (a generalization of multi-distribution learning introduced by [19]) as a tool for studying multicalibration and other related problems. The goal of multi-objective learning is to find a hypothesis that simultaneously minimizes a set of objective values.

**Definition 2.3** (Multi-objective learning). *A multi-objective learning problem $(\mathcal{D}, \mathcal{G}, \mathcal{H})$ consists of a set of objectives $\mathcal{G}$, a hypothesis class $\mathcal{H}$, and a set of data distributions $\mathcal{D}$. An $\varepsilon$-optimal solution to $(\mathcal{D}, \mathcal{G}, \mathcal{H})$ is a (potentially non-deterministic) hypothesis $p \in \Delta(\mathcal{H})$ where*

$$\max_{D \in \mathcal{D}, \ell \in \mathcal{G}} \mathcal{L}_{D,\ell}(p) \leq \min_{h^* \in \mathcal{H}} \max_{D^* \in \mathcal{D}, \ell^* \in \mathcal{G}} \mathcal{L}_{D^*,\ell^*}(h^*) + \varepsilon. \tag{1}$$

*We usually prefer a* deterministic *solution $p \in \mathcal{H}$ over a* non-deterministic *solution $p \in \Delta(\mathcal{H})$.*

We mainly consider *single-distribution* multi-objective problems, where $\mathcal{D} = \{D\}$, though *multi-distribution* multi-objective learning arises in conditional multicalibration (Section E.1) and group fairness (Section 5). We can also consider multi-objective learning in online adversarial settings.

**Definition 2.4** (Online multi-objective learning). *An online multi-objective learning problem $(\mathcal{D}, \mathcal{G}, \mathcal{H})$ consists of a set of distributions $\mathcal{D}$, objectives $\mathcal{G}$ and hypothesis class $\mathcal{H}$. At each timestep $t \in [T]$, a learner first picks a hypothesis $p^{(t)} \in \Delta(\mathcal{H})$. Nature, who sees $p^{(1:t)}$, responds with a data distribution $D^{(t)} \in \mathcal{D}$. We say the hypotheses $p^{(1:T)}$ are $\varepsilon$-optimal on the distributions $D^{(1:T)}$ if*

$$\frac{1}{T} \max_{\ell \in \mathcal{G}} \sum_{t=1}^{T} \mathcal{L}_{D^{(t)},\ell}(p^{(t)}) \leq \max_{D^* \in \mathcal{D}} \min_{h^* \in \mathcal{H}} \max_{\ell^* \in \mathcal{G}} \mathcal{L}_{D^*,\ell^*}(h^*) + \varepsilon. \tag{2}$$

In many problems we consider, like online multicalibration, Nature can pick from any data distribution; that is, $\mathcal{D}$ is unrestricted. Let us also note that the baseline in the right-hand-side of (2) differs from that of (1). This is intentional: when $\mathcal{D}$ is unrestricted, the min-max baseline of (1) may be large, since there may be no hypothesis that is simultaneously good for all distributions. Instead, the max-min baseline of (2) is the best hypothesis $h^*$ for the most difficult distribution $D^*$.

**Multicalibration as multi-objective learning.** (Batch) Multicalibration is a single-distribution multi-objective learning problem, whose objectives penalize over-estimation and under-estimation of label probabilities on subsets of the domain.

**Fact 2.5.** *Let $D$ be a data distribution for some $k$-class prediction problem and fix $\varepsilon > 0$, $\lambda \in \mathbb{Z}_+$, and a set of groups $\mathcal{S} \subseteq 2^{\mathcal{X}}$. For every direction $i \in \{\pm 1\}$, level set $v \in V_\lambda^k$, group $S \in \mathcal{S}$, and class $j \in [k]$, we define an objective $\ell_{i,j,S,v} : \mathcal{P} \to [0,1]$ where*

$$\ell_{i,j,S,v}(h, (x,y)) = 0.5 + 0.5 \cdot i \cdot 1[h(x) \in v, x \in S] \cdot (h(x)_j - \delta_{y,j}) \tag{3}$$

*and $\mathcal{G}_{\mathrm{mc}} := \{\ell_{i,j,S,v}\}_{i,j,S,v}$ is the set of these objectives. Predictor $p \in \Delta(\mathcal{P})$ is a $\varepsilon$-optimal solution to the multi-objective problem $(\{D\}, \mathcal{G}_{\mathrm{mc}}, \mathcal{P})$ if and only if $p$ is $(\mathcal{S}, 2\varepsilon, \lambda)$-multicalibrated for $D$.*

Online multicalibration is similarly an online multi-objective learning problem.

**Fact 2.6.** *Let $D^{(1:T)}$ be data distributions for some $k$-class prediction problem, $\mathcal{D}$ be the set of all data distributions, and $\mathcal{G}_{\mathrm{mc}}$ be as defined in (3). Fix $\varepsilon > 0$, $\lambda \in \mathbb{Z}_+$, and a set of groups $\mathcal{S} \subseteq 2^{\mathcal{X}}$. A sequence of predictors $p^{(1:T)} \in \Delta(\mathcal{P})$ is $\varepsilon$-optimal on $D^{(1:T)}$ for the online multi-objective learning problem $(\mathcal{D}, \mathcal{G}_{\mathrm{mc}}, \mathcal{P})$ if and only if $p^{(1:T)}$ is $(\mathcal{S}, 2\varepsilon, \lambda)$-online multicalibrated on $D^{(1:T)}$.*

# 3  Tools for Solving Multi-Objective Learning using Game Dynamics

A common approach to multi-objective learning is to imagine a game between a minimizing player who proposes hypotheses and a maximizing player who proposes objectives and data distributions. It is well-established (inspired by min-max equilibria, e.g., [14]) that a solution can be obtained when players use no-regret algorithms. However, considerations that arise in multicalibration motivate us to study a broader range of dynamics and their implications than has been commonly explored.

**Online learning.**  In an online learning problem, at each timestep $t \in [T]$, a learner chooses an action $a^{(t)} \in \mathcal{A}$ which an adversary observes and responds to with a cost function $c^{(t)} : \mathcal{A} \to [0, 1]$. We will usually assume costs to be linear maps. The learner's *regret* is defined as $\text{Reg}(a^{(1:T)}, c^{(1:T)}) := \sum_{t=1}^{T} c^{(t)}(a^{(t)}) - \min_{a^* \in \mathcal{A}} \sum_{t=1}^{T} c^{(t)}(a^*)$, which no-regret algorithms like Hedge [14] can bound.

**Lemma 3.1** (Hedge Regret [24])**.** *In an online learning problem where the action set is the simplex $\mathcal{A} = \Delta_k$ and costs are linear, the actions chosen by Hedge have a regret of at most $2\sqrt{\ln(k)T}$.*

We write no-regret algorithms as a function of a sequence of cost functions $\text{Alg} : ([0, 1]^{\mathcal{A}})^* \to \mathcal{A}$. For example, when $\mathcal{A} = \Delta_k$, the output of the Hedge algorithm is defined as $\text{Hedge}(c^{(1)}, \ldots, c^{(t)}) = [w_1/\|w\|_1, \ldots, w_k/\|w\|_1]$ where $w_i = \exp(-\eta \sum_{\tau=1}^{t} c^{(t)}(\delta_i))$, for a specific choice of $\eta$.

We sometimes define regret against a different baseline $B \in \mathbb{R}$, with $\text{Reg}_B(a^{(1:T)}, c^{(1:T)}) := \sum_{t=1}^{T} c^{(t)}(a^{(t)}) - B$. For example, we often consider the min-max baseline $B_{\text{weak}} := T \cdot \min_{a^* \in \mathcal{A}} \max_{c^* \in \mathcal{C}} c^*(a^*)$, where $\mathcal{C}$ is the set of cost functions that the adversary chooses from. We also often encounter *stochastic cost functions*, functions of form $c : \mathcal{A} \times (\mathcal{X} \times \mathcal{Y}) \to [0, 1]$ for which we want to minimize expected value $\mathcal{L}_{D,c} := \mathbb{E}_{(x,y) \sim D}[c(\cdot, (x, y))]$ on some distribution $D$. A stochastic cost $c$ is linear if, for every $x \in \mathcal{X}, y \in \mathcal{Y}$, $c(\cdot, (x, y))$ is a linear map. The regret of online learning algorithms on stochastic costs concentrates quickly.

**Lemma 3.2** (Stochastic Approximation [33], Lemma 3.1)**.** *Consider an online learning problem on the simplex $\Delta_k$ with linear stochastic costs. Suppose, after each timestep $t \in [T]$—that is, after picking the action $a^{(t)}$—we estimate the expected cost $\mathcal{L}_{D,c^{(t)}}$ with $\widehat{c}^{(t)}(a) := c^{(t)}(a, (x^{(t)}, y^{(t)}))$, where $(x^{(t)}, y^{(t)}) \overset{\text{i.i.d.}}{\sim} D$. With probability at least $1 - \delta$, $\left| \text{Reg}(a^{(1:T)}, \{\mathcal{L}_{D,c^{(t)}}\}^{(1:T)}) - \text{Reg}(a^{(1:T)}, \widehat{c}^{(1:T)}) \right| \leq O(\sqrt{T \ln(k/\delta)})$.*

We say an action $a$ is an $\varepsilon$-*best response* to a cost $c$ if $c(a) \leq \min_{a^* \in \mathcal{A}} c(a^*) + \varepsilon$. Similarly, an action $a$ is a *distribution-free $\varepsilon$-best response* to a stochastic cost $c$ if $\max_{x \in \mathcal{X}, y \in \mathcal{Y}} c(a, (x, y)) \leq \max_{x \in \mathcal{X}, y \in \mathcal{Y}} \min_{a^* \in \mathcal{A}} c(a^*, (x, y)) + \varepsilon$. We sometimes express the sample complexity of an algorithm in terms of the number of calls made to an *agnostic learning oracle*, which can compute $\varepsilon$-best-responses to stochastic cost functions using few samples. See Appendix A for a discussion.

**Game dynamics.**  We consider no-regret dynamics between a learner (minimizing player) who chooses hypotheses $p^{(t)} \in \Delta(\mathcal{H})$ and an adversary (maximizing player) who chooses data distributions and objectives $q^{(t)} \in \Delta(\mathcal{D} \times \mathcal{G})$ where the learner's loss is $\mathcal{L}_{q^{(t)}}(p^{(t)})$. When both players are no-regret, the time-average actions they picked quickly converge to an approximate solution for the multi-objective learning problem $(\mathcal{D}, \mathcal{G}, \mathcal{H})$. This method of *no-regret dynamics* has long played a role in empirical convergence to notions of equilibria [14]. Here, we review these dynamics and their convergence guarantees. While the proofs of these lemmas are standard at a high level (and deferred to Appendix C.1), they differ in fundamental ways from past work. In particular, these lemmas consider *weak regret*, *single timestep solutions* (instead of time-averaged ones), and the consequences of having *distribution-free best responses*, all of which play important roles in multicalibration.

**Lemma 3.3** (No-Regret vs. No-Regret (NRNR))**.** *Consider a multi-objective learning problem $(\mathcal{D}, \mathcal{G}, \mathcal{H})$, where a learner and adversary chose $p^{(1:T)} \in \Delta(\mathcal{H})$ and $q^{(1:T)} \in \Delta(\mathcal{D} \times \mathcal{G})$. If both players are no-regret, $\text{Reg}_{\text{weak}}\left(p^{(1:T)}, \{\mathcal{L}_{q^{(t)}}(\cdot)\}^{(1:T)}\right) \leq T\varepsilon$ and $\text{Reg}(q^{(1:T)}, \{-\mathcal{L}_{(\cdot)}(p^{(t)})\}^{(1:T)}) \leq T\varepsilon$, then the non-deterministic hypothesis $\overline{p} = \text{Uniform}(p^{(1:T)})$ is a $2\varepsilon$-optimal solution.*

The next dynamic focuses on obtaining a solution *from a single timestep, rather than time-averaged solutions*. To obtain this, we consider a dynamics in which the learner goes first and is no-regret, the adversary observes the learner's action and then best responds.

**Lemma 3.4** (No-Regret vs. Best-Response (NRBR)). *Consider a multi-objective learning problem* $(\mathcal{D}, \mathcal{G}, \mathcal{H})$*, where a learner chose* $p^{(1:T)} \in \Delta(\mathcal{H})$ *and an adversary chose* $q^{(1:T)} \in \Delta(\mathcal{D} \times \mathcal{G})$*. If the learner is no-regret,* $\text{Reg}_{weak}\left(p^{(1:T)}, \{\mathcal{L}_{q^{(t)}}(\cdot)\}^{(1:T)}\right) \leq T\varepsilon$*, and the adversary* $\varepsilon$*-best-responded to the costs* $\left\{-\mathcal{L}_{(\cdot)}(p^{(t)})\right\}^{(1:T)}$ *using* $q^{(1:T)}$*, then there is a* $t \in [T]$ *where* $p^{(t)}$ *is a* $2\varepsilon$*-optimal solution.*

Once the existence of a single-round solution $p^{(t)}$ is established by Lemma 3.4, it is easy to find which time step corresponds to this solution by using a few samples and testing all $p^{(1:T)}$, as follows.

**Lemma 3.5.** *Suppose a set of hypotheses* $p^{(1:T)}$ *contains an* $\varepsilon$*-optimal solution. We can find a* $5\varepsilon$*-optimal solution* $p^{(t)} \in p^{(1:T)}$ *using* $O(\varepsilon^{-2} |\mathcal{D}| \ln(|\mathcal{D}| |\mathcal{G}| T/\delta))$ *samples with probability* $1 - \delta$*.*

The next dynamic considers difficult distribution-free problems, such as online multi-objective learning. To enable learning in these scenarios, we consider a no-regret adversary that chooses objectives $q^{(1:T)} \in \Delta(\mathcal{G})$ and a learner who first observes $q^{(t)}$ and plays a distribution-free best-response. The following lemma considers the consequences of these interactions.

**Lemma 3.6** (Best-Response vs. No-Regret (BRNR)). *Consider an online multi-objective learning problem* $(\mathcal{D}, \mathcal{G}, \mathcal{H})$*, where a learner chose* $p^{(1:T)} \in \Delta(\mathcal{H})$*, an adversary chose* $q^{(1:T)} \in \Delta(\mathcal{G})$*, and* $D^{(1:T)} \in \mathcal{D}$ *is any sequence. Assume that the adversary is no-regret, i.e.,* $\text{Reg}(q^{(1:T)}, \{-\mathcal{L}_{D^{(t)},(\cdot)}(p^{(t)})\}^{(1:T)}) \leq T\varepsilon$*, and the learner's actions* $p^{(1:T)}$ *are distribution-free* $\varepsilon$*-best-responses to the stochastic costs* $q^{(1:T)}$*, i.e.,* $\max_{x,y} \mathcal{L}_{(x,y),q^{(t)}}(p^{(t)}) \leq \max_{x,y} \min_{p^*} \mathcal{L}_{(x,y),q^{(t)}}(p^*) + \varepsilon$*. Then, the hypotheses* $p^{(1:T)}$ *are* $2\varepsilon$*-optimal on* $D^{(1:T)}$*.*

The question of which dynamic should be used and their implementation hinges on the type of solution desired and what online learning and best-response guarantees are possible for each player. NRNR dynamics often offer maximum sample efficiency, since calculating $\varepsilon$-best-responses may be more sample intensive, but produce a time-averaged solution. NRBR dynamics, though less sample-efficient due to the adversary's repeated best-response computation, provides a single timestep solution and is crucial for, e.g., deterministic multicalibration. BRNR dynamics, where learners follow (act after the adversary) and have greater ease being no-regret, are crucial for online settings.

## 3.1 No-regret and Best Response Computation in Multicalibration

In this section, we introduce algorithms for obtaining (weak) no-regret and best response guarantees in multicalibration, so that we can apply the previously discussed dynamics.

Since the adversary picks from the—usually, small—set of objectives $\mathcal{G}_{\text{mc}}$, it can be no-regret using standard algorithms like Hedge. However, the learner picks from the—very large—space of all predictors $\mathcal{P}$; if it used Hedge, its regret would grow linearly in domain size $|\mathcal{X}|$. An important aspect of multicalibration is that its complexity must be independent of the domain size $\mathcal{X}$ (while it can depend on the complexity of the subgroups $\mathcal{S}$). We leverage structural properties of multicalibration objectives (see Appendix C.2 for formal treatment) to obtain generic no-regret and best response algorithms that give the learner domain-independent guarantees. Our first theorem gives a no-(weak)-regret learning algorithm for the learner that provides three important properties simultaneously: 1) domain-independent regret-bound that is also logarithmic in $k$, 2) uses no samples (or knowledge of) the underlying distribution $D$, and 3) deterministically outputs a deterministic predictor per round. Properties 1-2 lead to fast convergence and low sample complexity in the aforementioned dynamics and property 3 is key for obtaining deterministic multicalibration guarantees (via NRBR).

**Theorem 3.7.** *Consider* $\mathcal{P}$ *the set of* $k$*-class predictors and any adversarial sequence of stochastic costs* $q^{(1:T)} \in \Delta(\mathcal{G}_{\text{mc}})$*, where* $\mathcal{G}_{\text{mc}}$ *are the multicalibration objectives* (3)*. There is a no-regret algorithm that outputs (deterministic) predictors* $h^{(1:T)} \in \mathcal{P}$ *such that* $\text{Reg}_{weak}(h^{(1:T)}, \{\mathcal{L}_{D,q^{(t)}}\}^{(1:T)}) \leq 2\sqrt{\ln(k)T}$ *for every data distribution* $D$*. The algorithm does not need any samples from* $D$*.*

*Proof.* Consider the following algorithm. At each feature $x \in \mathcal{X}$, initialize a Hedge algorithm that picks an action $h^{(t)}(x) \in \Delta(\mathcal{Y})$ at each timestep $t \in [T]$. Aggregating each algorithm's action yields our learner's overall action $h^{(t)} \in \mathcal{P}$. For each $x \in \mathcal{X}$, let $h^{(t+1)}(x)$ be the outcome of Hedge at step

$t + 1$ after observing linear loss functions $f_{h^{(\tau)},x}^{(\tau)} : \mathbb{R}^k \to [0,1]$ for $\tau \in [t]$:

$$f_{h^{(\tau)},x}^{(\tau)}(z) := 0.5 + 0.5 \sum_{i \in \{\pm 1\}, j \in [k], S \in \mathcal{S}, v \in V_\lambda^k} z_j \cdot q_{i,j,S,v}^{(\tau)} \cdot i \cdot 1[h^{(\tau)}(x) \in v, x \in S], \quad (4)$$

where $q_{i,j,S,v}^{(\tau)}$ is the probability $q^{(\tau)}$ assigns to loss $\ell_{i,j,S,v}$. Hedge gives $\sum_{t=1}^T f_{h^{(t)},x}^{(t)}(h^{(t)}(x)) - \min_{z^* \in \Delta(\mathcal{Y})} \sum_{t=1}^T f_{h^{(t)},x}^{(t)}(z^*) \leq 2\sqrt{\ln(k)T}$ (Lemma 3.1). Since this inequality holds for all $x \in \mathcal{X}$,

$$2\sqrt{\ln(k)T} \geq \mathbb{E}_{(x,y)\sim D}\left[\sum_{t=1}^T f_{h^{(t)},x}^{(t)}(h^{(t)}(x))\right] - \mathbb{E}_{(x,y)\sim D}\left[\min_{z^* \in \Delta(\mathcal{Y})} \sum_{t=1}^T f_{h^{(t)},x}^{(t)}(z^*)\right] \quad (5)$$

$$= \mathbb{E}_{(x,y)\sim D}\left[\sum_{t=1}^T f_{h^{(t)},x}^{(t)}(h^{(t)}(x))\right] - \min_{h^* \in \mathcal{P}} \mathbb{E}_{(x,y)\sim D}\left[\sum_{t=1}^T f_{h^{(t)},x}^{(t)}(h^*(x))\right]$$

by law of total expectation, where the last transition is by defining $h^*$ such that $h^*(x) = z^*$ for every $x$-dependent choice of $z^*$ in (5). Add and subtract $\sum_{t=1}^T \mathbb{E}_{(x,y)\sim D}\left[f_{h^{(t)},x}^{(t)}(\delta_y)\right]$ to obtain

$$\sum_{t=1}^T \mathbb{E}_{(x,y)\sim D}\left[f_{h^{(t)},x}^{(t)}(h^{(t)}(x) - \delta_y)\right] \leq 2\sqrt{\ln(k)T} + 0.5T, \quad (6)$$

where the last transition is by the fact $h^*(x) = \mathbb{E}[\delta_y | x]$, $\sum_{t=1}^T \mathbb{E}\left[f_{h^{(t)},x}^{(t)}(h^*(x) - \delta_y)\right] = 0.5T$. Next, we show the LHS of (6) is equivalent to $\mathrm{Reg}_{\mathrm{weak}}(h^{(1:T)}, \{\mathcal{L}_{D,q^{(t)}}\}^{(1:T)})$. First recall from multicalibration objectives that $\ell_{i,j,S,v}(h, (x,y)) = 0.5 + 0.5 \cdot i \cdot 1[h(x) \in v, x \in S] \cdot (h(x)_j - \delta_{y,j})$. Since $q_{i,j,S,v}^{(t)}$ is the probability $q^{(t)}$ assigns to $\ell_{i,j,S,v}(\cdot)$ in (4), we have that $q^{(t)}(h^{(t)}, (x,y)) = f_{h^{(t)},x}^{(t)}(h^{(t)}(x) - \delta_y)$. Therefore, (6) implies that $\sum_{t=1}^T \mathcal{L}_{D,q^{(t)}}(h^{(t)}) \leq 0.5T + 2\sqrt{\ln(k)T}$. It is left to establish that the min-max baseline of these losses is indeed at least $0.5$. This is implied by Fact 2.5 and its proof is deferred to Appendix C.2. Thus $\mathrm{Reg}_{\mathrm{weak}}(h^{(1:T)}, \{\mathcal{L}_{D,q^{(t)}}\}^{(1:T)}) \leq 2\sqrt{\ln(k)T}$. □

Our second theorem proves the existence of a distribution-free best-response algorithm for the learner, which is key for obtaining online multicalibration guarantees (via BRNR). Note that, as a distribution-free algorithm, it requires no samples to compute these best-responses.

**Theorem 3.8.** *Consider the set of $k$-class predictors $\mathcal{P}$. Fix an $\varepsilon > 0$ and let $q \in \Delta(\mathcal{G}_{\mathrm{mc}})$ be a mixture of multicalibration objectives* (3). *There always exists a (non-deterministic) predictor $p \in \Delta(\mathcal{P})$ that is a distribution-free $\varepsilon$-best-response* (7) *to the stochastic cost function $q(\cdot, (x,y))$.*

*Proof Sketch.* At a high level, this statement and proof are similar to the min-max proof of calibration [10, 20], with additional details in Appendix C.2. Let $\widetilde{\Delta}(\mathcal{Y})$ be a finite $\varepsilon$-covering of $\Delta(\mathcal{Y})$ and $\phi_x : \Delta(\widetilde{\Delta}(\mathcal{Y})) \times \Delta(\mathcal{Y}) \to [-1,1]$ the bilinear function $\phi_x(a,b) = \mathbb{E}_{(i,j,S,v)\sim q, \widehat{a}\sim a}\left[i \cdot 1[\widehat{a} \in v, x \in s] \cdot (\widehat{a}_j - b_j)\right]$. Consider $p(x) = \arg\min_{a \in \Delta(\widetilde{\Delta}(\mathcal{Y}))} \max_{b \in \Delta(\mathcal{Y})} \phi_x(a,b)$. By the minmax theorem, $\max_{b \in \Delta(\mathcal{Y})} \phi_x(p(x), b) = \max_{b \in \Delta(\mathcal{Y})} \min_{a \in \widetilde{\Delta}(\mathcal{Y})} \phi_x(a,b) \leq \max_{b \in \Delta(\mathcal{Y})} \min_{a \in \Delta(\mathcal{Y})} \phi_x(a,b) + \varepsilon$. Thus, $\max_{y \in \mathcal{Y}} q(p, (x,y)) \leq 0.5 \cdot \varepsilon + \max_{y \in \mathcal{Y}} \min_{h^* \in \mathcal{P}} q(h^*, (x,y))$ for all $x \in \mathcal{X}$. □

## 4  Multicalibration with Game Dynamics

We match and improve a broad set of previous results in multicalibration (See Table 1) that had received individualized and ad hoc treatments in the past. Our work establishes that not only is there a unified approach for obtaining these results but that it all comes back to game dynamics empowered by our no-regret and distribution-free-best response results for multicalibration—Theorems 3.7 and 3.8. Below we focus on three main results highlighting NRNR, NRBR, and BRNR dynamics. We defer the proofs, formal statement of algorithms, and additional results to Appendix D and E.

**Multicalibration.** Our first algorithm uses no-regret no-regret (NRNR) dynamics to find non-deterministic multicalibrated predictors. This algorithm matches the fastest known sample complexity rates for multicalibration [18, 35] of order $O(\ln(|\mathcal{S}|\,\lambda^k)/\varepsilon^2)$. It also improves upon existing fast-rate algorithms of Gupta et al. [18], Noarov et al. [35] by producing predictors with a succinct support and small circuit size. These properties were previously only known to be attained by the less sample efficient multicalibration algorithms of [21]. In this way, Algorithm 1 simultaneously attains the best aspects of the algorithms of [18] and [21].

**Theorem 4.1.** *Fix $\varepsilon > 0$, $\lambda, k \in \mathbb{Z}_+$, set of groups $\mathcal{S} \subseteq 2^{\mathcal{X}}$, and data distribution $D$. The below algorithm, with probability $1 - \delta$, returns a non-deterministic $k$-class predictor that is $(\mathcal{S}, \varepsilon, \lambda)$-multicalibrated on $D$ and takes no more than $O\left(\varepsilon^{-2}(\ln(|\mathcal{S}|/\delta) + k\ln(\lambda))\right)$ samples from $D$.*

---
No-Regret vs No-Regret

*Construct the problem $(\{D\}, \mathcal{G}_{\mathrm{mc}}, \mathcal{P})$ from Fact 2.5 and let $T = C\varepsilon^{-2}\ln(|\mathcal{S}|\,\lambda^k/\delta)$ for some universal constant $C$. Over $T$ rounds, have an adversary choose $q^{(1:T)} \in \Delta(\mathcal{G}_{\mathrm{mc}})$ by applying Hedge to the costs $\{1 - \ell_{(\cdot)}(h^{(t)}, (x^{(t)}, y^{(t)}))\}^{(1:T)}$ where $(x^{(t)}, y^{(t)}) \overset{\text{i.i.d.}}{\sim} D$. In parallel, have a a learner choose predictors $h^{(1:T)} \in \mathcal{P}$ by applying the no-regret learning algorithm of Theorem 3.7 to the stochastic costs $\ell^{(1:T)}$, where $\ell^{(t)} \overset{\text{i.i.d.}}{\sim} q^{(t)}$. Return the predictor $p = \mathrm{Uniform}(h^{(1:T)})$. This algorithm is written explicitly in Algorithm 1.*

---

Theorem 4.1—along with all other results in this section—can be rewritten with $\mathrm{VC}(\mathcal{S})\log(1/\varepsilon)$ replacing $\ln(|\mathcal{S}|)$; this is done by taking a cover of $\mathcal{S}$. We further note that Algorithm 1 can be instantiated with different choices of no-regret algorithms for the adversary and different versions of Theorem 3.7 for the learner. In Section 6, we empirically compare such variants of Algorithm 1.

Our second algorithm uses no-regret best-response (NRBR) dynamics to find deterministic multicalibrated predictors. This algorithm improves on Gopalan et al. [17]'s oracle complexity of $O\left(k/\varepsilon^2\right)$ with $O\left(\ln(k)/\varepsilon^2\right)$, an exponential reduction of the dependence on the number of classes $k$.[4]

**Theorem 4.2.** *Fix $\varepsilon > 0$, $\lambda, k \in \mathbb{Z}_+$, a set of groups $\mathcal{S} \subseteq 2^{\mathcal{X}}$, and a data distribution $D$. The following algorithm returns a deterministic $k$-class predictor that is $(\mathcal{S}, \varepsilon, \lambda)$-multicalibrated on $D$ and makes $O(\ln(k)/\varepsilon^2)$ calls to an agnostic learning oracle. Moreover, with probability $1 - \delta$, the oracle calls can be implemented with $\widetilde{O}\left(\frac{1}{\varepsilon^3}(\sqrt{\ln(k)}\ln(k\,|\mathcal{S}|/\delta) + k\ln(\lambda))\right)$ samples from $D$.*

---
No-Regret vs Best-Response

*Construct the problem $(\{D\}, \mathcal{G}_{\mathrm{mc}}, \mathcal{P})$ from Fact 2.5 and let $T = C\varepsilon^{-2}\ln(|\mathcal{S}|\,\lambda^k\delta)$ for some universal constant $C$. Over $T$ rounds, have a learner choose predictors $h^{(1:T)} \in \mathcal{P}$ by applying the no-regret learning algorithm of Theorem 3.7 to the stochastic costs $\ell^{(1:T)}$. Have an adversary choose $\ell^{(1:T)}$ by calling an agnostic learning oracle at each $t \in [T]$: $\ell^{(t)} = \mathcal{A}_{\varepsilon/8}(1 - \mathcal{L}_{D,(\cdot)}(h^{(t)}))$. Using $C\ln(T/\delta)/\varepsilon^2$ samples from $D$, return the predictor $h^{(t^*)}$ with the lowest empirical multicalibration error. This algorithm is written explicitly in Algorithm 2.*

---

**Online multicalibration.** Our next algorithm uses best-response no-regret (BRNR) dynamics for online multicalibration, matching the best known regret bounds for online multicalibration [18, 35]. As with Theorem 4.1, our analysis simplifies that of [18, 35] by avoiding exponential potential arguments in favor of no-regret dynamics. We state the following result for binary classification, where $\mathcal{Y} = \{0, 1\}$ but note it trivially extends to multi-class settings. For convenience, we will say that predictors $\mathcal{P}$ for binary classification output real-valued $h(x) \in [0, 1]$, where $h(x)$ is the predicted probability of class 1 and $1 - h(x)$ is the predicted probability of class 0.

**Theorem 4.3.** *Fix $\varepsilon > 0$, $\lambda \in \mathbb{Z}_+$, and a set of groups $\mathcal{S} \subseteq 2^{\mathcal{X}}$. The following algorithm guarantees $(\mathcal{S}, \varepsilon, \lambda)$-online multicalibration with probability $1 - \delta$.*

---
Best-Response vs No-Regret

*Construct the online multi-objective learning problem $(\mathcal{D}, \mathcal{G}_{\mathrm{mc}}, \mathcal{P})$ in Fact 2.6 and let $T = C\varepsilon^{-2}\ln(|\mathcal{S}|\,\lambda\delta)$ for some universal constant $C$. Over $T$ rounds, have an adversary choose $q^{(1:T)} \in \Delta(\mathcal{G}_{\mathrm{mc}})$ by applying Hedge to the costs $\{1 - \ell_{(\cdot)}(p^{(t)}, (x^{(t)}, y^{(t)}))\}^{(1:T)}$, where*

---

[4]In concurrent work, Dwork et al. [8] also showed that $O\left(\ln(k)/\varepsilon^2\right)$ oracle calls are sufficient for multi-class multicalibration, with an algorithm similar to Algorithm 2.

$\left(x^{(t)}, y^{(t)}\right) \overset{\text{i.i.d.}}{\sim} D^{(t)}$. *Have a learner best-respond to each stochastic cost $q^{(t)}$ with the $(\varepsilon/2)$-distribution-free best-response $p^{(t)} \in \Delta(\mathcal{P})$ of Theorem 3.8. This algorithm is written explicitly in Algorithm 3.*

The high-probability condition of Theorem 4.3 can be removed if we assume nature presents data-points rather than data distributions, as is assumed in prior works. Interestingly, Algorithm 3's use of best-response no-regret dynamics exactly recovers the online multicalibration algorithm of [18, 35]. The analysis of Theorem 4.3 is, however, significantly simpler because we make explicit the role of the no-regret dynamics, whereas [18, 35] use potential arguments that ultimately prove no-regret dynamics and the multiplicative weights algorithm from scratch.

**Additional results.** We can also use game dynamics to improve guarantees for various other multicalibration settings and considerations. We can use Algorithms 1 and 2 to achieve domain-independent sample complexity rates for agnostic multicalibration (Theorem E.9), improving upon Shabat et al. [38]'s uniform convergence guarantees. Plugging in an online learning algorithm with second-order regret bounds allows us to achieve the first non-trivial sample complexity guarantees for conditional multicalibration in the batch setting (Theorem E.3) and a multicalibration algorithm that—with a minor less than cube-root increase in sample complexity—guarantees an error tolerance for each group that scales with the square-root of the group's probability mass (Theorem 4.4). We highlight the latter result in the following theorem.

**Theorem 4.4.** *Fix $\varepsilon > 0$, $\lambda, k \in \mathbb{Z}_+$, two sets of groups $\mathcal{S} \subseteq 2^{\mathcal{X}}$, and a data distribution $D$. There is an algorithm that, with probability at least $1 - \delta$, takes $O(\ln(k) \cdot (\ln(|\mathcal{S}|/\varepsilon\delta) + k\ln(\lambda))/\varepsilon^4)$ samples from $D$ and returns a deterministic $k$-class predictor $h$ satisfying*

$$\left| \mathop{\mathbb{E}}_{(x,y) \sim D, h \sim p} \left[ (h(x)_j - \delta_{y,j}) \cdot 1[h(x) \in v, x \in S] \right] \right| \leq \varepsilon \sqrt{\Pr(x \in S)},$$

*for all $S \in \mathcal{S}, v \in V_\lambda^k, j \in [k]$.*

Plugging in an online learning algorithm with strongly adaptive regret bounds allows us to parallelize the moment multicalibration algorithm of Jung et al. [22] (Theorem E.13).

# 5 Other Fairness Notions

This general framework of approaching multi-objective learning with game dynamics can be extended beyond multicalibration to derive results on multi-group learning. Recall that, in agnostic multi-objective learning, the trade-off between objectives is arbitrated by the worst-off objective, which can be suboptimal when some objectives are inherently more difficult. An alternative is to, given a problem $(\mathcal{D}, \mathcal{G}, \mathcal{H})$, define a competitor class $\mathcal{H}'$ and try to learn a hypothesis $h$ so that there is no objective that a competitor performs significantly better on, i.e. $\mathcal{L}^*_{\mathcal{H}'}(h) - \min_{h^* \in \mathcal{H}} \mathcal{L}^*_{\mathcal{H}'}(h^*) \leq \varepsilon$ where $\mathcal{L}^*_{\mathcal{H}'}(h) := \max_{D \in \mathcal{D}} \max_{\ell \in \mathcal{G}} \mathcal{L}_{D,\ell}(h) - \min_{h^* \in \mathcal{H}'} \mathcal{L}_{D,\ell}(h^*)$. We refer to such a solution $h$ as being $\varepsilon$-competitive. A simple modification to multi-objective learning can find such a solution: replace the objectives $\mathcal{G}$ with new objectives $\mathcal{G}' := \left\{ \frac{1}{2}(1 + \ell(\cdot) - \ell(h')) \mid \ell \in \mathcal{G}, h' \in \mathcal{H}' \right\}$ and solve as usual. Since the sample complexity of multi-objective learning is $O(\varepsilon^{-2}(\ln(|\mathcal{H}|) + |\mathcal{D}|\ln(|\mathcal{D}||\mathcal{G}|/\delta)))$ [19], we have the below sample complexity bound on finding $\varepsilon$-competitive solutions.

**Theorem 5.1.** *Consider a multi-objective learning problem $(\mathcal{D}, \mathcal{G}, \mathcal{H})$ and competitor class $\mathcal{H}'$. There is an algorithm that takes $O(\varepsilon^{-2}(\ln(|\mathcal{H}|) + |\mathcal{D}|\ln(|\mathcal{D}||\mathcal{H}'||\mathcal{G}|/\delta)))$ samples and with probability $1 - \delta$ returns a solution $p \in \Delta(\mathcal{H})$ that is $\varepsilon$-competitive against $\mathcal{H}'$.*

Consider the *multi-group learning* problem, where we seek to simultaneously minimize a general loss function on different subsets of the domain [37].

**Definition 5.2.** *Fix $\varepsilon > 0$, a set of groups $\mathcal{S} \subseteq 2^{\mathcal{X}}$, a hypothesis class $\mathcal{H}$ and a loss $\ell : \mathcal{H} \times (\mathcal{X} \times \mathcal{Y}) \to [0, 1]$. An $\varepsilon$-optimal solution to the multi-group learning problem $(\mathcal{S}, \mathcal{H})$ is a randomized hypothesis $p \in \Delta(\mathcal{H})$ that satisfies, for all $S \in \mathcal{S}$, $\mathbb{E}[\ell(p, (x,y)) \cdot 1[x \in S]] \leq \min_{h^* \in \mathcal{H}} \mathbb{E}[\ell(h^*, (x,y)) \cdot 1[x \in S]] + \varepsilon$. We always assume such a hypothesis $p$ exists in class $\mathcal{H}$.*

A (near) optimal sample complexity for multi-group learning of $O\left(\ln(|\mathcal{S}||\mathcal{H}|)/\varepsilon^2\right)$ was attained by [39] using a reduction to sleeping experts. [39] also asked whether there exists a simpler optimal algorithm that does not rely on sleeping experts. We answer this affirmatively by designing an optimal

algorithm that just runs two Hedge algorithms. The following is a direct implication of Lemma 3.3 and the observation that multi-group learning directly reduces to learning an $\varepsilon$-competitive solution.

**Theorem 5.3.** *Fix a set of groups $\mathcal{S} \subseteq 2^{\mathcal{X}}$, loss $\ell : \mathcal{Y} \times \mathcal{Y} \to [0,1]$, hypothesis class $\mathcal{H} : \mathcal{X} \to \mathcal{Y}$, and distribution $D$. There is a no-regret no-regret algorithm that takes $2T = O\left(\ln(|\mathcal{S}|\,|\mathcal{H}|)/\varepsilon^2\right)$ samples from $D$ and returns an $\varepsilon$-optimal solution to the multi-group learning problem $(\mathcal{S}, \mathcal{H})$.*

## 6 Empirical Results

In this section, we study the empirical performance of multicalibration algorithms on the UCI Adult Income dataset [26], a real-world dataset for predicting individuals' incomes based on the US Census. We defer additional results, datasets, and methods to the Appendix G.

**Experiment setup.** This experiment, summarized in Table 2, aims to learn a predictor on the UCI dataset that predicts the 'income' label and is multicalibrated on the values of eight other labeled attributes including 'age'. We discretize with $0.1$-width bins ($\lambda = 10$), and perform random 80-20 train/test splits of the dataset. This results in approximately 24000 training samples, 6000 test samples, and 130 groups. Each multicalibration algorithm is evaluated on 10 seeds, each after 50 iterations, with performance measured by their average iterate's and last iterate's multicalibration violations.

We study four multicalibration algorithms based on no-regret best-response dynamics, which use an empirical risk minimizer as the adversary and implements either Hedge [14], Prod [28], Optimistic Hedge (OptHedge) [36], or Gradient Descent (GD) as the learner. We study two algorithms based on no-regret no-regret dynamics, which plays against itself either Hedge (Hedge-Hedge) or Optimistic Hedge (OptHedge-OptHedge). Learning rate decay is tuned on the training set by sweeping over $[0.8, 0.85, 0.9, 0.95]$ for the learner and $[0.9, 0.95, 0.98, 0.99]$ for the adversary.

| Algorithm | Train Error | Test Error | Test Error (Ergodic) |
|---|---|---|---|
| Hedge-Hedge (NRNR) | 2.0e-2 $\pm$ 2.0e-3 | **3.0e-2** $\pm$ 3.0e-3 | **2.3e-4** $\pm$ 2.7e-5 |
| OptHedge-OptHedge (NRNR) | 7.0e-3 $\pm$ 0.0 | **2.7e-2** $\pm$ 3.0e-3 | 2.6e-4 $\pm$ 2.8e-5 |
| OptHedge-ERM (NRBR) | **0.0** $\pm$ 0.0 | 4.7e-2 $\pm$ 1.0e-3 | 4.8e-4 $\pm$ 9.0e-6 |
| Hedge-ERM (NRBR) | **0.0** $\pm$ 0.0 | 6.4e-2 $\pm$ 1.0e-3 | 6.4e-4 $\pm$ 1.1e-5 |
| Prod-ERM (NRBR) | **0.0** $\pm$ 0.0 | 5.3e-2 $\pm$ 4.0e-3 | 5.3e-4 $\pm$ 4.4e-5 |
| GD-ERM (NRBR) | 5.3e-2 $\pm$ 1.1e-2 | 8.3e-2 $\pm$ 3.0e-3 | 9.5e-4 $\pm$ 6.5e-5 |

Table 2: Average ($\pm$ standard error) of multicalibration violations on UCI Adult Dataset. *Train Error* and *Test Error* evaluate the last iterate (deterministic predictor) on training and test splits; *Test Error (Ergodic)* measures the average iterate (non-deterministic predictor) on test split. GD-ERM (NRBR) is worst and OptHedge-OptHedge (NRNR) is best on both deterministic and last iterates.

**One's choice of no-regret algorithm matters.** The original multicalibration algorithm of [21], which is based on gradient descent, consistently attains the worst multicalibration errors, both in terms of average-iterate and last-iterate. This is consistent with gradient descent being a theoretically less effective no-regret algorithm, due to instability near the boundaries of a probability simplex. Due to the superficial similarity between boosting and multicalibration, the field has already begun adopting multicalibration algorithms with Hedge's multiplicative updates rather than gradient descent's additive ones [25]. Our findings offer the first theoretical and empirical endorsement of this shift.

**The last iterates of no-regret no-regret dynamics are surprisingly multicalibrated.** The algorithms based on no-regret no-regret dynamics, namely Hedge-Hedge and OptHedge-OptHedge, consistently yield not only among the most multicalibrated randomized predictors (with their average iterate) but also the most multicalibrated deterministic predictors (with their last iterate). Note that these algorithms only enjoy a theoretical advantage over no-regret best-response algorithms in terms of average iterate guarantees. This does not appear to be an artifact of early stopping or learning rates (Figure 1), but may rather indicate that their more stable adversary updates provide regularization.

**Acknowledgements.** This work was supported in part by the National Science Foundation under grant CCF-2145898, a C3.AI Digital Transformation Institute grant, and the Mathematical Data Science program of the Office of Naval Research. This work was partially done while Haghtalab and Zhao were visitors at the Simons Institute for the Theory of Computing.

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

# Contents

# A    Additional Background

In this section, we supplement our discussion of online learning with some additional notations and results.

**No-regret algorithms.**   In the following lemma, we state the regret bound of the Hedge algorithm [14] for general choices of the learning rate $\eta \in (0, 1)$.

**Lemma A.1** (Hedge [24]).  *Consider an online learning problem on the simplex $\Delta_k$ and any adversarial sequence of linear cost functions $c^{(1:T)}$. If Hedge, with learning rate $\eta \in (0, 1)$, outputs $a^{(1:T)}$, then $\mathrm{Reg}(a^{(1:T)}, c^{(1:T)}) \leq T\eta + \frac{\ln(k)}{\eta}$. If $\eta = \sqrt{\ln(|\mathcal{A}|)/T}$, $\mathrm{Reg}(a^{(1:T)}, c^{(1:T)}) \leq 2\sqrt{\ln(k)T}$.*

It is helpful to note that the learning rate of the Hedge algorithm also bounds how far its iterates move in the primal space.

**Lemma A.2** (Hedge Iterate Stability).  *Consider an online learning problem on the interval $[0, 1]$ and any sequence of linear costs. Let $a^{(1:T)} \in [0, 1]$ be the actions that get picked by Hedge instantiated with learning rate $\eta \in (0, 1)$. Then, for every timestep $t \in [T]$, $\left| a^{(1)} - a^{(t)} \right| \leq 2\eta T$.*

*Proof.*   By the triangle inequality, it suffices to prove that the learner's actions move by at most $2\eta$ at each timestep. That is, $\left| a^{(t+1)} - a^{(t)} \right| \leq 2\eta$ at every timestep $t \in [T]$. By definition of the Hedge algorithm, the learner's action at timestep $t$ is given by

$$a^{(t+1)} = \frac{\exp(-\eta \sum_{\tau=1}^{t} c^{(\tau)}(1))}{\exp(-\eta \sum_{\tau=1}^{t} c^{(\tau)}(0)) + \exp(-\eta \sum_{\tau=1}^{t} c^{(\tau)}(1))}.$$

We can rearrange this as $a^{(t+1)} = a^{(t)} \cdot \alpha \cdot \exp(-\eta c^{(t)}(1))$, where $\alpha$ is a ratio of normalization terms defined as

$$\alpha := \frac{\exp(-\eta \sum_{\tau=1}^{t-1} c^{(\tau)}(0)) + \exp(-\eta \sum_{\tau=1}^{t-1} c^{(\tau)}(1))}{\exp(-\eta \sum_{\tau=1}^{t-1} c^{(\tau)}(0)) \cdot \exp(-\eta c^{(t)}(0)) + \exp(-\eta \sum_{\tau=1}^{t-1} c^{(\tau)}(1)) \cdot \exp(-\eta c^{(t)}(1))}.$$

Note that $\alpha \geq 1$. Let us write $x_1 = \exp(-\eta \sum_{\tau=1}^{t-1} c^{(\tau)}(0))$, $x_2 = \exp(-\eta \sum_{\tau=1}^{t-1} c^{(\tau)}(1))$, $x_3 = \exp(-\eta c^{(t)}(0))$ and $x_4 = \exp(-\eta c^{(t)}(1))$, noting that the values $x_1, x_2, x_3, x_4$ are all positive. Observe that $\frac{x_1+x_2}{x_1 \cdot x_3 + x_2 \cdot x_4} \leq \max\{\frac{1}{x_3}, \frac{1}{x_4}\}$. To see this, suppose without loss of generality that $x_4 \geq x_3$. Then $x_2 \leq \frac{x_2 \cdot x_4}{x_3}$, which implies that $x_1 + x_2 \leq \frac{x_1 \cdot x_3 + x_2 \cdot x_4}{x_3}$, or equivalently, $\frac{x_1+x_2}{x_1 \cdot x_3 + x_2 \cdot x_4} \leq \frac{1}{x_3}$. Substituting in our values for $x_1, x_2, x_3, x_4$, we therefore have

$$1 \leq \alpha \leq \max\{\frac{1}{\exp(-\eta c^{(t)}(0))}, \frac{1}{\exp(-\eta c^{(t)}(1))}\} \leq \frac{1}{\exp(-\eta)}.$$

The second inequality is because costs are bounded in $[0, 1]$.

This gives the iterate movement bound of

$$\left| a^{(t+1)} - a^{(t)} \right| \leq \max\{\left| a^{(t)} \frac{\exp(-\eta c^{(t)}(1))}{\exp(-\eta)} - a^{(t)} \right|, \left| a^{(t)} \exp(-\eta c^{(t)}(1)) - a^{(t)} \right|\}$$

$$\leq \max\left\{\left| a^{(t)} \exp(\eta) - a^{(t)} \right|, \left| \eta c^{(t)}(1) \right|\right\}.$$

Here, the second inequality applies the fact that $|e^{-x} - 1| \leq |x|$ for any $x \geq 0$ to the right-hand value. Since $\eta \in (0, 1)$, we can use the fact that $|\exp(\eta) - 1| \leq 2\eta$. Thus, we attain the desired bound $\left| a^{(t+1)} - a^{(t)} \right| \leq 2\eta$.   $\square$

We can modify the Hedge algorithm to obtain *strongly adaptive* regret bounds that provides guarantees for every contiguous interval.

**Lemma A.3** (Strongly Adaptive Regret [5]).  *Consider an online learning problem on the simplex $\Delta_k$. There is a modified Hedge algorithm [5] that, for any adversarial sequence of linear costs $c^{(1:T)}$,*

*outputs actions $a^{(1:T)}$ such that, for every interval $T_1 < T_2 \leq T$,*

$$\text{Reg}(a^{(T_1:T_2)}, c^{(T_1:T_2)}) \coloneqq \sum_{t=T_1}^{T_2} c^{(t)}(a^{(t)}) - \min_{a^* \in \mathcal{A}} \sum_{t=T_1}^{T_2} c^{(t)}(a^*)$$

$$\leq O\left(\left(\sqrt{\ln(k)} + \ln(T)\right)\sqrt{T_2 - T_1}\right).$$

There are also online learning algorithms that provide *second-order* regret bounds. We present the regret bound of one such algorithm, Prod [15].

**Lemma A.4** (Second-Order Regret Bound of Prod [15]). *Consider an online learning problem on the simplex $\mathcal{A} = \Delta_k$ and any adversarial sequence of linear costs $c^{(1:T)}$. If the Prod algorithm of [15] is used and outputs the actions $a^{(1:T)}$, then*

$$\text{Reg}(a^{(1:T)}, c^{(1:T)}) \leq O\left(\max_{a^* \in \mathcal{A}} \sqrt{\ln(k) \sum_{t=1}^{T} (c^{(t)}(a^{(t)}) - c^{(t)}(a^*))^2}\right).$$

We also know that there are adversarial bandit algorithms with sublinear regret guarantees.

**Lemma A.5** (Semi-Bandit Regret Bounds [19]). *Consider an online learning problem on the simplex $\Delta_k$, and any adversarial sequence of linear costs $c^{(1:T)}$. Let $\mathcal{I}$ be a partition of $[k]$ into $r$ groups. There is a high-probability variant of [29]'s ELP algorithm that, for any adversarial sequence of costs $c^{(1:T)}$, outputs a sequence of mixtures $a^{(1:T)} \in \Delta_k$ such that, with probability at least $1 - \delta$,*

$$\text{Reg}(a^{(1:T)}, c^{(1:T)}) \leq O\left(\sqrt{r \ln(k/\delta) T}\right).$$

*Moreover, after each time the algorithm chooses $a^{(t)}$, the algorithm samples an integer $i^{(t)} \sim a^{(t)}$ (unseen by the adversary). Let $\mathcal{I}(i^{(t)})$ is the group in $\mathcal{I}$ that $i^{(t)}$ belongs to. The algorithm will only ever observe the components of its cost vector corresponding to $\mathcal{I}(i^{(t)})$: $\left\{c^{(\tau)}(\delta_i) \mid i \in I(i^{(t)})\right\}^{(t)}$.*

**Best responses.** An action $a$ is an $\varepsilon$-*best response* to a cost function $c : \mathcal{A} \to [0, 1]$ if $c(a) \leq \min_{a^* \in \mathcal{A}} c(a^*) + \varepsilon$. An *agnostic learning oracle* $\mathcal{A}_\varepsilon : (\mathcal{A} \to [0, 1]) \times 2^{\mathcal{A}} \to \mathcal{A}$ is a function that, given a cost function $c$ and subset of actions $\mathcal{A}' \subseteq \mathcal{A}$, computes an $\varepsilon$-best response $\mathcal{A}_\varepsilon(c, \mathcal{A}')$ to $c$ from $\mathcal{A}'$. When $\mathcal{A}' = \mathcal{A}$, we may write $\mathcal{A}_\varepsilon(c)$ without the second argument. Agnostic learning oracles are given this name because they are usually used to find best-responses to the expected values of stochastic cost functions using some number of samples. Through this paper, we design learning algorithms that only interact with a data distribution through querying an agnostic learning oracle; the *oracle complexity* of such an algorithm is defined as the number of agnostic oracle calls that the algorithm makes. Using standard sample complexity bounds, all our algorithms also have a corresponding sample complexity that circumvent the use of agnostic learning oracles.

Another concept we encounter in online settings is the *distribution-free best response*. An action $a$ is a *distribution-free $\varepsilon$-best response* to a stochastic cost $c$ if

$$\max_{x \in \mathcal{X}, y \in \mathcal{Y}} c(a, (x, y)) \leq \max_{x \in \mathcal{X}, y \in \mathcal{Y}} \min_{a^* \in \mathcal{A}} c(a^*, (x, y)) + \varepsilon. \tag{7}$$

**Best-response algorithms.** We can efficiently find an $\varepsilon$ best-response to the expected value of a stochastic cost function using agnostic learning oracles. For example, computing a single $\varepsilon$ best-response requires at most $O(\ln(|\mathcal{A}|)/\varepsilon^2)$ samples by uniform convergence. In game dynamics, we will often need to provide best-responses for a sequence of stochastic cost functions. These sequences are usually *adaptive*, in that which stochastic cost functions appear later in the sequence depending on how we responded to previous stochastic cost functions. Adaptive data analysis provides sample-efficient algorithms for these settings.

**Lemma A.6** (Adaptive Data Analysis [1] Corollary 6.4). *There is an algorithm that, for any adaptive sequence of stochastic costs $c^{(1:T)}$, is guaranteed with probability $1 - \delta$ to $\varepsilon$-best respond to each cost in $\left\{\mathcal{L}_{D, c^{(t)}}\right\}^{(1:T)}$ while drawing at most*

$$O\left(\frac{\sqrt{T}}{\varepsilon^2} \ln\left(\frac{|\mathcal{A}|}{\varepsilon}\right) \ln^{3/2}\left(\frac{1}{\varepsilon\delta}\right)\right) \approx \widetilde{O}\left(\frac{\sqrt{T}\ln(|\mathcal{A}|/\delta)}{\varepsilon^2}\right)$$

*samples from D. Here, $\widetilde{O}(\cdot)$ suppresses $\ln(1/\varepsilon)$ and $\ln^{1/2}(1/\delta)$ factors.*

**Remark A.7.** *It often suffices, for our results, that a sequence of actions $a^{(1:T)}$ is on average $\varepsilon$-best responding to a cost sequence $c^{(1:T)}$; that is, $\sum_{t=1}^{T} c^{(t)}(a^{(t)}) \leq \sum_{t=1}^{T} \min_{a^* \in \mathcal{A}} c^{(t)}(a^*) + T\varepsilon$. Thus, it may be possible to use more efficient minimization oracles than Lemma A.6 in our algorithms.*

# B  Proofs for Section 2

This section proves Fact 2.5 and Fact 2.6 from Section 2.

**Fact 2.5.** *Let $D$ be a data distribution for some $k$-class prediction problem and fix $\varepsilon > 0$, $\lambda \in \mathbb{Z}_+$, and a set of groups $\mathcal{S} \subseteq 2^{\mathcal{X}}$. For every direction $i \in \{\pm 1\}$, level set $v \in V_\lambda^k$, group $S \in \mathcal{S}$, and class $j \in [k]$, we define an objective $\ell_{i,j,S,v} : \mathcal{P} \to [0,1]$ where*

$$\ell_{i,j,S,v}(h,(x,y)) = 0.5 + 0.5 \cdot i \cdot 1[h(x) \in v, x \in S] \cdot (h(x)_j - \delta_{y,j}) \tag{3}$$

*and $\mathcal{G}_{\mathrm{mc}} := \{\ell_{i,j,S,v}\}_{i,j,S,v}$ is the set of these objectives. Predictor $p \in \Delta(\mathcal{P})$ is a $\varepsilon$-optimal solution to the multi-objective problem $(\{D\}, \mathcal{G}_{\mathrm{mc}}, \mathcal{P})$ if and only if $p$ is $(\mathcal{S}, 2\varepsilon, \lambda)$-multicalibrated for $D$.*

*Proof.* In the multi-objective learning problem $(\mathcal{D}, \mathcal{G}_{\mathrm{mc}}, \mathcal{P})$, the multi-objective value of a predictor $p$, $\mathcal{L}^*(p) := \max_{D^* \in \mathcal{D}, \ell^* \in \mathcal{G}} \mathcal{L}_{D^*, \ell^*}(p)$, is exactly the (rescaled and shifted) magnitude of the predictor's multicalibration violation. Formally,

$$\mathcal{L}^*(p) = \frac{1}{2} + \frac{1}{2} \max_{j \in [k], S \in \mathcal{S}, v \in V_\lambda^k} \left| \mathop{\mathbb{E}}_{(x,y) \sim D, h \sim p} [(h(x)_j - \delta_{y,j}) \cdot 1[h(x) \in v, x \in S]] \right|.$$

Since we can write the absolute value of an abstract value $w$ as $|w| = \max_{i \in \{\pm 1\}} i \cdot w$, we next observe that the optimal multi-objective value of the problem $(\mathcal{D}, \mathcal{G}_{\mathrm{mc}}, \mathcal{P})$ is 0.5.

$$\min_{h^* \in \mathcal{P}} \mathcal{L}^*(h^*) = \frac{1}{2} + \frac{1}{2} \max_{i \in \{\pm 1\}} i \cdot \left[ \max_{\substack{j \in [k], S \in \mathcal{S} \\ v \in V_\lambda^k}} \mathop{\mathbb{E}}_{\substack{(x,y) \sim D \\ h \sim p}} [(h(x)_j - \delta_{y,j}) \cdot 1[h(x) \in v, x \in S]] \right] = \frac{1}{2}.$$

This fact is because multicalibration objectives $\mathcal{G}_{\mathrm{mc}}$ are symmetric around 0.5 (where $i = +1$ and $i = -1$ penalize over and under estimation), leading to the worst loss to be at least 0.5. Furthermore, the Bayes classifier $h^*$ neither overestimates nor underestimates the label distribution and therefore achieves the loss of 0.5 exactly. Combining these inequalities, we have

$$\mathcal{L}^*(p) - \min_{h^* \in \mathcal{P}} \mathcal{L}^*(h^*) = \frac{1}{2} \max_{\substack{j \in [k], S \in \mathcal{S} \\ v \in V_\lambda^k}} \left| \mathop{\mathbb{E}}_{\substack{(x,y) \sim D \\ h \sim p}} [(h(x)_j - \delta_{y,j}) \cdot 1[h(x) \in v, x \in S]] \right|.$$

Therefore $\mathcal{L}^*(p) - \min_{h^* \in \mathcal{P}} \mathcal{L}^*(h^*) = \varepsilon$ if and only if our multicalibration violation is $2\varepsilon$. $\square$

**Fact 2.6.** *Let $D^{(1:T)}$ be data distributions for some $k$-class prediction problem, $\mathcal{D}$ be the set of all data distributions, and $\mathcal{G}_{\mathrm{mc}}$ be as defined in (3). Fix $\varepsilon > 0$, $\lambda \in \mathbb{Z}_+$, and a set of groups $\mathcal{S} \subseteq 2^{\mathcal{X}}$. A sequence of predictors $p^{(1:T)} \in \Delta(\mathcal{P})$ is $\varepsilon$-optimal on $D^{(1:T)}$ for the online multi-objective learning problem $(\mathcal{D}, \mathcal{G}_{\mathrm{mc}}, \mathcal{P})$ if and only if $p^{(1:T)}$ is $(\mathcal{S}, 2\varepsilon, \lambda)$-online multicalibrated on $D^{(1:T)}$.*

*Proof.* Note that online multicalibration is an online multi-objective learning problem by construction. To analyze its optimality condition, we expand the definition of the objectives in $\mathcal{G}_{\mathrm{mc}}$ as

$$\max_{\ell \in \mathcal{G}} \frac{1}{T} \sum_{t=1}^{T} \mathcal{L}_{D^{(t)}, \ell}(p^{(t)})$$

$$= \frac{1}{2} + \frac{1}{2} \max_{\substack{j \in [k], S \in \mathcal{S} \\ v \in V_\lambda^k}} \left| \mathop{\mathbb{E}}_{\substack{h \sim p^{(t)} \\ (x,y) \sim D^{(t)}}} \left[ \frac{1}{T} \sum_{t=1}^{T} 1[x \in S] \cdot 1[h(x) \in v] \cdot (h(x)_j - \delta_{y,j}) \right] \right|.$$

We again observe that the optimal value of the problem is 0.5; formally,

$$\max_{D^* \in \mathcal{D}} \min_{h^* \in \mathcal{H}} \max_{\ell^* \in \mathcal{G}} \mathcal{L}_{D^*, \ell^*}(h^*) = 0.5.$$

This is because choosing $h^*$ to be the Bayes classifier for $D^*$ achieves the value of 0.5, which given the absolute value in the second term is the minimum achievable value. Thus, $p^{(1:T)}$ is $\varepsilon$-optimal on $D^{(1:T)}$ if and only if $p^{(1:T)}$ is $(\mathcal{S}, 2\varepsilon, \lambda)$-optimal on $D^{(1:T)}$. $\square$

# C   Proofs for Section 3

We first recall our characterization of multi-objective learning as a two-player zero-sum game. In this game, a *learner* player chooses a non-deterministic hypothesis $p \in \Delta(\mathcal{H})$ and an *adversary* player chooses a joint distribution over data distributions and objectives $q \in \Delta(\mathcal{D} \times \mathcal{G})$. The payoff of the game is the expected objective value $\mathcal{L}_q(p)$. In single-distribution multi-objective learning problems where the adversary only has one data distribution $D$ to choose from, we sometimes write $q \in \Delta(\mathcal{G})$ for simplicity. In online multi-objective learning, the adversary does not have control over which data distribution $D$ is chosen by nature. In these cases, the adversary only chooses objectives $q \in \Delta(\mathcal{G})$ and the game payoff function becomes $\mathcal{L}_{D,q}(p)$.

## C.1   Game Dynamics

We now prove formalizations of Lemma 3.3, Lemma 3.4, Lemma 3.5, and Lemma 3.6 from Section 3.

The following is a formal restatement of Lemma 3.3.

**Lemma C.1** (No-Regret vs. No-Regret). *Consider a multi-objective learning problem $(\mathcal{D}, \mathcal{G}, \mathcal{H})$, and two sequences $p^{(1:T)} \in \Delta(\mathcal{H})$ and $q^{(1:T)} \in \Delta(\mathcal{D} \times \mathcal{G})$. Suppose $\mathrm{Reg}_B\left(p^{(1:T)}, \{\mathcal{L}_{q^{(t)}}(\cdot)\}^{(1:T)}\right) \leq T\varepsilon$ and $\mathrm{Reg}(q^{(1:T)}, \{1 - \mathcal{L}_{(\cdot)}(p^{(t)})\}^{(1:T)}) \leq T\varepsilon$ where $B \in \mathbb{R}$. Then, the non-deterministic hypothesis $\overline{p} \in \Delta(\mathcal{H})$ defined as $\overline{p} \coloneqq Uniform(p^{(1:T)})$ satisfies $\mathcal{L}^*(\overline{p}) \leq B + 2\varepsilon$. If the baseline $B$ is the min-max baseline $B_{weak}$, then $\overline{p}$ is a $2\varepsilon$-optimal solution for the problem $(\mathcal{D}, \mathcal{G}, \mathcal{H})$.*

*Proof.* By connecting the adversary's regret bound and the learner's (weak) regret bounds of

$$\max_{D^* \in \mathcal{D}, \ell^* \in \mathcal{G}} \sum_{t=1}^T \mathcal{L}_{D^*, \ell^*}(p^{(t)}) - T\varepsilon \leq \sum_{t=1}^T \mathcal{L}_{q^{(t)}}(p^{(t)}) \text{ and } \sum_{t=1}^T \mathcal{L}_{q^{(t)}}(p^{(t)}) \leq T(\varepsilon + B),$$

we directly observe that $\max_{D^* \in \mathcal{D}, \ell^* \in \mathcal{G}} \frac{1}{T} \sum_{t=1}^T \mathcal{L}_{D^*, \ell^*}(p^{(t)}) \leq 2\varepsilon + B$. Linearity of expectation allows us to equate $\frac{1}{T} \sum_{t=1}^T \mathcal{L}_{D^*, \ell^*}(p^{(t)}) = \mathcal{L}_{D^*, \ell^*}(\overline{p}))$, which yields our first claim that $\mathcal{L}^*(\overline{p}) \leq 2\varepsilon + B$. The second claim just plugs $B_{\mathrm{weak}}$ into the previous inequality to obtain the definition of a $2\varepsilon$-optimal solution. $\square$

The following is a formal restatement of Lemma 3.4.

**Lemma C.2** (No-Regret vs.   Best-Response). *Consider a multi-objective learning problem $(\mathcal{D}, \mathcal{G}, \mathcal{H})$ and two sequences $p^{(1:T)} \in \Delta(\mathcal{H})$ and $q^{(1:T)} \in \Delta(\mathcal{D} \times \mathcal{G})$. Suppose $\mathrm{Reg}_B\left(p^{(1:T)}, \{\mathcal{L}_{q^{(t)}}(\cdot)\}^{(1:T)}\right) \leq T\varepsilon$ where $B \in \mathbb{R}$. Further suppose $q^{(1:T)}$ are, on average, $\varepsilon$ best-responses to the cost functions $\{1 - \mathcal{L}_{(\cdot)}(p^{(t)})\}^{(1:T)}$. Then there exists a $t \in [T]$ where $\mathcal{L}^*(p^{(t)}) \leq B + 2\varepsilon$. If $B = B_{weak}$, then $p^{(t)}$ is a $2\varepsilon$-optimal solution for the problem $(\mathcal{D}, \mathcal{G}, \mathcal{H})$.*

*Proof.* Assume the contrary, namely that $\mathcal{L}^*(p^{(t)}) > B + 2\varepsilon$ at all $t \in [T]$. Then

$$
\begin{aligned}
T\varepsilon &\geq \sum_{t=1}^T \mathcal{L}_{q^{(t)}}(p^{(t)}) - TB, &&\text{(bounded regret w.r.t. } B\text{)} \\
&\geq \sum_{t=1}^T \mathcal{L}^*(p^{(t)}) - T\varepsilon - TB, &&\text{(on average } \varepsilon\text{-best-responding)} \\
&> \sum_{t=1}^T (B + 2\varepsilon - \varepsilon - B), &&\text{(assumption to contrary)}
\end{aligned}
$$

gives us a contradiction that $T(2\varepsilon - 2\varepsilon) > 0$. This proves our first claim. The second claim follows by plugging $B = B_{\mathrm{weak}}$ into our inequality $\mathcal{L}^*(p^{(t)}) \leq B + 2\varepsilon$ to obtain the definition of a $2\varepsilon$-optimal solution. $\square$

The following is a formal restatement of Lemma 3.5.

**Lemma C.3.** *Consider a multi-objective learning problem $(\mathcal{D}, \mathcal{G}, \mathcal{H})$ and a sequence $p^{(1:T)} \in \Delta(\mathcal{H})$ of hypotheses where at least one hypothesis $p^{(t)}$ is $\varepsilon$-optimal. We can find a $3\varepsilon$-optimal solution $p^{(t^*)} \in p^{(1:T)}$ with probability at least $1 - \delta$ by taking only $O(\varepsilon^{-2} \ln(T |\mathcal{D}| \cdot |\mathcal{G}| / \delta))$ samples from each distribution $D \in \mathcal{D}$. If we further have access to a sequence $q^{(1:T)} \in \Delta(\mathcal{D} \times \mathcal{G})$ of $\varepsilon$ best-responses to the cost functions $\{1 - \mathcal{L}_{(\cdot)}(p^{(t)})\}^{(1:T)}$, then a $5\varepsilon$-optimal solution can be found with only $O(\varepsilon^{-2} \ln(T/\delta))$ samples from each $D \in \mathcal{D}$.*

*Proof.* Let the $\varepsilon$-optimal solution be denoted $p^{(t_{\mathrm{good}})}$. This claim is a simple uniform convergence argument. Suppose that $N$ i.i.d. samples $\mathbf{x}_D$ are drawn from each distribution $D \in \mathcal{D}$. We can then use $\widehat{\mathcal{L}}_{D,\ell}(p)$ to denote the empirical value of the loss function $\ell$ of hypothesis $p$ on distribution $D$, as approximated by the samples $\mathbf{x}_D$. We similarly define $\widehat{\mathcal{L}}^*$ as the empirical analog of the multi-objective value $\mathcal{L}^*$. Since the range of each *objective* is bounded in $[0, 1]$, Chernoff's bound guarantees that for any $q \in \Delta(\mathcal{D} \times \mathcal{G})$ and $p \in \Delta(\mathcal{H})$ we have $\Pr_{\mathbf{x}}\left[\left|\widehat{\mathcal{L}}_q(p) - \mathcal{L}_q(p)\right| \geq \varepsilon\right] \leq 2 \exp\left(-2N\varepsilon^2\right)$. Taking a union bound over each $D \in \mathcal{D}, t \in [T]$ and $\ell \in \mathcal{G}$ gives

$$\Pr_{\mathbf{x}}\left[\exists D \in \mathcal{D}, \ell \in \mathcal{G}, t \in [T] : \left|\widehat{\mathcal{L}}_{D,\ell}(p^{(t)}) - \mathcal{L}_{D,\ell}(p^{(t)})\right| \geq \varepsilon\right] \leq 2 |\mathcal{D}| |\mathcal{G}| T \exp\left(-2N\varepsilon^2\right).$$

Letting $t^* = \arg\min_{t \in [T]} \widehat{\mathcal{L}}^*(p^{(t)})$ and $N = O\left(\varepsilon^{-2} \ln(T |\mathcal{D}| \cdot |\mathcal{G}| / \delta)\right)$ guarantees

$$\mathcal{L}_{D^*,\ell^*}^*(p^{(t^*)}) - \varepsilon \leq \widehat{\mathcal{L}}^*(p^{(t^*)}) \leq \widehat{\mathcal{L}}^*(p^{(t_{\mathrm{good}})}) \leq \mathcal{L}^*(p^{(t_{\mathrm{good}})}) + \varepsilon \leq \min_{h^* \in \mathcal{H}} \mathcal{L}^*(h^*) + 2\varepsilon,$$

with probability at least $1 - \delta$. We can therefore return $p^{(t^*)}$ as our solution.

To prove our second claim, take instead a union bound over each $t \in [T]$ so that

$$\Pr_{\mathbf{x}}\left[\exists t \in [T] : \left|\widehat{\mathcal{L}}_{q^{(t)}}(p^{(t)}) - \mathcal{L}_{q^{(t)}}(p^{(t)})\right| \geq \varepsilon\right] \leq 2T \exp\left(-2N\varepsilon^2\right).$$

Letting $t^* = \arg\min_{t \in [T]} \mathcal{L}_{q^{(t)}}(p^{(t)})$ and $N = O\left(\varepsilon^{-2} \ln(T/\delta)\right)$ guarantees with probability at least $1 - \delta$ that we can return $p^{(t^*)}$ as our solution, as

$$\underbrace{\mathcal{L}^*(p^{(t^*)}) - 2\varepsilon \leq \mathcal{L}_{q^{(t^*)}}(p^{(t^*)}) - \varepsilon}_{\text{(since } q^{(t^*)} \text{ is an } \varepsilon\text{-best response.)}} \leq \widehat{\mathcal{L}}_{q^{(t^*)}}(p^{(t^*)}) \leq \widehat{\mathcal{L}}_{q^{(t_{\mathrm{good}})}}(p^{(t_{\mathrm{good}})}) \leq \mathcal{L}_{q^{(t_{\mathrm{good}})}}(p^{(t_{\mathrm{good}})}) + \varepsilon$$

$$\leq \mathcal{L}^*(p^{(t_{\mathrm{good}})}) + 2\varepsilon \leq \min_{h^* \in \mathcal{H}} \mathcal{L}^*(h^*) + 3\varepsilon.$$

$\square$

The following is a formal restatement of Lemma 3.6.

**Lemma C.4** (Best-Response vs. No-Regret)**.** *Consider a online multi-objective learning problem $(\mathcal{D}, \mathcal{G}, \mathcal{H})$ and the sequences $p^{(1:T)} \in \Delta(\mathcal{H})$, $q^{(1:T)} \in \Delta(\mathcal{G})$, and $D^{(1:T)} \in \mathcal{D}$. Suppose $\mathrm{Reg}(q^{(1:T)}, \{1 - \mathcal{L}_{D^{(t)},(\cdot)}(p^{(t)})\}^{(1:T)}) \leq T\varepsilon$ and $p^{(1:T)} \in \Delta(\mathcal{H})$ are distribution-free $\varepsilon'$ best-responses to $q^{(1:T)}(\cdot, (x, y))$. Then, $p^{(1:T)}$ are $(\varepsilon + \varepsilon')$-optimal on $D^{(1:T)}$.*

*Proof.* Let $\mathcal{D}^*$ be the set of all data distributions over $\mathcal{X} \times \mathcal{Y}$. By linearity of expectation, the maximum $\arg\max_{D \in \mathcal{D}^*} \mathcal{L}_D(p^{(t)})$ can always be attained on a degenerate distribution supported only on a single point $(x, y)$. Thus, the learner's distribution-free best-response guarantee provides the bound

$$\mathcal{L}_{D^{(t)},\ell^{(t)}}(p^{(t)}) - \varepsilon' \leq \max_{x \in \mathcal{X}, y \in \mathcal{Y}} q^{(t)}(p^{(t)}, (x, y)) - \varepsilon' \leq \max_{D^* \in \mathcal{D}^*} \min_{h^* \in \mathcal{H}} \mathcal{L}_{D^*,\ell^{(t)}}(h^*).$$

We can obtain our desired claim by recovering (2) with a triangle inequality between the adversary's regret bound and a summation of the previous inequality over $t \in [T]$:

$$\max_{\ell^* \in \mathcal{G}} \frac{1}{T} \sum_{t=1}^{T} \mathcal{L}_{D^{(t)},\ell^*}(p^{(t)}) - T\varepsilon \leq \frac{1}{T} \sum_{t=1}^{T} \mathcal{L}_{D^{(t)},q^{(t)}}(p^{(t)}),$$

$$\frac{1}{T} \sum_{t=1}^{T} \mathcal{L}_{D^{(t)},\ell^{(t)}}(p^{(t)}) - T\varepsilon' \leq \max_{D^* \in \mathcal{D}^*} \min_{h \in \mathcal{H}} \mathcal{L}_{D^*,\ell^{(t)}}(h) \leq \max_{D^* \in \mathcal{D}^*} \min_{h \in \mathcal{H}} \max_{\ell^* \in \mathcal{G}} \mathcal{L}_{D^*,\ell^*}(h).$$

$\square$

## C.2 No-Regret and Best-Response Computation in Multicalibration

In this section, we prove generalizations of Theorem 3.7 and Theorem 3.8 from Section 3, which describe why multicalibration objectives are amenable to efficient online learning, to a more general class of multi-objective learning problems. We refer to such problems as having *separable objectives*.

**Definition C.5.** *Consider a multi-objective learning problem* $(\mathcal{D}, \mathcal{G}, \mathcal{H})$ *where* $\mathcal{H}$ *is the set of all functions of form* $h : \mathcal{X} \to \mathcal{W}$ *and* $\mathcal{W}$ *is some convex space (and not necessarily the label space* $\mathcal{Y}$*). We say the objectives* $\mathcal{G}$ *of such a problem are* separable *if every* $\ell \in \mathcal{G}$ *is of form*

$$\ell(h, (x, y)) = c + f_\ell(x, h(x)) \cdot (h(x) - g_\ell(y)),$$

*where* $f_\ell : \mathcal{X} \times \mathcal{W} \to \mathcal{W}$ *and* $g_\ell : \mathcal{Y} \to \mathcal{W}$ *are arbitrary functions and* $c$ *is some constant offset.*

We will see that, like with multicalibration objectives, having a learner's cost functions all be separable objectives introduces two major advantages generally.

**Advantage 1: There exist no-regret learning strategies that do not require one to randomize their actions, or even sample any data, yet still guarantee domain-independent regret bounds.** The following theorem, which proves this advantage of separable objectives, is a generalization of Theorem 3.7.

**Theorem C.6.** *Consider a multi-objective learning problem* $(\mathcal{D}, \mathcal{G}, \mathcal{H})$ *where all objectives in* $\mathcal{G}$ *are separable and all distributions in* $\mathcal{D}$ *are absolutely continuous with respect to a common distribution* $D^*$. *Let* $\mathcal{A}$ *be an online algorithm that, for any linear costs* $c^{(1:T)}$, *outputs actions* $a^{(1:T)} \in \mathcal{W}$ *where* $\mathrm{Reg}(a^{(1:T)}, c^{(1:T)}) \leq R(T)$. *If* $|\mathcal{D}| > 1$, *further assume* $\mathrm{Reg}(a^{(1:T)}, c^{(1:T)}) \leq R(T) \cdot \max_{a^* \in \mathcal{W}} \sqrt{\frac{1}{T} \sum_{t=1}^{T} (c^{(t)}(a^{(t)}) - c^{(t)}(a^*))^2}$. *Then for any stochastic costs* $q^{(1:T)} \in \Delta(\mathcal{G})$ *and distributions* $D^{(1:T)} \in \mathcal{D}$, *the below algorithm outputs predictors* $h^{(1:T)} \in \mathcal{H}$ *where* $\mathrm{Reg}_{B^*}(h^{(1:T)}, \{\mathcal{L}_{D^{(t)}, q^{(t)}}(\cdot)\}^{(1:T)}) \leq \sqrt{|\mathcal{D}|} R(T)$ *and* $B^*$ *is the baseline*

$$B^* := c + \min_{h^* \in \mathcal{H}} \frac{1}{T} \sum_{t=1}^{T} \max_{h' \in \mathcal{H}} \mathop{\mathbb{E}}_{\substack{(x,y) \sim D^{(t)} \\ \ell \sim q^{(t)}}} \left[ f_\ell(x, h'(x)) \cdot (h^*(x) - g_\ell(y)) \right]. \tag{8}$$

---

*At each timestep* $t$, *construct the predictor* $h^{(t+1)}(x)$ *by setting, for all* $x \in \mathcal{X}$,

$$h^{(t+1)}(x) := \mathcal{A}(c_x^{(1)}, \dots, c_x^{(t)}) \text{ where } c_x^{(\tau)}(w) := \frac{1}{2} \cdot \frac{dD^{(\tau)}(x)}{dD^*(x)} \cdot \left( 1 + \mathop{\mathbb{E}}_{\ell \sim q^{(\tau)}} \left[ f_\ell(x, h^{(\tau)}(x)) \right] \cdot w \right),$$

*where* $\frac{dD^{(\tau)}(x)}{dD^*(x)}$ *is the Radon-Nikodym derivative of* $D^{(\tau)}$ *with respect to* $D^*$ *at* $x$.

---

*Proof.* First, suppose that $\mathcal{D} = \{D\}$, that is $|\mathcal{D}| = 1$. Then, for any $x \in \mathcal{X}$, $c_x^{(1:T)}$ are linear costs and thus $\mathcal{A}$ guarantees $\mathrm{Reg}(\{h^{(t)}(x)\}^{(1:T)}, c_x^{(1:T)}) \leq R(T)$. By the law of total expectation,

$2R(T)$

$$\geq \mathop{\mathbb{E}}_{x \sim D^*} \left[ \sum_{t=1}^{T} \mathop{\mathbb{E}}_{\ell \sim q^{(t)}} \left[ \frac{dD(x)}{dD^*(x)} f_\ell(x, h^{(t)}) \right] \cdot h^{(t)}(x) - \min_{w^* \in \mathcal{W}} \sum_{t=1}^{T} \mathop{\mathbb{E}}_{\ell \sim q^{(t)}} \left[ \frac{dD(x)}{dD^*(x)} f_\ell(x, h^{(t)}) \right] \cdot w^* \right]$$

$$= \mathop{\mathbb{E}}_{x \sim D} \left[ \sum_{t=1}^{T} \mathop{\mathbb{E}}_{\ell \sim q^{(t)}} \left[ f_\ell(x, h^{(t)}) \right] \cdot h^{(t)}(x) \right] - \mathop{\mathbb{E}}_{x \sim D} \left[ \min_{w^* \in \mathcal{W}} \sum_{t=1}^{T} \mathop{\mathbb{E}}_{\ell \sim q^{(t)}} \left[ f_\ell(x, h^{(t)}) \right] \cdot w^* \right]$$

$$= \sum_{t=1}^{T} \mathop{\mathbb{E}}_{x \sim D^{(t)}, \ell \sim q^{(t)}} \left[ f_\ell(x, h^{(t)}) \cdot h^{(t)}(x) \right] - \min_{h^* \in \mathcal{H}} \sum_{t=1}^{T} \mathop{\mathbb{E}}_{x \sim D^{(t)}, \ell \sim q^{(t)}} \left[ f_\ell(x, h^{(t)}) \cdot h^*(x) \right].$$

If $|\mathcal{D}| > 1$, $\mathcal{A}$ further guarantees

$$\Pr_{D^*}(x) \cdot \text{Reg}(\{h^{(t)}(x)\}^{(1:T)}, c_x^{(1:T)}) \leq \Pr_{D^*}(x) \cdot \max_{w \in \mathcal{W}} R(T) \sqrt{\frac{1}{T} \sum_{t=1}^{T} (c_x^{(t)}(h^{(t)}(x)) - c_x^{(t)}(w))^2}$$

$$\leq \Pr_{D^*}(x) \cdot R(T) \sqrt{\frac{1}{2T} \sum_{t=1}^{T} \left( \frac{dD^{(t)}(x)}{dD^*(x)} \right)^2}$$

$$\leq R(T) \cdot \sqrt{\frac{1}{2T} \sum_{t=1}^{T} \Pr_{D^{(t)}}(x)^2}.$$

For simplicity, we assumed above that $D^{(1:T)}$ each have discrete support. The law of total expectation gives

$$\sqrt{2} \cdot R(T) \cdot \sum_{x \in \mathcal{X}} \sqrt{\frac{1}{2T} \sum_{t=1}^{T} \Pr_{D^{(t)}}(x)^2}$$

$$\geq \mathbb{E}_{x \sim D^*} \left[ \sum_{t=1}^{T} \mathbb{E}_{\ell \sim q^{(t)}} \left[ \frac{dD^{(t)}(x)}{dD^*(x)} f_\ell(x, h^{(t)}) \right] \cdot h^{(t)}(x) - \min_{w^* \in \mathcal{W}} \sum_{t=1}^{T} \mathbb{E}_{\ell \sim q^{(t)}} \left[ \frac{dD^{(t)}(x)}{dD^*(x)} f_\ell(x, h^{(t)}) \right] \cdot w^* \right]$$

$$= \sum_{t=1}^{T} \mathbb{E}_{x \sim D^{(t)}, \ell \sim q^{(t)}} \left[ f_\ell(x, h^{(t)}) \cdot h^{(t)}(x) \right] - \min_{h^* \in \mathcal{H}} \sum_{t=1}^{T} \mathbb{E}_{x \sim D^{(t)}, \ell \sim q^{(t)}} \left[ f_\ell(x, h^{(t)}) \cdot h^*(x) \right].$$

To bound the left-hand term, we can apply Cauchy-Schwartz to get

$$\sum_{x \in \mathcal{X}} \sqrt{\frac{1}{T} \sum_{t=1}^{T} \Pr_{D^{(t)}}(x)^2} = \sum_{x \in \mathcal{X}} \sqrt{\sum_{D \in \mathcal{D}} \frac{T_D}{T} \Pr_D(x)^2}$$

$$\leq \sum_{x \in \mathcal{X}} \sum_{D \in \mathcal{D}} \Pr_D(x) \sqrt{\frac{T_D}{T}}$$

$$= \sum_{D \in \mathcal{D}} \sqrt{\frac{T_D}{T}} \sum_{x \in \mathcal{X}} \Pr_D(x)$$

$$\leq \sqrt{|\mathcal{D}|},$$

where $T_D$ is the number of timesteps $t \in [T]$ where $D^{(t)} = D$. Thus, whether $|\mathcal{D}| = 1$ or $|\mathcal{D}| > 1$, we have

$$\sum_{t=1}^{T} \mathbb{E}_{x \sim D^{(t)}, \ell \sim q^{(t)}} \left[ f_\ell(x, h^{(t)}) \cdot h^{(t)}(x) \right] - \min_{h^* \in \mathcal{H}} \sum_{t=1}^{T} \mathbb{E}_{x \sim D^{(t)}, \ell \sim q^{(t)}} \left[ f_\ell(x, h^{(t)}) \cdot h^*(x) \right] \leq 2\sqrt{|\mathcal{D}|} R(T).$$

Adding and subtracting the term $\sum_{t=1}^{T} \mathbb{E}_{(x,y) \sim D^{(t)}, \ell \sim q^{(t)}} \left[ f_\ell(x, h^{(t)}(x))(g_\ell(y)) \right]$, we have

$$\sum_{t=1}^{T} \mathbb{E}_{\substack{(x,y) \sim D^{(t)} \\ \ell \sim q^{(t)}}} \left[ f_\ell(x, h^{(t)}(x))(h^{(t)}(x) - g_\ell(y)) \right]$$

$$\leq 2\sqrt{|\mathcal{D}|} R(T) + \min_{h^* \in \mathcal{H}} \sum_{t=1}^{T} \mathbb{E}_{\substack{(x,y) \sim D^{(t)} \\ \ell \sim q^{(t)}}} \left[ f_\ell(x, h^{(t)}(x))(h^*(x) - g_\ell(y)) \right].$$

Adding the constant $c$ to both sides, we have

$$\sum_{t=1}^{T} \mathcal{L}_{D^{(t)}, q^{(t)}}(h^{(t)}) \le 2\sqrt{|\mathcal{D}|}R(T) + T \min_{h^* \in \mathcal{H}} \frac{1}{T} \sum_{t=1}^{T} \mathop{\mathbb{E}}_{\substack{(x,y) \sim D^{(t)} \\ \ell \sim q^{(t)}}} \left[ c + f_\ell(x, h^{(t)}(x))(h^*(x) - g_\ell(y)) \right]$$

$$\le 2\sqrt{|\mathcal{D}|}R(T) + T \min_{h^* \in \mathcal{H}} \frac{1}{T} \max_{h' \in \mathcal{H}} \sum_{t=1}^{T} \mathop{\mathbb{E}}_{\substack{(x,y) \sim D^{(t)} \\ \ell \sim q^{(t)}}} \left[ c + f_\ell(x, h'(x))(h^*(x) - g_\ell(y)) \right].$$

$\square$

We can directly prove Theorem 3.7 by instantiating the algorithm of Theorem C.6 with Hedge as the no-regret learning algorithm $\mathcal{A}$.

**Theorem 3.7.** *Consider $\mathcal{P}$ the set of $k$-class predictors and any adversarial sequence of stochastic costs $q^{(1:T)} \in \Delta(\mathcal{G}_{\mathrm{mc}})$, where $\mathcal{G}_{\mathrm{mc}}$ are the multicalibration objectives* (3). *There is a no-regret algorithm that outputs (deterministic) predictors $h^{(1:T)} \in \mathcal{P}$ such that $\mathrm{Reg}_{weak}(h^{(1:T)}, \{\mathcal{L}_{D,q^{(t)}}\}^{(1:T)}) \le 2\sqrt{\ln(k)T}$ for every data distribution $D$. The algorithm does not need any samples from $D$.*

*Proof.* First, we observe that every objective in multicalibration is separable. Specifically, for every $i \in \{\pm1\}, j \in [k], S \in \mathcal{S}, v \in V_\lambda^k$, the corresponding objective $\ell_{i,j,S,v} \in \mathcal{G}_{\mathrm{mc}}$ can be written in the form $\ell_{i,j,S,v}(h, (x,y)) = c + f_{i,j,S,v}(x, h(x)) \cdot (h(x) - g_{i,j,S,v}(y))$ where $c = 0.5$, $\mathcal{W} = \Delta_k \cup -\Delta_k$, $g_{i,j,S,v}(y) = \delta_y$ and $f_{i,j,S,v}(x, h(x)) = \delta_j \cdot 0.5 \cdot i \cdot \mathbb{1}[h(x) \in v, x \in S]$. Next, we recall that by Lemma A.1, the Hedge algorithm guarantees a regret bound of $4\sqrt{\ln(k)T}$ for any linear costs on $\mathcal{W} = \Delta_k \cup -\Delta_k$ (Lemma A.1). The algorithm of Theorem C.6, for any stochastic costs $q^{(1:T)} \in \Delta(\mathcal{G}_{\mathrm{mc}})$, outputs $h^{(1:T)}$ where

$$\sum_{t=1}^{T} q^{(t)}(h^{(t)}) \le 8\sqrt{\ln(k)T} + T \cdot \min_{h^* \in \mathcal{H}} \max_{\ell^* \in \mathcal{G}} \max_{h' \in \mathcal{H}} \mathop{\mathbb{E}}_{(x,y) \sim D} \left[ c + f_{\ell^*}(x, h'(x)) \cdot (h^*(x) - g(y)) \right]$$

$$\le 8\sqrt{\ln(k)T} + T \cdot \max_{\ell^* \in \mathcal{G}} \max_{h' \in \mathcal{H}} \mathop{\mathbb{E}}_{(x,y) \sim D} \left[ c + f_{\ell^*}(x, h'(x)) \cdot (g(y) - g(y)) \right]$$

$$\le 8\sqrt{\ln(k)T} + Tc,$$

if we choose $h^*(x) = \mathbb{E}_{(x,y) \sim D}\left[ g^{(t)}(y) \mid x \right]$. Since the adversary is choosing from objectives symmetric around $c$, the min-max baseline $B_{\mathrm{weak}}$ is at least $c$, meaning that $\mathrm{Reg}_{\mathrm{weak}}(h^{(1:T)}, \{\mathcal{L}_{D,q^{(t)}}(\cdot)\}^{(1:T)}) \le 8\sqrt{\ln(k)T}$. $\square$

**Advantage 2: There (almost) always exists distribution-free best responses.** This advantage is what allows multicalibration to be achievable in online settings where data arrives adversarially. [10, 20] first showed that this advantage exists in calibrated online forecasting when Hart pointed out that Foster's asymptotic calibrated forecasting result could be obtained by appealing to Blackwell approachability. Moreover, their proof extends trivially to online multicalibration. Later on, [18, 35] independently rediscovered this same advantage and proof. In the following theorem, we provide a rigorous treatment—largely in line with the prior works—that generalizes to all multi-objective learning problems that have separable objectives and satisfy some weak compactness conditions.

**Theorem C.7.** *Consider an online multi-objective learning problem $(\mathcal{D}, \mathcal{G}, \mathcal{H})$ with separable objectives and where $\mathcal{D}$ is the unrestricted set of all data distributions. Further, consider any objective mixture $q \in \Delta(\mathcal{G})$ with finite support and suppose that for every objective $\ell \in \mathcal{G}$, (1) there is a finite subset $\mathcal{Y}_\ell \subseteq \mathcal{Y}$ s.t. $\{g_\ell(y) \mid y \in \mathcal{Y}_\ell\}$ is an $(\varepsilon/3)$-net for $\{g_\ell(y) \mid y \in \mathcal{Y}\}$, (2) the range of $g_\ell$ is convex, compact and finite-dimensional, and (3) fixing any $x, y$ pair, $\ell(h, (x, y))$ is, in the argument $h$, a function of bounded variation. Then, for any $\varepsilon > 0$, there is a non-deterministic hypothesis $p \in \Delta(\mathcal{H})$ (given by (9)) that is a distribution-free $\varepsilon$ best-response for $q$.*

*Proof.* Let $\mathcal{G}_q = \mathrm{Support}(q)$ be the support of our objective mixture, noting that $|\mathcal{G}_q| < \infty$ by assumption. Let $\mathcal{W}_g \subseteq \prod_{\ell \in \mathcal{G}_q} \mathcal{W}$ denote the range of the vector-valued function $g : \mathcal{Y} \to \mathcal{W}_g$ where $g(y) := [g_\ell(y)]_{\ell \in \mathcal{G}_q}$. Fixing some $x \in \mathcal{X}$, observe that we can rewrite our loss $q(h, (x, y))$ as the

function $\widetilde{q}_x : \mathcal{W} \times \mathcal{W}_g \to [0,1]$ where $q(h,(x,y)) = \widetilde{q}_x(h(x), g(y))$ and we define $\widetilde{q}_x(w,v) :=$ $c + \mathbb{E}_{\ell \sim q}[f_\ell(w,x)(w - v_\ell)]$.

By assumption (3), $\widetilde{q}_x(w,v)$ is a function of bounded variation in $w$ for all $x \in \mathcal{X}, v \in \mathcal{W}_g$. By Lemma C.9, for any $x \in \mathcal{X}, y \in \mathcal{Y}$, there must exist some finite subset $\mathcal{W}_{x,y} \subseteq \mathcal{W}$ such that $\{\widetilde{q}_x(w,(x,g(y)))\}_{w \in \mathcal{W}_{x,y}}$ is an $(\varepsilon/3)$-net of $\{\widetilde{q}_x(w,(x,g(y)))\}_{w \in \mathcal{W}}$. Let us choose $\mathcal{W}_x :=$ $\bigcup_{\ell \in \mathcal{G}_q} \bigcup_{y \in \mathcal{Y}_\ell} \mathcal{W}_{x,y}$, which is a finite set as $|\mathcal{Y}_\ell| < \infty$ by assumption (1) and we $|\mathcal{G}_q| < \infty$ by assumption. Moreover, by construction, for any $y \in \mathcal{Y}$, $\{\widetilde{q}_x(w,(x,y))\}_{w \in \mathcal{W}_x}$ is an $(2\varepsilon/3)$-net of $\{\widetilde{q}_x(w,(x,y))\}_{w \in \mathcal{W}}$.

We now define our best response $p$ pointwise at each $x \in \mathcal{X}$, letting

$$p(x) = \underset{w^* \in \Delta(\mathcal{W}_x)}{\arg\min} \, \underset{y \in \mathcal{Y}}{\max} \, \underset{w \sim w^*}{\mathbb{E}} \, [\widetilde{q}_x(w,(x,g(y)))] = \underset{w^* \in \Delta(\mathcal{W}_x)}{\arg\min} \, \underset{g(y) \in \mathcal{W}_g}{\max} \, \underset{w \sim w^*}{\mathbb{E}} \, [\widetilde{q}_x(w,(x,g(y)))].$$
(9)

Because each objective $\ell$ is separable, $\widetilde{q}_x(w,(x,g(y)))$ is linear in $g(y)$. By linearity of expectation, we also know that $\mathbb{E}_{w \sim w^*}[\widetilde{q}_x(w,(x,g(y)))]$ is linear in $w^*$. Thus, for any $x \in \mathcal{X}$,

$$\underset{g(y) \in \mathcal{W}_g}{\max} \, \widetilde{q}_x(p(x),(x,g(y))) = \underset{w^* \in \Delta(\mathcal{W}_x)}{\min} \, \underset{g(y) \in \mathcal{W}_g}{\max} \, \underset{w \sim w^*}{\mathbb{E}} \, [\widetilde{q}_x(w,(x,g(y)))], \quad \text{(construction of } f)$$

$$= \underset{g(y) \in \mathcal{W}_g}{\max} \, \underset{w^* \in \Delta(\mathcal{W}_x)}{\min} \, \underset{w \sim w^*}{\mathbb{E}} \, [\widetilde{q}_x(w,(x,g(y)))], \quad \text{(minimax theorem)}$$

$$\leq \underset{g(y) \in \mathcal{W}_g}{\max} \, \underset{w^* \in \Delta(\mathcal{W})}{\min} \, \underset{w \sim w^*}{\mathbb{E}} \, [\widetilde{q}_x(w,(x,g(y)))] + r\varepsilon, \quad \text{(discretization error)}$$

$$\leq \underset{g(y) \in \mathcal{W}_g}{\max} \, \underset{p^* \in \Delta(\mathcal{H})}{\min} \, \widetilde{q}_x(p^*(x),(x,g(y))) + r\varepsilon.$$

In the above, we were able to apply the minimax theorem because of assumption (2). Recall that our construction of $\mathcal{W}_x$, guarantees, for any $w \in \mathcal{W}, y \in \mathcal{Y}$, there is a $w' \in \mathcal{W}_x, y' \in \mathcal{Y}'$ such that

$$|\widetilde{q}_x(w,(x,g(y))) - \widetilde{q}_x(w,(x,g(y')))| \leq \varepsilon/3, \ (y' \in \mathcal{Y}')$$
$$|\widetilde{q}_x(w,(x,g(y'))) - \widetilde{q}_x(w',(x,g(y')))| \leq \varepsilon/3, \ (w' \in \mathcal{W}_{x,y'})$$
$$|\widetilde{q}_x(w',(x,g(y'))) - \widetilde{q}_x(w',(x,g(y)))| \leq \varepsilon/3, \ (y' \in \mathcal{Y}')$$
$$|\widetilde{q}_x(w,(x,g(y))) - \widetilde{q}_x(w',(x,g(y)))| \leq \varepsilon. \ \text{(triangle inequality)}$$

We, therefore, have that

$$\underset{D \in \mathcal{D}}{\max} \, \mathcal{L}_{D,q}(p) = \underset{x \in \mathcal{X}, y \in \mathcal{Y}}{\max} \, \widetilde{q}_x(p(x),(x,g(y)))$$

$$= \underset{x \in \mathcal{X}, g(y) \in \mathcal{W}_g}{\max} \, \widetilde{q}_x(p(x),(x,g(y)))$$

$$\leq \underset{x \in \mathcal{X}, g(y) \in \mathcal{W}_g}{\max} \, \underset{p^* \in \Delta(\mathcal{H})}{\min} \, \widetilde{q}_x(p^*(x),(x,g(y))) + r\varepsilon$$

$$= \underset{x \in \mathcal{X}, y \in \mathcal{Y}}{\max} \, \underset{p^* \in \Delta(\mathcal{H})}{\min} \, \widetilde{q}_x(p^*(x),(x,g(y))) + r\varepsilon$$

$$= \underset{D \in \mathcal{D}}{\max} \, \underset{p^* \in \Delta(\mathcal{H})}{\min} \, \mathcal{L}_{D,q}(p^*) + r\varepsilon.$$

$\square$

We can now directly prove Theorem 3.8.

**Theorem 3.8.** *Consider the set of $k$-class predictors $\mathcal{P}$. Fix an $\varepsilon > 0$ and let $q \in \Delta(\mathcal{G}_{\mathrm{mc}})$ be a mixture of multicalibration objectives* (3). *There always exists a (non-deterministic) predictor $p \in \Delta(\mathcal{P})$ that is a distribution-free $\varepsilon$-best-response* (7) *to the stochastic cost function $q(\cdot,(x,y))$.*

*Proof.* Let $(\mathcal{D}', \mathcal{G}'_{\mathrm{mc}}, \mathcal{P})$ be the relaxation of the problem $(\mathcal{D}, \mathcal{G}_{\mathrm{mc}}, \mathcal{P})$ where, instead of assuming that nature can only sample discrete labels $y \in [k]$, nature too can sample mixed labels $y \in \Delta_k$. By linearity of expectation, we have that $\max_{D' \in \mathcal{D}'} \mathcal{L}_{D',q'}(p) = \max_{D \in \mathcal{D}} \mathcal{L}_{D,q}(p)$ for every predictor $p \in \Delta(\mathcal{P})$ and objective $q \in \mathcal{G}_{\mathrm{mc}}$, where $q' \in \mathcal{G}'_{\mathrm{mc}}$ is the relaxation of $q$. Thus, a distribution-free $\varepsilon$ best-response (7) for the relaxed problem is a distribution-free $\varepsilon$ best-response for our original problem $(\mathcal{D}, \mathcal{G}_{\mathrm{mc}}, \mathcal{P})$. We next observe that $(\mathcal{D}', \mathcal{G}'_{\mathrm{mc}}, \mathcal{P})$ satisfies the conditions of Theorem C.7.

Recall that every objective $q' \in \mathcal{G}'_{\mathrm{mc}}$ can be written in a separable format where, for every $i \in \{\pm 1\}, j \in [k], S \in \mathcal{S}, v \in V^k_\lambda$, we can write $\ell_{i,j,S,v}(h,(x,y)) = c + f_{i,j,S,v}(x,h(x)) \cdot (h(x) - g_{i,j,S,v}(y))$ where $c = 0.5$, $g_{i,j,S,v}(y) = y$ and $f_{i,j,S,v}(x,h(x)) \in [0,1]^k$ is $\delta_j \cdot 0.5 \cdot i \cdot \mathbb{1}[h(x) \in v, x \in S]$. The domain and range of identity $g_{i,j,S,v}(y) = y$ is exactly $\Delta_k$ which is convex, compact, and $k$-dimensional; thus, it has a finite $\varepsilon$-covering. Finally, we observe that $f_{i,j,S,v}(h(x),(x,y))$, and by extension $\ell_{i,j,S,v}(h,(x,y))$, is always a piecewise constant function in $h(x)$ with finite discontinuities. It follows that $\ell_{i,j,S,v}$ must be of bounded total variation in $h(x)$ as variation along any linear segment is at most 1. Thus, Theorem C.7 states there exists a distribution-free $\varepsilon$ best-response for $(\mathcal{D}', \mathcal{G}'_{\mathrm{mc}}, \mathcal{P})$ and by extension $(\mathcal{D}, \mathcal{G}_{\mathrm{mc}}, \mathcal{P})$. $\qquad\square$

**Remark C.8** (Closed-form of distribution-free best-responses). *For calibration and multicalibration problems, the best-response implicitly defined in Theorem C.7 takes a clean closed-form: for any given $x \in \mathcal{X}$, $p(x)$ randomizes on two neighboring actions. This simple closed form was first derived by [10] and independently rediscovered by [18].*

In Theorem C.7, we referenced the notion of a function being of bounded variation. This is a common notion in analysis which says that a function cannot go up and down too many times; thus reasonable objectives (including every finite loss function you can think of) should all be of bounded variation. For completeness, we prove below that bounded variation implies finite domain coverings.

**Lemma C.9.** *Consider a function $f : K \to [0,1]$ where $K$ is a convex compact subset of $\mathbb{R}^n$. Suppose that $f$ has pathwise bounded variation on $K$: that is, there exists a finite constant $M$ such that for any linear path $\gamma : [0,1] \to K$, $V_\gamma(f) = \sup_{0=t_0 \le t_1 \le \cdots \le t_N = 1} \sum_{i=1}^N \|f(\gamma(t_i)) - f(\gamma(t_{i-1}))\| \le M$. Then for any $\varepsilon > 0$, there exists a finite subset $S \subseteq K$ such that for every $y \in f(K)$, there exists an $x_i \in S$ with $\|f(x_i) - y\| < \varepsilon$.*

*Proof.* Since $K$ is compact, it is totally bounded. For any $\delta > 0$, there exists a finite set $T$ such that $K$ is covered by balls of radius $\delta$ centered at points in $T$. Let $\delta = \frac{\varepsilon}{2M}$, where $M$ is the constant from the pathwise bounded variation condition. For each point $t \in T$, choose a point $s_t \in K$ such that $\|s_t - t\| < \delta$. Define $S = \{s_t : t \in T\}$. Let $y \in f(K)$. Then there exists an $x \in K$ with $f(x) = y$. Since $K$ is covered by balls of radius $\delta$ centered at points in $T$, there exists a $t \in T$ with $\|x - t\| < \delta$. By the construction of $S$, we also have $\|s_t - t\| < \delta$. Then, by the triangle inequality, $\|s_t - x\| \le \|s_t - t\| + \|t - x\| < 2\delta = \frac{\varepsilon}{M}$. Consider a linear continuous path $\gamma : [0,1] \to K$ with $\gamma(0) = x$ and $\gamma(1) = s_t$; such a path must exist by the convexity of $K$. By the pathwise bounded variation condition, we have $V_\gamma(f) \le M$. For the partition $0 = t_0 \le t_1 = 1$, we have $\|f(s_t) - y\| = \|f(s_t) - f(x)\| \le V_\gamma(f) \cdot \|s_t - x\| \le M \cdot \frac{\varepsilon}{M} = \varepsilon$. Thus, for every $y \in f(K)$, there exists an $s_t \in S$ such that $\|f(s_t) - y\| < \varepsilon$. $\qquad\square$

# D    Proofs and Algorithms for Section 4

In the algorithms throughout this section, given an objective $\ell^{(t)} \in \mathcal{G}_{\mathrm{mc}}$, we write $i^{(t)}, j^{(t)}, S^{(t)}, v^{(t)}$ such that $\ell^{(t)} = \ell_{i,j,S,v}$.

## D.1    Batch Multicalibration

**Multicalibration with non-deterministic predictors.**    Our first algorithm uses no-regret no-regret (NRNR) dynamics to find non-deterministic multicalibrated predictors. We now restate and prove its guarantees from Theorem 4.1.

---

**Algorithm 1** Non-Deterministic Multicalibration Algorithm (Theorem 4.1)

1: Input: $\mathcal{S} \subseteq 2^{\mathcal{X}}$, $\varepsilon \in (0,1)$, $k, \lambda, T \in \mathbb{Z}_+$, and distribution $D$;
2: Initialize Hedge iterate $q^{(1)} = \text{Uniform}(\mathcal{G}_{\text{mc}})$ and Hedge iterate $h^{(1)} = [1/k, \ldots, 1/k]^{\mathcal{X}}$;
3: **for** $t = 1$ to $T$ **do**
4:    Sample objective $\ell^{(t)} \sim q^{(t)}$ and datapoint $(x^{(t)}, y^{(t)}) \sim D$;
5:    For every $x \in \mathcal{X}$, let $h^{(t+1)}(x) := \text{Hedge}(c_x^{(1:t)})$, where

$$c_x^{(t)}(\widehat{y}) := \frac{1}{2}(1 + i^{(t)} \cdot 1[h(x) \in v^{(t)}, x \in S^{(t)}] \cdot \widehat{y}_{j^{(t)}});$$

6:    Let $q^{(t+1)} := \text{Hedge}(c_{\text{adv}}{}^{(1:t)})$, where $c_{\text{adv}}{}^{(t)}(\ell) := 1 - \ell(h^{(t)}, (x^{(t)}, y^{(t)}))$;
7: **end for**
8: Return: $p^*$, a uniform distribution over $h^{(1)}, \ldots, h^{(T)}$;

---

**Theorem 4.1.** *Fix $\varepsilon > 0$, $\lambda, k \in \mathbb{Z}_+$, set of groups $\mathcal{S} \subseteq 2^{\mathcal{X}}$, and data distribution $D$. The below algorithm, with probability $1 - \delta$, returns a non-deterministic $k$-class predictor that is $(\mathcal{S}, \varepsilon, \lambda)$-multicalibrated on $D$ and takes no more than $O\left(\varepsilon^{-2}(\ln(|\mathcal{S}|/\delta) + k\ln(\lambda))\right)$ samples from $D$.*

---

No-Regret vs No-Regret

*Construct the problem $(\{D\}, \mathcal{G}_{\text{mc}}, \mathcal{P})$ from Fact 2.5 and let $T = C\varepsilon^{-2} \ln(|\mathcal{S}| \lambda^k/\delta)$ for some universal constant $C$. Over $T$ rounds, have an adversary choose $q^{(1:T)} \in \Delta(\mathcal{G}_{\text{mc}})$ by applying Hedge to the costs $\{1 - \ell_{(\cdot)}(h^{(t)}, (x^{(t)}, y^{(t)}))\}^{(1:T)}$ where $(x^{(t)}, y^{(t)}) \overset{\text{i.i.d.}}{\sim} D$. In parallel, have a a learner choose predictors $h^{(1:T)} \in \mathcal{P}$ by applying the no-regret learning algorithm of Theorem 3.7 to the stochastic costs $\ell^{(1:T)}$, where $\ell^{(t)} \overset{\text{i.i.d.}}{\sim} q^{(t)}$. Return the predictor $p = \text{Uniform}(h^{(1:T)})$. This algorithm is written explicitly in Algorithm 1.*

---

*Proof.* By Theorem 3.7, if $T \geq 64\varepsilon^{-2}\ln(k)$, the predictors $h^{(1:T)}$ guarantee the learner a regret bound of $\text{Reg}_{\text{weak}}(h^{(1:T)}, \{\mathcal{L}_{\ell^{(t)}}\}^{(1:T)}) \leq T\varepsilon/4$. Similarly, by Lemma 3.1, if $T \geq 576\varepsilon^{-2}\ln(2k\lambda^k|\mathcal{S}|)$, the objective mixtures $q^{(1:T)}$ guarantee the adversary a regret bound of $\text{Reg}\left(q^{(1:T)}, \{1 - \ell_{(\cdot)}(h^{(t)}, (x^{(t)}, y^{(t)}))\}^{(1:T)}\right) \leq T\frac{\varepsilon}{12}$.

We now argue that the adversary's regret with respect to the costs $1 - \ell_{(\cdot)}(h^{(t)}, (x^{(t)}, y^{(t)}))$ approximates its regret with respect to the costs $1 - \mathcal{L}_{D,(\cdot)}(h^{(t)})$. Since $\mathbb{E}_{x^{(t)}, y^{(t)} \sim D}\left[\ell_{(\cdot)}(h^{(t)}, (x^{(t)}, y^{(t)}))\right] = \mathcal{L}_{D,(\cdot)}(h^{(t)})$, by Lemma 3.2, there is a universal constant $C$ such that, if $T \geq C\varepsilon^{-2}\ln(2k\lambda^k|\mathcal{S}|/\delta)$,

$$\left|\text{Reg}\left(q^{(1:T)}, \{1 - \ell_{(\cdot)}(h^{(t)}, (x^{(t)}, y^{(t)}))\}^{(1:T)}\right) - \text{Reg}\left(q^{(1:T)}, \{1 - \mathcal{L}_{D,(\cdot)}(h^{(t)})\}^{(1:T)}\right)\right| \leq T\frac{\varepsilon}{12},$$

with probability $1 - \delta$. Similarly, since $\mathbb{E}_{\ell^{(t)} \sim q^{(t)}}\left[\mathcal{L}_{D,\ell^{(t)}}(h^{(t)})\right] = \mathcal{L}_{D,q^{(t)}}(h^{(t)})$, Lemma 3.2 guarantees

$$\left|\text{Reg}\left(q^{(1:T)}, \{1 - \mathcal{L}_{D,(\cdot)}(h^{(t)})\}^{(1:T)}\right) - \text{Reg}\left(\ell^{(1:T)}, \{1 - \mathcal{L}_{D,(\cdot)}(h^{(t)})\}^{(1:T)}\right)\right| \leq T\frac{\varepsilon}{12},$$

with probability $1 - \delta$. Taking a triangle inequality and union bound, we can see that for sufficiently large $C$, $\text{Reg}\left(\ell^{(1:T)}, \{1 - \mathcal{L}_{D,(\cdot)}(h^{(t)})\}^{(1:T)}\right) \leq T\varepsilon/4$.

By Lemma 3.3, the ergodic iterate $h^*$ is an $(\varepsilon/2)$-optimal solution. By Fact 2.5, $h^*$ is therefore a $(\mathcal{S}, \varepsilon, \lambda)$-multicalibrated predictor. The sample complexity is exactly $T$ since the algorithm only samples one datapoint at each iteration. $\qquad\square$

**Multicalibration with deterministic predictors.** We are often specifically interested in finding deterministic multicalibrated predictors. This is usually because non-deterministic predictors can be multicalibrated in a very weak sense, as we show in the following example.

**Example D.1.** *Consider a data distribution $D$ supported uniformly on $\mathcal{X} = \{x_1, x_2\}$, where $\mathcal{Y} = [0,1]$, $\Pr(Y = 1 \mid X = x_1) = 0$ and $\Pr(Y = 1 \mid X = x_2) = 1$. The non-deterministic predictor that is supported uniformly on predictors $h_1$ and $h_2$, where $h_1(x_1) = 0$ and $h_1(x_2) = 0.5$ and $h_2(x_1) = 0.5$ and $h_2(x_2) = 1$, is technically multicalibrated. However, neither $h_1$ nor $h_2$ are calibrated.*

Our second algorithm uses no-regret best-response (NRBR) dynamics to find deterministic multicalibrated predictors. We now restate and prove its guarantees from Theorem 4.2.

---

**Algorithm 2** Deterministic Multicalibration Algorithm (Theorem 4.2)

---

1: Input: $\mathcal{S} \subseteq 2^{\mathcal{X}}$, $\varepsilon \in (0,1)$, $k, \lambda, T, C \in \mathbb{Z}_+$, distribution $D$, and agnostic learning oracle $\mathcal{A}$;
2: Initialize Hedge iterate $h^{(1)} = [1/k, \ldots, 1/k]^{\mathcal{X}}$;
3: **for** $t = 1$ to $T$ **do**
4:     Let $\ell^{(t)} = \mathcal{A}_{\varepsilon/8}(c_{\mathrm{adv}}{}^{(t)}, \mathcal{G}_{\mathrm{mc}})$ where $c_{\mathrm{adv}}{}^{(t)}(\ell) := 1 - \mathcal{L}_{D,\ell}(h^{(t)})$;
5:     For every $x \in \mathcal{X}$, let $h^{(t+1)}(x) := \mathrm{Hedge}(c_x^{(1:t)})$, where
$$c_x^{(t)}(\widehat{y}) := \frac{1}{2}(1 + i^{(t)} \cdot 1[h(x) \in v^{(t)}, x \in S^{(t)}] \cdot \widehat{y}_{j^{(t)}});$$
6: **end for**
7: Take $C \ln(T/\delta)/\varepsilon^2$ samples $\mathbf{x} \sim D$ and let $t^* = \underset{t \in [T]}{\arg\min} \sum_{(x,y) \in \mathbf{x}} \ell^{(t)}(h^{(t)}, (x, y))$;
8: Return the predictor $h^{(t^*)}$;

---

**Theorem 4.2.** *Fix $\varepsilon > 0$, $\lambda, k \in \mathbb{Z}_+$, a set of groups $\mathcal{S} \subseteq 2^{\mathcal{X}}$, and a data distribution $D$. The following algorithm returns a deterministic $k$-class predictor that is $(\mathcal{S}, \varepsilon, \lambda)$-multicalibrated on $D$ and makes $O(\ln(k)/\varepsilon^2)$ calls to an agnostic learning oracle. Moreover, with probability $1 - \delta$, the oracle calls can be implemented with $\widetilde{O}\left(\frac{1}{\varepsilon^3}(\sqrt{\ln(k)} \ln(k \, |\mathcal{S}| \, /\delta) + k \ln(\lambda))\right)$ samples from $D$.*

| No-Regret vs Best-Response |
| --- |
| *Construct the problem $(\{D\}, \mathcal{G}_{\mathrm{mc}}, \mathcal{P})$ from Fact 2.5 and let $T = C\varepsilon^{-2} \ln(|\mathcal{S}| \, \lambda^k \delta)$ for some universal constant $C$. Over $T$ rounds, have a learner choose predictors $h^{(1:T)} \in \mathcal{P}$ by applying the no-regret learning algorithm of Theorem 3.7 to the stochastic costs $\ell^{(1:T)}$. Have an adversary choose $\ell^{(1:T)}$ by calling an agnostic learning oracle at each $t \in [T]$: $\ell^{(t)} = \mathcal{A}_{\varepsilon/8}(1 - \mathcal{L}_{D,(\cdot)}(h^{(t)}))$. Using $C \ln(T/\delta)/\varepsilon^2$ samples from $D$, return the predictor $h^{(t^*)}$ with the lowest empirical multicalibration error. This algorithm is written explicitly in Algorithm 2.* |

*Proof.* By Theorem 3.7, if $T \geq 256\varepsilon^{-2} \ln(k)$, the predictors $h^{(1:T)}$ guarantee the learner a regret bound of $\mathrm{Reg}_{\mathrm{weak}}(h^{(1:T)}, \{\mathcal{L}_{D,\ell^{(t)}}\}^{(1:T)}) \leq T\varepsilon/8$. Moreover, by construction, every $\ell^{(t)}$ is an $(\varepsilon/8)$ best-response to the cost $1 - \mathcal{L}_{D,(\cdot)}(h^{(t)})$. By Lemma 3.4, there must exist a timestep where $h^{(t)}$ is $(\varepsilon/4)$-optimal. By Lemma 3.5, the $h^{(t^*)}$ found by the algorithm is $(\varepsilon/2)$-optimal with probability at least $1 - \delta$. By Fact 2.5, $h^{(t^*)}$ is a $(\mathcal{S}, \varepsilon, \lambda)$-multicalibrated predictor. The oracle complexity is exactly $T$, while the sample complexity of oracle $\mathcal{A}$ is a standard adaptive data analysis result (Lemma A.6). $\qquad\square$

We note that our use of no-regret best-response dynamics recovers a multicalibration algorithm similar to the original multicalibration algorithm of [21] and which can also be found in [7, 25]. We also remark that the guarantees of Theorem 4.2 hold in weaker settings. In particular, Theorem 4.2 holds with the same analysis even if we only asked that our agnostic learning oracle $\mathcal{A}$ to be best-responding with respect to a min-max baseline. Furthermore, the last step of Algorithm 2, which explicitly samples datapoints to find timestep $t^*$, can be removed if one assumes that our agnostic learning oracle $\mathcal{A}$ signals to us when it cannot find a greater than $\varepsilon$ violation of multicalibration in the current predictor. This assumption would allow Algorithm 2 to terminate early and return the current predictor, and is assumed by prior multicalibration literature.

### D.2   Online Multicalibration

Our next algorithm uses best-response no-regret (BRNR) dynamics for online multicalibration. We now restate and prove its guarantees from Theorem 4.3.

---

**Algorithm 3** Online Multicalibration Algorithm

---

1: Input: $\mathcal{S} \subseteq 2^{\mathcal{X}}, \varepsilon \in (0,1), \lambda, T \in \mathbb{Z}_+$;
2: Initialize Hedge iterate $q^{(1)} = \text{Uniform}(\mathcal{G}_{\text{mc}})$;
3: **for** $t = 1$ to $T$ **do**
4: $\quad p^{(t)}(x) := \min\limits_{p^*(x) \in \Delta([0,\varepsilon/4\lambda,...,1])} \max\limits_{y \in [0,1]} \frac{1}{2} \mathop{\mathbb{E}}\limits_{\widehat{y} \sim p^*(x)} \left[ 1 + \mathop{\mathbb{E}}\limits_{\ell_{i,S,v} \sim q^{(t)}} [i \cdot 1[x \in S, y \in v] \cdot (\widehat{y} - y)] \right]$;
5: $\quad$ Announce predictor $p^{(t)}$ to Nature and observe Nature's data distribution $D^{(t)}$;
6: $\quad$ Sample $(x^{(t)}, y^{(t)}) \sim D^{(t)}$ and let $q^{(t+1)} := \text{Hedge}(c_{\text{adv}}^{(1:t)})$ where $c_{\text{adv}}^{(t)}(\ell) := 1 - \ell(h^{(t)}, (x^{(t)}, y^{(t)}))$;
7: **end for**

---

**Theorem 4.3.** *Fix $\varepsilon > 0$, $\lambda \in \mathbb{Z}_+$, and a set of groups $\mathcal{S} \subseteq 2^{\mathcal{X}}$. The following algorithm guarantees $(\mathcal{S}, \varepsilon, \lambda)$-online multicalibration with probability $1 - \delta$.*

---

Best-Response vs No-Regret

*Construct the online multi-objective learning problem $(\mathcal{D}, \mathcal{G}_{\text{mc}}, \mathcal{P})$ in Fact 2.6 and let $T = C\varepsilon^{-2} \ln(|\mathcal{S}| \lambda \delta)$ for some universal constant $C$. Over $T$ rounds, have an adversary choose $q^{(1:T)} \in \Delta(\mathcal{G}_{\text{mc}})$ by applying Hedge to the costs $\{1 - \ell_{(\cdot)}(p^{(t)}, (x^{(t)}, y^{(t)}))\}^{(1:T)}$, where $(x^{(t)}, y^{(t)}) \overset{\text{i.i.d.}}{\sim} D^{(t)}$. Have a learner best-respond to each stochastic cost $q^{(t)}$ with the $(\varepsilon/2)$-distribution-free best-response $p^{(t)} \in \Delta(\mathcal{P})$ of Theorem 3.8. This algorithm is written explicitly in Algorithm 3.*

---

*Proof.* By Lemma 3.1, if $T \geq 576\varepsilon^{-2} \ln(2k\lambda |\mathcal{S}|)$, the objective mixtures $q^{(1:T)}$ guarantee the adversary a regret bound of $\text{Reg}\left(q^{(1:T)}, \{1 - \ell_{(\cdot)}(p^{(t)}, (x^{(t)}, y^{(t)}))\}^{(1:T)}\right) \leq T\frac{\varepsilon}{12}$. By Lemma 3.2, there is a universal constant $C$ such that, if $T \geq C\varepsilon^{-2} \ln(2k\lambda |\mathcal{S}| / \delta)$,

$$\left| \text{Reg}\left(q^{(1:T)}, \{1 - \ell_{(\cdot)}(p^{(t)}, (x^{(t)}, y^{(t)}))\}^{(1:T)}\right) - \text{Reg}\left(q^{(1:T)}, \{1 - \mathcal{L}_{D^{(t)},(\cdot)}(p^{(t)})\}^{(1:T)}\right) \right| \leq T\frac{\varepsilon}{12},$$

with probability $1 - \delta$. Thus, $\text{Reg}\left(q^{(1:T)}, \{1 - \mathcal{L}_{D^{(t)},(\cdot)}(h^{(t)})\}^{(1:T)}\right) \leq T\varepsilon/4$ with probability $1 - \delta$.

Since each $p^{(t)}$ is a $(\varepsilon/4)$ distribution-free best-response to $q^{(t)}$, whose existence is proven by Theorem 3.8, by Lemma 3.6, the predictors $p^{(1:T)}$ are $(\varepsilon/2)$-optimal on $D^{(1:T)}$ with probability $1 - \delta$. By Fact 2.6, $p^{(1:T)}$ are also $(\mathcal{S}, \varepsilon, \lambda)$-online multicalibrated on $D^{(1:T)}$. $\qquad\square$

The high-probability condition of Theorem 4.3 can be removed if we assume nature presents data-points rather than data distributions, as is assumed in prior works. Interestingly, Algorithm 3's use of best-response no-regret dynamics exactly recovers the online multicalibration algorithm of [18, 35]. The analysis of Theorem 4.3 is, however, significantly simpler because we make explicit the role of the no-regret dynamics, whereas [18, 35] use potential arguments that ultimately prove no-regret dynamics and the multiplicative weights algorithm from scratch.

**An online-to-batch reduction.** The online multicalibration algorithm of Theorem 4.3 can also be used to obtain a non-deterministic multicalibrated predictor in batch settings through an online-to-batch reduction. This exactly recovers the non-deterministic multicalibration algorithm of [18, 35].

**Theorem D.2** (Analysis of Algorithm 3 for batch multicalibration). *Fix $\varepsilon > 0$, $\lambda \in \mathbb{Z}_+$, a set of groups $\mathcal{S} \subseteq 2^{\mathcal{X}}$, and a data distribution $D$. Simulate Algorithm 3 by having Nature choose $D$ at every timestep and return a uniform distribution $\overline{p}$ over its outputs $p^{(1:T)}$. With probability $1 - \delta$, $\overline{p}$ is $(\mathcal{S}, \varepsilon, \lambda)$-multicalibrated on $D$. This algorithm takes $O(\ln(|\mathcal{S}| \lambda/\delta)/\varepsilon^2)$ samples from $D$.*

*Proof.* By Theorem 4.3, $\max_{\ell^* \in \mathcal{G}_{\text{mc}}} \sum_{t=1}^{T} \mathcal{L}_{D,\ell^*}(p^{(t)}) \leq T\varepsilon$. Thus, the non-deterministic predictor $\overline{p}$ given by taking a uniform distribution over $p^{(1)}, \ldots, p^{(T)}$ is a $(\mathcal{S}, \varepsilon, \lambda)$-multicalibrated predictor. $\qquad\square$

Unlike the algorithm of Theorem 4.1, the predictor $\overline{p}$ output by Theorem D.2 is neither guaranteed to be succinct nor guaranteed to be of small circuit size. The lack of succinctness is because, even if $\overline{p}$'s

predicted label distribution on any feature $x \in \mathcal{X}$ is succinct, $\overline{p}$ itself can require an exponentially large support over $\mathcal{P}$. The large circuit size is because, at each timestep $t$, the learner is best-responding to a distribution $q^{(t)}$ over objectives $\mathcal{G}$ with a non-zero weight on every objective. In contrast, in the algorithm of Theorem 4.1, the learner only interacts with a single new objective $\ell^{(t)}$ at each timestep $t$.

# E  Additional Multicalibration Considerations

In this section, we present additional results on multicalibration that all follow from the same game dynamics presented in Section 3.

## E.1  Conditional Multicalibration

One shortcoming of existing definitions of multicalibration is that they measure violations marginally over the entire distribution $D$. That is, the amount that a predictor is allowed to violate calibration on a subgroup is inversely proportional to the probability mass of the subgroup, as reflected by the use of the indicator function in Definition 2.1. This can lead to predictors that are certifiably multicalibrated, but still poorly calibrated on underrepresented or minority groups. In contrast, the original definition of multicalibration proposed by [21], for which no non-trivial guarantee is known, is a *conditional* notion of multicalibration.

Below, we generalize this notion of conditional multicalibration so that setting $\mathcal{S} = \mathcal{S}'$ recovers the original multicalibration definition of [21].

**Definition E.1.** *Fix $\varepsilon > 0$, $\lambda \in \mathbb{Z}_+$, and two sets of groups $\mathcal{S}, \mathcal{S}' \subseteq 2^{\mathcal{X}}$. A $k$-class predictor $p \in \Delta(\mathcal{P})$ is $(\mathcal{S}, \mathcal{S}', \varepsilon, \lambda)$-conditionally multicalibrated for some data distribution $D$ if*

$$\forall S \in \mathcal{S}, S' \in \mathcal{S}', v \in V_{\lambda}^k, j \in [k] : \left| \mathop{\mathbb{E}}_{(x,y) \sim D, h \sim p} [(h(x)_j - \delta_{y,j}) \cdot 1[h(x) \in v, x \in S] \mid x \in S'] \right| \le \varepsilon.$$

It is not possible to obtain conditional multicalibration generally, so we will assume sample access to the conditional distributions $\{D_{S'}\}_{S' \in \mathcal{S}'}$, where $D_{S'} := D \mid x \in S'$. In practice, this assumption means that we are able to sample data from certain protected groups that may otherwise be underrepresented. For convenience, in this section, we will assume prior knowledge of $\{\Pr_D(x \in S')\}_{S' \in \mathcal{S}'}$ but note these probabilities can be cheaply estimated beforehand.

To derive conditional multicalibration algorithms from game dynamics, we first write conditional multicalibration as a multi-distribution multi-objective learning problem.

**Fact E.2.** *Let $D$ be a data distribution for some $k$-class prediction problem and fix $\varepsilon > 0$, $\lambda \in \mathbb{Z}_+$, and two sets of groups $\mathcal{S}, \mathcal{S}' \subseteq 2^{\mathcal{X}}$. Let $\mathcal{G}_{\mathrm{mc}}$ be the set of objectives $\{\ell_{i,j,S,v}\}$ as defined in Fact 2.5 and let $\mathcal{D} = \{D'_S\}_{S' \in \mathcal{S}'}$. Predictor $p \in \Delta(\mathcal{P})$ is a $\varepsilon$-optimal solution to the multi-objective learning problem $(\mathcal{D}, \mathcal{G}_{\mathrm{mc}}, \mathcal{P})$ if and only if $p$ is $(\mathcal{S}, \mathcal{S}', 2\varepsilon, \lambda)$-conditionally multicalibrated for $D$.*

*Proof.* In the multi-objective learning problem $(\{D'_S\}_{S' \in \mathcal{S}'}, \mathcal{G}_{\mathrm{mc}}, \mathcal{P})$, the multi-objective value of a predictor $p$, $\mathcal{L}^*(p) := \max_{D^* \in \mathcal{D}, \ell^* \in \mathcal{G}} \mathcal{L}_{D^*, \ell^*}(p)$, is exactly the (rescaled and shifted) magnitude of the predictor's multicalibration violation. Formally,

$$\mathcal{L}^*(p) = \frac{1}{2} + \frac{1}{2} \max_{j \in [k], S \in \mathcal{S}, S' \in \mathcal{S}', v \in V_{\lambda}^k} \left| \mathop{\mathbb{E}}_{(x,y) \sim D'_S, h \sim p} [(h(x)_j - \delta_{y,j}) \cdot 1[h(x) \in v, x \in S]] \right|.$$

The optimal multi-objective value of the problem $(\mathcal{D}, \mathcal{G}_{\mathrm{mc}}, \mathcal{P})$ is 0.5, as $\mathcal{G}_{\mathrm{mc}}$ are symmetric around 0.5 and the Bayes classifier still achieves a loss of 0.5.

$$\min_{h^* \in \mathcal{P}} \mathcal{L}^*(h^*) = \frac{1}{2} + \frac{1}{2} \max_{i \in \{\pm 1\}} i \cdot \left[ \max_{\substack{j \in [k], S \in \mathcal{S} \\ v \in V_{\lambda}^k}} \mathop{\mathbb{E}}_{\substack{(x,y) \sim D \\ h \sim p}} [(h(x)_j - \delta_{y,j}) \cdot 1[h(x) \in v, x \in S]] \right] = \frac{1}{2}.$$

Thus, $\mathcal{L}^*(p) - \min_{h^* \in \mathcal{P}} \mathcal{L}^*(h^*) = \varepsilon$ if and only if our conditional multicalibration violation is $2\varepsilon$. □

**Conditional multicalibration algorithms.** The following Theorem E.3 is, to the best of our knowledge, the first non-trivial guarantee for conditional multicalibration.

**Theorem E.3.** *Fix $\varepsilon > 0$, $\lambda, k \in \mathbb{Z}_+$, two sets of groups $\mathcal{S}, \mathcal{S}' \subseteq 2^{\mathcal{X}}$, and a data distribution $D$. The following algorithm returns a deterministic $k$-class predictor that is $(\mathcal{S}, \mathcal{S}', \varepsilon, \lambda)$-conditionally multi-calibrated on $D$ and makes $O(\ln(k)/\varepsilon^2)$ calls to an agnostic learning oracle. Moreover, with probability $1 - \delta$, the oracle calls can be implemented with $\widetilde{O}\left( \frac{1}{\varepsilon^3}(\sqrt{\ln(k)}\ln(k|\mathcal{S}||\mathcal{S}'|/\delta) + k\ln(\lambda)) \right)$ samples from each conditional distribution $\{D'_S\}_{S' \in \mathcal{S}'}$.*

---

### No-Regret vs Best-Response

*Construct the problem $(\mathcal{D}, \mathcal{G}_{\mathrm{mc}}, \mathcal{P})$ from Fact E.2 and let $T = C\varepsilon^{-2}|\mathcal{S}'|\ln(k)$ for some universal constant $C$. Over $T$ rounds, have a learner choose predictors $h^{(1:T)}$ by applying the no-regret learning algorithm of Theorem C.6 to the stochastic costs $\ell^{(1:T)}$ and distributions $\{D_{S'(t)}\}^{(1:T)}$. Have an adversary choose $S'^{(1:T)}$ and $\ell^{(1:T)}$ by calling an agnostic learning oracle at each $t \in [T]$: $S'^{(t)}, \ell^{(t)} = \mathcal{A}_{\varepsilon/8}(1 - \mathcal{L}_{D_{(\cdot)},(\cdot)}(h^{(t)}))$. Using $C\ln(T|\mathcal{S}'|/\delta)/\varepsilon^2$ samples from each distribution in $\{D'_S\}_{S' \in \mathcal{S}'}$, return the predictor $h^{(t^*)}$ with the lowest empirical multicalibration error. This algorithm is written explicitly in Algorithm 4.*

---

*Proof.* In order to apply Theorem C.6, we first observe that every objective in $\mathcal{G}_{\mathrm{mc}}$ is separable and every distribution $D'_S$ is absolutely continuous with respect to $D$, By definition, for any $S' \in \mathcal{S}'$, the Radon-Nikodym derivative of $D'_S$ with respect to $D$ at $x \in \mathcal{X}$ is $1[x \in S']/\Pr_D(x \in S')$. Thus, the sequence $h^{(1:T)}$ corresponds to running the no-regret algorithm of Theorem C.6 when $\mathcal{A}$ is chosen to be Prod. By Lemma A.4, we know that choosing the Prod algorithm as $\mathcal{A}$ satisfies the conditions of Theorem C.6 with $R(T) \in O(\sqrt{\ln(k)T})$. Theorem C.6 therefore guarantees that $\mathrm{Reg}_{B^*}(h^{(1:T)}, \{\mathcal{L}_{D_{S'(t)},\ell^{(t)}}(\cdot)\}^{(1:T)}) \le O(\sqrt{|\mathcal{S}'|\ln(k)T})$, where $B^*$ is as defined in (8). Since we can lower bound the baseline $B^* \ge 0.5$ by plugging the Bayes classifier $h^*(x) = \mathbb{E}_{(x,y) \sim D}[y \mid x]$ into (8), $\mathrm{Reg}_{\mathrm{weak}}(h^{(1:T)}, \{\mathcal{L}_{D_{S'(t)},\ell^{(t)}}(\cdot)\}^{(1:T)}) \le \mathrm{Reg}_{B^*}(h^{(1:T)}, \{\mathcal{L}_{D_{S'(t)},\ell^{(t)}}(\cdot)\}^{(1:T)}) \le O(\sqrt{|\mathcal{S}'|\ln(k)T})$. Thus, if $T \ge C|\mathcal{S}'|\ln(k)/\varepsilon^2$, $\mathrm{Reg}_{\mathrm{weak}}(h^{(1:T)}, \{\mathcal{L}_{D_{S'(t)},\ell^{(t)}}(\cdot)\}^{(1:T)}) \le T\varepsilon/8$.

By construction, every pair $S'^{(t)}, \ell^{(t)}$ is an $(\varepsilon/8)$ best-response to the cost function $1 - \mathcal{L}_{D_{(\cdot)},(\cdot)}(h^{(t)})$. By Lemma 3.4, since the learner has at most $T\varepsilon/8$ weak regret and the adversary is $\varepsilon/8$ best-responding, there must exist a timestep $t \in [T]$ where $h^{(t)}$ is $(\varepsilon/4)$-optimal. By Lemma 3.5, the $h^{(t^*)}$ found by the algorithm is $(\varepsilon/2)$-optimal with probability at least $1 - \delta$. By Fact E.2, $h^{(t^*)}$ is a $(\mathcal{S}, \mathcal{S}', \varepsilon, \lambda)$-conditionally multicalibrated predictor. The oracle complexity is exactly $T$, while the sample complexity of oracle $\mathcal{A}$ is a standard adaptive data analysis result (Lemma A.6). $\qquad\square$

We can also implement faster non-deterministic conditional multicalibration algorithms.

**Theorem E.4.** *Fix $\varepsilon > 0$, $\lambda, k \in \mathbb{Z}_+$, two sets of groups $\mathcal{S}, \mathcal{S}' \subseteq 2^{\mathcal{X}}$, and a data distribution $D$. The following algorithm returns a non-deterministic $k$-class predictor that is $(\mathcal{S}, \mathcal{S}', \varepsilon, \lambda)$-conditionally multicalibrated on $D$ and takes $O\left(|\mathcal{S}'|(\ln(|\mathcal{S}||\mathcal{S}'|/\delta) + k\ln(\lambda))/\varepsilon^2\right)$ samples in total from the distributions in $\{D'_S\}_{S' \in \mathcal{S}'}$.*

---

### No-Regret vs No-Regret

*Construct the problem $(\mathcal{D}, \mathcal{G}_{\mathrm{mc}}, \mathcal{P})$ from Fact E.2 and let $T = C\varepsilon^{-2}|\mathcal{S}'|\ln(|\mathcal{S}|\lambda^k\delta)$ for some universal constant $C$. Over $T$ rounds, have a learner choose predictors $h^{(1:T)} \in \mathcal{P}$ by applying the no-regret learning algorithm of Theorem C.6 to the stochastic costs $\ell^{(1:T)}$ and distributions $\{D_{S'(t)}\}^{(1:T)}$. In parallel, have an adversary choose $S'^{(1:T)}, \ell^{(1:T)}$ by applying ELP to costs $\{1 - \mathcal{L}_{D_{(\cdot)},(\cdot)}(h^{(t)})\}^{(1:T)}$. Return the predictor $p = \mathrm{Uniform}(h^{(1:T)})$. This algorithm is written explicitly in Algorithm 5.*

---

*Proof.* As we proved in Theorem E.3, Theorem C.6 guarantees that if $T \ge C|\mathcal{S}'|\ln(k)/\varepsilon^2$, the learner's regret is bounded by $\mathrm{Reg}_{\mathrm{weak}}(h^{(1:T)}, \{\mathcal{L}_{D_{S'(t)},\ell^{(t)}}(\cdot)\}^{(1:T)}) \le T\varepsilon/4$. We now turn to the adversary.

We will define a sequence of costs $\widetilde{c_{\mathrm{adv}}}^{(1:T)}$ that mirrors the adversary's costs $\{1 - \mathcal{L}_{D_{(\cdot),(\cdot)}}(h^{(t)})\}^{(1:T)}$. For all timesteps $t \in [T]$, rename $(x_{S'^{(t)}}^{(t)}, y_{S'^{(t)}}^{(t)}) := (x^{(t)}, y^{(t)})$ while for all other $S' \in \mathcal{S}'$ define the random samples $(x_{S'}^{(t)}, y_{S'}^{(t)}) \overset{\text{i.i.d.}}{\sim} D_{S'}$ as the data-point that the adversary would have hypothetically sampled if it had chosen to sample from $D'_S$ instead of $D_{S'^{(t)}}$ at timestep $t$. We now can define $\widetilde{c_{\mathrm{adv}}}^{(t)}(S', (i, j, S, v)) := (0.5 + 0.5 \cdot i \cdot 1[h(x_{S'}^{(t)}) \in v, x_{S'}^{(t)} \in S] \cdot (\delta_{y_{S'}^{(t)}, j} - h(x_{S'}^{(t)})_j))$. Note that, since the ELP algorithm only observes values of $c_{\mathrm{adv}}^{(t)}((i, j, S, S', v))$ where $S' = S'^{(t)}$, $\mathrm{Reg}(\{(S'^{(t)}, \ell^{(t)})\}^{(1:T)}, c_{\mathrm{adv}}^{(1:T)}) = \mathrm{Reg}(\{(S'^{(t)}, \ell^{(t)})\}^{(1:T)}, \widetilde{c_{\mathrm{adv}}}^{(1:T)})$.

By Lemma A.5, the adversary satisfies $\mathrm{Reg}(\{(S'^{(t)}, \ell^{(t)})\}^{(1:T)}, c_{\mathrm{adv}}^{(1:T)}) \leq O(\sqrt{|\mathcal{S}'| \ln(k/\delta)T})$. Thus, we also have that $\mathrm{Reg}(\{(S'^{(t)}, \ell^{(t)})\}^{(1:T)}, \widetilde{c_{\mathrm{adv}}}^{(1:T)}) \leq O(\sqrt{|\mathcal{S}'| \ln(k/\delta)T})$. By Lemma 3.2, there is a universal constant $C$ such that, if $T \geq C\varepsilon^{-2} \ln(2k\lambda^k |\mathcal{S}| |\mathcal{S}'| /\delta)$,

$$|\mathrm{Reg}(\{(S'^{(t)}, \ell^{(t)})\}^{(1:T)}, \widetilde{c_{\mathrm{adv}}}^{(1:T)}) - \mathrm{Reg}(\{(S'^{(t)}, \ell^{(t)})\}^{(1:T)}, \{1 - \mathcal{L}_{D_{(\cdot),(\cdot)}}(h^{(t)})\}^{(1:T)})| \leq T\varepsilon/12,$$

with probability $1 - \delta$. Taking a triangle inequality and union bound, we can see that for sufficiently large $C$, $\mathrm{Reg}(\{(S'^{(t)}, \ell^{(t)})\}^{(1:T)}, \{1 - \mathcal{L}_{D_{(\cdot),(\cdot)}}(h^{(t)})\}^{(1:T)}) \leq T\varepsilon/4$.

By Lemma 3.3, the ergodic iterate $h^*$ is an $(\varepsilon/2)$-optimal solution. By Fact E.2, $h^*$ is a $(\mathcal{S}, \mathcal{S}', \varepsilon, \lambda)$-conditionally multicalibrated predictor. The sample complexity is exactly $T$ since the algorithm only samples one datapoint at each iteration. $\square$

**Stronger multicalibration guarantees for (almost) free.** We can also drop the assumption of access to the conditional distributions $\{D_{S'}\}_{S' \in \mathcal{S}'}$ and ask for a weaker guarantee than conditional multicalibration where a subgroup's error tolerance scales with $\sqrt{\Pr(x \in S')}$. For simplicity, we will fix $\mathcal{S}' = \mathcal{S}$ here. Note that unlike Theorem E.3, Theorem 4.4 does not assume knowledge of the vector $[\Pr_D(x \in S)]_{S \in \mathcal{S}}$.

**Theorem 4.4.** *Fix $\varepsilon > 0$, $\lambda, k \in \mathbb{Z}_+$, two sets of groups $\mathcal{S} \subseteq 2^{\mathcal{X}}$, and a data distribution $D$. There is an algorithm that, with probability at least $1 - \delta$, takes $O(\ln(k) \cdot (\ln(|\mathcal{S}| /\varepsilon\delta) + k\ln(\lambda))/\varepsilon^4)$ samples from $D$ and returns a deterministic $k$-class predictor $h$ satisfying*

$$\left| \mathop{\mathbb{E}}_{(x,y) \sim D, h \sim p} [(h(x)_j - \delta_{y,j}) \cdot 1[h(x) \in v, x \in S]] \right| \leq \varepsilon \sqrt{\Pr(x \in S)},$$

*for all $S \in \mathcal{S}, v \in V_\lambda^k, j \in [k]$.*

The sample complexity of Theorem 4.4 should be improvable to a sample complexity of $\widetilde{O}((\sqrt{\ln(k)} \ln(|\mathcal{S}| /\delta) + k\ln(\lambda))/\varepsilon^3)$ using adaptive data analysis [30, 1]. Even without using adaptive data analysis, Theorem E.3 guarantees a sample complexity that is only a $1/\varepsilon$ factor greater (less than a cube-root increase) than the best known sample complexity for deterministic multicalibration. In fact, it matches the best sample complexity for deterministic multicalibration that is known to be possible without adaptive data analysis. This means that we can attain this strictly stronger multicalibration guarantee for free (compared to its non-adaptive data analysis counterpart) or with only a minor cube-root increase in sample complexity (compared to its adaptive data analysis counterpart). We defer a complete proof and algorithm statement to Section E.4.

## E.2 Agnostic Multicalibration

An implicit assumption of multicalibration is that one has unlimited freedom to vary their predictions based on which subgroups a datapoint belongs to. This assumption arises because we assume that groups are defined as subsets of the domain, $S \subseteq \mathcal{X}$, and predictors are allowed to condition arbitrarily on the domain. This is impractical in many settings including, for example, when subgroups correspond to protected demographics. This is an important concern for fairness applications of multicalibration.

One can extend multicalibration to a more general *agnostic* setting by assuming that group membership is passed to the predictor separately from the covariate. Note that, in this definition, different groups may not share a Bayes classifier and a fully multicalibrated predictor may not exist.

**Definition E.5.** *Consider a data distribution $D$ supported on $\mathcal{X} \times \mathcal{X}' \times \mathcal{Y}$ and where the set $\mathcal{P}$ of $k$-class predictors that are functions of the visible covariates $\mathcal{X}$ but not the protected covariates $\mathcal{X}'$.*

*Fix $\varepsilon > 0$, $\lambda \in \mathbb{Z}_+$, and a set of groups $\mathcal{S} \subseteq 2^{\mathcal{X}'}$. A (possibly non-deterministic) k-class predictor $p \in \Delta(\mathcal{P})$ is $(\mathcal{S}, \varepsilon, \lambda)$-agnostic multicalibrated with respect to a baseline $B$ if*

$$\forall S \in \mathcal{S}, v \in V_\lambda^k, j \in [k] : \left| \mathop{\mathbb{E}}_{(x,x',y)\sim D, h\sim p} [(h(x)_j - \delta_{y,j}) \cdot 1[h(x) \in v, x' \in S]] \right| \leq \varepsilon + B.$$

In the original definition of multicalibration, it is implicitly assumed that one knows, a priori, the probability that a covariate $x$ belongs to a certain group; this is necessary to attain sample complexity rates independent of domain size $\mathcal{X}$. We will similarly allow agnostic multicalibration to take for granted knowledge of the group memberships of each covariate $x \in \mathcal{X}$.

**Choosing a baseline.** One challenge with defining agnostic multicalibration is choosing a proper baseline $B$. An immediate option is to choose a min-max baseline:

$$B_{\text{weak}} := \min_{h^* \in \mathcal{H}^*} \max_{j,v,S} \left| \mathop{\mathbb{E}}_{(x,x',y)\sim D} [(h^*(x)_j - y_j) \cdot 1[h^*(x) \in v, x' \in S]] \right|.$$

However, achieving this baseline is intractable, with a sample and oracle complexity that depends— potentially linearly—on domain size. Moreover, the predictor attaining the min-max baseline is odd and undesirable: when two groups disagree on the label for a set of covariates, the predictor hedges its losses by spreading its predictions for those covariates into as many bins as possible.

**Proposition E.6.** *Define $\mathcal{X} = [m]$ for some large $m \in \mathbb{Z}_+$ and uniformly sample a random half of $\mathcal{X}$ as $\widetilde{\mathcal{X}} \subseteq \mathcal{X}$ where $|\widetilde{\mathcal{X}}| = |\mathcal{X}|/2$. Let $\mathcal{X}' = \{1, 2\}$, $\lambda = 3$ and $k = 2$, and define the marginal distributions $\Pr_D(x \mid x' = 1)$ and $\Pr_D(x \mid x' = 2)$ to be uniform on $\mathcal{X}$. For $x \notin \widetilde{\mathcal{X}}$, let labels be random: $\Pr_D(y = 1 \mid x' = 1, x) = \Pr_D(y = 1 \mid x' = 2, x) = 0.5$. For $x \in \widetilde{\mathcal{X}}$, let labels be deterministic: $\Pr_D(y = 1 \mid x' = 1, x) = 0$ and $\Pr_D(y = 1 \mid x' = 2, x) = 1$. Finding a predictor that is $(\mathcal{S}, \varepsilon, \lambda)$-agnostic multicalibrated with respect to the min-max baseline requires at least $\Theta(|x|)$ samples must be taken from $D$ for $\varepsilon = 0.1$.*

*Proof.* The min-max optimal multicalibated predictor divides $\widetilde{\mathcal{X}}$ into 4 equal parts: $\widetilde{\mathcal{X}}_1, \ldots, \widetilde{\mathcal{X}}_4$. It predicts $h(x) = 0.5$ for all $x \notin \widetilde{\mathcal{X}}$, $x \in \widetilde{\mathcal{X}}_1$ and $x \in \widetilde{\mathcal{X}}_2$. It predicts $h(x) = 1/3$ for all $x \in \widetilde{\mathcal{X}}_3$. It predicts $h(x) = 2/3$ for all $x \in \widetilde{\mathcal{X}}_4$. This attains a multicalibration violation of $1/6$. Thus, achieving an $\varepsilon$-optimal predictor requires determining the membership (in $\widetilde{\mathcal{X}}$) of at least a $3/4 - \varepsilon$ fraction of $\mathcal{X}$. $\square$

To provide a more satisfactory definition of agnostic multicalibration, we ask that one's predictor assigns the same label probabilities to datapoints with the same group memberships and relaxes the baseline so that hedging is not necessary.

**Definition E.7.** *We say that a k-predictor $h$ is agnostic multicalibrated if the predictor $h$ provides the same prediction probabilities $h(x_1) = h(x_2)$ to datapoints $x_1, x_2 \in \mathcal{X}$ that share the same group membership distributions $\forall S \in \mathcal{S} : \Pr(x' \in S \mid x = x_1) = \Pr(x' \in S \mid x = x_2)$; and (2) is $(\mathcal{S}, \varepsilon, \lambda)$-agnostic multicalibrated with respect to the multi-accuracy baseline:*

$$B_{\text{multi-acc}} := \min_{h^* \in \mathcal{H}^*} \max_{j \in [k], S \in \mathcal{S}} \left| \mathop{\mathbb{E}}_{(x,x',y)\sim D} [(h^*(x)_j - \delta_{y,j}) \cdot 1[x' \in S]] \right|. \tag{10}$$

This baseline is zero when the groups share a Bayes classifier, recovering Definition 2.1. With a slight tweak to Fact 2.5, we can write agnostic multicalibration as the following multi-objective learning problem.

**Fact E.8.** *Let $D$ be a data distribution for some k-class prediction problem and fix $\varepsilon > 0$, $\lambda \in \mathbb{Z}_+$, and a set of groups $\mathcal{S} \subseteq 2^{\mathcal{X}'}$. For every protected feature $\mathbf{x}' \in (\mathcal{X}')^{\mathcal{X}}$, direction $i \in \{\pm 1\}$, level set $v \in V_\lambda^k$, group $S \in \mathcal{S}$, and class $j \in [k]$, we define an objective $\ell_{\mathbf{x}',i,j,S,v} : \mathcal{P} \times (\mathcal{X} \times \mathcal{Y}) \to [0,1]$ where*

$$\ell_{\mathbf{x}',i,j,S,v}(h, (x,y)) = 0.5 + 0.5 \cdot i \cdot 1[h(x) \in v, \mathbf{x}'_x \in S] \cdot (h(x)_j - \delta_{y,j}), \tag{11}$$

*and $\mathcal{G}_{\text{ag}} = \{\ell_{\mathbf{x}',i,j,S,v}\}_{\mathbf{x}',i,j,S,v}$ is the set of these objectives. We also define objectives without subscript $\mathbf{x}'$, where $\ell_{i,j,S,v}(h, (x, x', y)) = 0.5 + 0.5 \cdot i \cdot 1[h(x) \in v, x' \in S] \cdot (h(x)_j - \delta_{y,j})$ and*

$\mathcal{G}'_{\mathrm{ag}} = \{\ell_{i,j,S,v}\}_{i,j,S,v}$. *For every objective $\ell \in \mathcal{G}'_{\mathrm{ag}}$, let $q_\ell \in \Delta(\mathcal{G}_{\mathrm{ag}})$ denote the objective mixture where $\mathbb{E}_{x \sim D, \ell \sim q_\ell}[\ell(\cdot)] = \mathbb{E}_{x', x \sim D}[\ell(\cdot)]$. If a predictor $p \in \Delta(\mathcal{P})$ is a $\varepsilon$-optimal solution to the multi-objective learning problem $(\{D\}, \mathcal{G}'_{\mathrm{ag}}, \mathcal{P})$ with respect to the multi-accuracy baseline, then $p$ is $(\mathcal{S}, \mathcal{S}', 2\varepsilon, \lambda)$-agnostic multicalibrated for $D$ with respect to the multi-accuracy baseline.*

A simple modification to our batch multicalibration algorithm suffices to achieve agnostic multicalibration.

**Theorem E.9.** *Fix $\varepsilon > 0$, $\lambda, k \in \mathbb{Z}_+$, a set of groups $\mathcal{S} \subseteq 2^{\mathcal{X}'}$, and a data distribution $D$. The following algorithm returns a deterministic $k$-class predictor that is $(\mathcal{S}, \varepsilon, \lambda)$-agnostic multicalibrated on $D$ with respect to the multi-accuracy baseline. It makes $O(\ln(k)/\varepsilon^2)$ calls to an agnostic learning oracle and, with probability $1 - \delta$, these oracle calls can be implemented with $\widetilde{O}\left(\frac{1}{\varepsilon^3}(\sqrt{\ln(k)}\ln(k|\mathcal{S}|/\delta) + k\ln(\lambda))\right)$ samples from $D$.*

---
No-Regret vs Best-Response

*Construct the problem $(\{D\}, \mathcal{G}_{\mathrm{ag}}, \mathcal{P})$ from Fact 2.5 and let $T = C\varepsilon^{-2}\ln(|\mathcal{S}|\lambda^k\delta)$ for some universal constant $C$. Over $T$ rounds, have an adversary choose $\ell^{(1:T)} \in \mathcal{G}'_{\mathrm{ag}}$ by calling an agnostic learning oracle at each $t \in [T]$: $\ell^{(t)} = \mathcal{A}_{\varepsilon/8}(1 - \mathcal{L}_{D,(\cdot)}(h^{(t)}), \mathcal{G}_{\mathrm{ag}})$. Have a learner choose predictors $h^{(1:T)} \in \mathcal{P}$ by applying the no-regret learning algorithm of Theorem C.6 to the stochastic costs $\{q_{\ell^{(t)}}\}^{(1:T)} \in \Delta(\mathcal{G}_{\mathrm{ag}})$. Using $C\ln(T/\delta)/\varepsilon^2$ samples from $D$, return the predictor $h^{(t^*)}$ with the lowest empirical multicalibration error. This algorithm is written explicitly in Algorithm 6.*

---

*Proof.* By Theorem C.6, if $T \geq 256\varepsilon^{-2}\ln(k)$, the predictors $h^{(1:T)}$ satisfy $\mathrm{Reg}_{B^*}(h^{(1:T)}, \{q_{\ell^{(t)}}\}^{(1:T)}) \leq T\varepsilon/8$. Note that, since predictors $h^{(1:T)}$ can only observe $x$ and not $x'$, we applied the algorithm of Theorem C.6 to (mixtures of) objectives from $\mathcal{G}_{\mathrm{ag}}$ that depend only on $(x, y)$. Since $\mathbb{E}_{x \sim D, \ell_{\mathbf{x}', i, j, S, v} \sim q_{\ell^{(t)}}}[\ell_{\mathbf{x}', i, j, S, v}(\cdot)] = \mathbb{E}_{x', x \sim D}[\ell^{(t)}(\cdot)]$ however, we can still bound regret with respect to the loss functions in $\mathcal{G}'_{\mathrm{ag}}$, which do depend on $x'$, by $\mathrm{Reg}_{B^*}(h^{(1:T)}, \ell^{(1:T)}) = \mathrm{Reg}_{B^*}(h^{(1:T)}, \{q_{\ell^{(t)}}\}^{(1:T)}) \leq T\varepsilon/8$; we define both of these regrets with respect to the set of all predictors from $\mathcal{X} \to \mathcal{Y}$. Also note that the $B^*$ baseline bounded by the multiaccuracy baseline (10):

$$B^* = \min_{h^* \in \mathcal{H}^*} \frac{1}{T} \sum_{t=1}^{T} \max_{h' \in \mathcal{H}^*} \mathop{\mathbb{E}}_{(x,y) \sim D}\left[f^{(t)}(x, h'(x)) \cdot (h^*(x) - g^{(t)}(y))\right]$$

$$\leq \min_{h^* \in \mathcal{H}^*} \max_{\ell^* \in \mathcal{G}} \max_{h' \in \mathcal{H}^*} \mathop{\mathbb{E}}_{(x,y) \sim D}[f^*(x, h'(x)) \cdot (h^*(x) - g^*(y))]$$

$$\leq \min_{h^* \in \mathcal{H}^*} \max_{j,S} \left| \mathop{\mathbb{E}}_{(x,x',y) \sim D}[(h^*(x)_j - y_j) \cdot \mathbb{1}[x' \in S]] \right|$$

$$= B_{\mathrm{multi\text{-}acc}}.$$

Thus, we have regret $\mathrm{Reg}_{B_{\mathrm{multi\text{-}acc}}}(h^{(1:T)}, \ell^{(1:T)}) \leq \mathrm{Reg}_{B^*}(h^{(1:T)}, \ell^{(1:T)}) \leq T\frac{\varepsilon}{8}$. By construction, every $\ell^{(t)}$ is an $(\varepsilon/8)$ best-response to the cost $1 - \mathcal{L}_{D,(\cdot)}(h^{(t)})$. By Lemma 3.4, there must exist a timestep where $h^{(t)}$ is $(\varepsilon/4)$-optimal with respect to $B_{\mathrm{multiacc}}$. By Lemma 3.5, the $h^{(t^*)}$ found by the algorithm is $(\varepsilon/2)$-optimal with respect to $B_{\mathrm{multiacc}}$ with probability at least $1 - \delta$. By Fact E.8, $h^{(t^*)}$ is a $(\mathcal{S}, \varepsilon, \lambda)$-agnostic multicalibrated predictor. Lastly, we verify that datapoints with identical group membership probabilities must have the same label probabilities as the update rules for their predicted labels are identical. $\square$

The best existing sample complexity bounds for agnostic multicalibration come from uniform convergence [38], and scale with $\log(|\mathcal{H}|) \approx |\mathcal{X}|$—that is, sample complexity scales with the size of one's domain. In contrast, Theorem E.9 provides a sample complexity that still depends only on the complexity of the groups on which one desires multicalibration. This can be an exponential (in $|\mathcal{X}|$) reduction in sample complexity.

Similarly to Algorithm 6, we can use the same modification to scale our non-deterministic multicalibration algorithm (Algorithm 1) to agnostic settings.

**Theorem E.10.** *Fix $\varepsilon > 0$, $\lambda \in \mathbb{Z}_+$ and a set of groups $\mathcal{S} \subseteq 2^{\mathcal{X}}$. There exists an algorithm guarantees, with probability at least $1 - \delta$, the algorithm returns a randomized $(\mathcal{S}, \varepsilon, \lambda)$-agnostic multicalibrated predictor while taking no more than $O\left((\ln(|\mathcal{S}|/\delta) + k\ln(\lambda))/\varepsilon^2\right)$ samples from $D$.*

### E.3 Moment Multicalibration

Our treatment of multicalibration, and extensions of multicalibration, hold more generally for any problem that can be written as multi-objective learning with objectives that look separable. As an example, we demonstrate that we can use the same approach to devise algorithms for *moment multicalibration*, which concerns not only learning a multicalibrated label predictor but also a second predictor that estimates the higher-order moments of the label distribution [22]. As in online multicalibration, we will work with only binary classification problems in this section, and so will say that predictors output real-valued $h_\mu(x), h_m(x) \in [0, 1]$.

**Definition E.11.** *Let $D$ be a data distribution for a binary classification problem and fix $\varepsilon > 0$, $\lambda, m \in \mathbb{Z}_+$, and a set of groups $\mathcal{S} \subseteq 2^{\mathcal{X}}$. A pair of predictors $p = (p_\mu, p_m)$ where $p_\mu, p_m : \mathcal{X} \to [0, 1]$ is $(\mathcal{S}, \varepsilon, \lambda, m)$-mean-conditioned moment-multicalibrated[5] on $D$ if, for all $S \in \mathcal{S}, v_\mu, v_m \in V_\lambda$,*

$$\varepsilon \geq \left| \mathop{\mathbb{E}}_{(x,y)\sim D,(h_\mu,h_m)\sim p} [(h_\mu(x) - y) \cdot \mathbb{1}[h_\mu(x) \in v_\mu, h_m \in v_m, x \in S]] \right|,$$

$$\varepsilon \geq \left| \mathop{\mathbb{E}}_{(x,y)\sim D,(h_\mu,h_m)\sim p} [((h_\mu(x) - y)^m - h_m(x)) \cdot \mathbb{1}[h_\mu(x) \in v_\mu, h_m \in v_m, x \in S]] \right|.$$

We will write $h = (h_\mu, h_m)$ as shorthand, where $h(x) = (h_\mu(x), h_m(x))$. Moment multicalibration can also be expressed as a multi-objective learning problem.

**Fact E.12.** *Consider the set of all pairs of predictors $\mathcal{H} = \{(h_\mu, h_m) : h_\mu, h_m : \mathcal{X} \to [0, 1]\}$. Fix $\varepsilon > 0$ and $\lambda, m \in \mathbb{Z}_+$, and a set of groups $\mathcal{S} \subseteq 2^{\mathcal{X}}$. Define the objectives $\mathcal{G}^\mu_{\mathrm{mc}} := \{\ell_{\mu,i,S,v}\}_{i\in\{\pm 1\}, S\in\mathcal{S}, v\in V^2_\lambda}$ and $\mathcal{G}^m_{\mathrm{mc}} := \{\ell_{m,i,S,v}\}_{i\in\{\pm 1\}, S\in\mathcal{S}, v\in V^2_\lambda}$ where*

$$\ell_{m,i,S,v}(h, (x, y)) = 0.5 + 0.5 \cdot i \cdot \mathbb{1}[h(x) \in v, x \in S] \cdot (h_m(x) - (y - h_\mu(x))^m),$$
$$\ell_{\mu,i,S,v}(h, (x, y)) = 0.5 + 0.5 \cdot i \cdot \mathbb{1}[h(x) \in v, x \in S] \cdot (h_\mu(x) - y).$$

*A pair of predictors is an $\varepsilon$-optimal solution to the single-distribution multi-objective learning problem $(\{D\}, \mathcal{G}^m_{\mathrm{mc}} \cup \mathcal{G}^\mu_{\mathrm{mc}}, \mathcal{P})$ if and only if the pair is also $(\mathcal{S}, 2\varepsilon, \lambda, m)$ mean-conditioned moment multicalibrated.*

Thus, we can follow the same steps as before to derive a moment multicalibration algorithm. The algorithm we derive for Theorem E.13 recovers the sample complexity of [22] for moment multicalibration. Moreover, the algorithm is parallelized in that it learns the mean and moment estimators $h_\mu, h_m$ simultaneously, in contrast with [22]'s algorithm which uses nested optimization.

**Theorem E.13.** *Fix $\varepsilon > 0$, $\lambda, r \in \mathbb{Z}_+$, a set of groups $\mathcal{S} \subseteq 2^{\mathcal{X}}$, and a data distribution $D$. The following algorithm returns a pair of predictors that are $(\mathcal{S}, \varepsilon, \lambda, m)$-mean-conditioned moment multicalibrated on $D$ and makes $O(1/\varepsilon^2)$ calls to an agnostic learning oracle. Moreover, with probability $1 - \delta$, the oracle calls can be implemented with $\widetilde{O}(m\ln(|\mathcal{S}|\lambda/\delta)/\varepsilon^4)$ samples from $D$.*

---

No-Regret vs Best-Response

*Construct the problem $(\{D\}, \mathcal{G}^m_{\mathrm{mc}} \cup \mathcal{G}^\mu_{\mathrm{mc}}, \mathcal{P})$ from Fact E.12 and let $T = C\varepsilon^{-2}\ln(|\mathcal{S}|\lambda^k\delta)$ for some universal constant $C$. Over $T$ rounds, have a learner choose predictors $h_m^{(1:T)} \in \mathcal{P}$ by applying the no-regret learning algorithm of Theorem C.6 to the stochastic costs $\ell_m^{(1:T)}$, instantiating $\mathcal{A}$ to be Hedge with learning $T^{-3/4}$. In parallel, have a learner choose predictors $h_\mu^{(1:T)} \in \mathcal{P}$ by applying the no-regret learning algorithm of Theorem C.6 to the stochastic costs $\ell_\mu^{(1:T)}$, instantiating $\mathcal{A}$ to be the strongly adaptive Hedge algorithm (Lemma A.3). Have an adversary choose $\ell_m^{(t)}, \ell_m^{(t)}$ by calling an agnostic learning oracle at each $t \in [T]$: $\ell_m^{(t)} = \mathcal{A}_{\varepsilon/8}(1 - \ell_{(\cdot)}(h^{(t)}), \mathcal{G}^m_{\mathrm{mc}})$ and*

---

[5]This definition aligns with the definition of "pseudo-moment multicalibration" proposed by [22]. We adopt this particular definition as previous studies [22, 18] develop their algorithms and analyses based on this definition of moment multicalibration, with translations to other definitions occurring only retrospectively.

$\boxed{\ell_\mu^{(t)} = \mathcal{A}_{\varepsilon/8}(1 - \ell_{(\cdot)}(h^{(t)}), \mathcal{G}_{\text{mc}}^\mu). \textit{ Using } C\ln(T/\delta)/\varepsilon^2 \textit{ samples from } D, \textit{ return the predictor } h^{(t^*)} \textit{ with the lowest empirical multicalibration error. This algorithm is written explicitly in Algorithm 7.}}$

First, we prove an intermediate result.

**Lemma E.14.** *In the Algorithm 7, the sequence of hypotheses $h^{(1:T)}$ satisfies the bounded regret condition* $\text{Reg}_{weak}(h^{(1:T)}, \{\ell_m^{(t)} + \ell_\mu^{(t)}\}^{(1:T)}) \in O\left(\sqrt{m}T^{3/4}\right)$.

*Proof.* For simplicity, we'll round $T$ to the next largest square. Recall that using Hedge to select $T$ actions from the interval $[0,1]$ with learning rate $T^{-3/4}m^{-1/2}$ guarantees a regret bound of $2\ln(2)\sqrt{m}T^{3/4}$ (Lemma A.1). By Theorem C.6, since we chose $\mathcal{A}$ to be Hedge with learning rate $T^{-3/4}$ when learning $h_\mu^{(t)}$, $\sum_{t=1}^T \ell_\mu^{(t)}(h^{(t)}) \leq 0.5T + 2\ln(2)\sqrt{m}T^{3/4}$. Similarly, by Theorem C.6, having chosen $\mathcal{A}$ to be strongly adaptive Hedge when learning $h_m^{(t)}$, we have for any $T_1 \in [T]$ and $h_m^* \in \mathcal{H}_m$,

$$\sum_{t=T_1}^{T_1+\sqrt{T}} \left( \mathcal{L}_{\ell_m^{(t)}}(h^{(t)}) - 0.5 - \mathop{\mathbb{E}}_{(x,y)\sim D}\left[ f_m^{(t)}(x, h^{(t)}(x))(h_m^*(x) - g_m^{(t)}(y)) \right] \right) \leq O\left(T^{1/4}\ln(T)\right).$$

In the above equation, $f_m^{(t)}$ and $g_m^{(t)}$ are the separable components of the loss $\ell_m^{(t)}$, as defined in Definition C.5. We now turn to bound the expectation term. By triangle inequality, $f_m^{(t)}(x, h^{(t)}(x))(h_m^*(x) - g_m^{(t)}(y)) \leq f_m^{(t)}(x, h^{(t)}(x))(h_m^*(x) - g_m^{(T_1)}(y)) + \left|g_m^{(T_1)}(y) - g_m^{(t)}(y)\right|$. Using the inequality that $|a^m - b^m| \leq m|a - b|$ when $a, b \in [0,1]$, and the movement upper bound of Hedge (Lemma A.2), we have $\left|g_m^{(T_1)}(y) - g_m^{(t)}(y)\right| \leq m\left|h_\mu^{(t)}(x) - h_\mu^{(T_1)}(x)\right| \leq m\eta_\mu\sqrt{T}$, where $\eta_\mu = T^{-1/2}T^{-3/4}$ is the learning rate of the mean predictor $h_\mu$. Thus, for all $T_1$, by choosing $h_m^*(x) = \mathbb{E}\left[g_m^{(T_1)}(y)|x\right]$, we have that

$$\sum_{t=T_1}^{T_1+\sqrt{T}} (\mathcal{L}_{\ell_m^{(t)}}(h^{(t)}) - 0.5)$$

$$\leq \min_{h_m^* \in \mathcal{H}_m} \sum_{t=T_1}^{T_1+\sqrt{T}} \mathop{\mathbb{E}}_{(x,y)\sim D}\left[ f_m^{(t)}(x, h^{(t)}(x))(h_m^*(x) - g_m^{(T_1)}(y)) \right] + mT\eta_\mu + O\left(T^{3/4}\ln(T)\right)$$

$$\leq \sqrt{m}T^{3/4} + O\left(T^{3/4}\ln(T)\right).$$

This gives $\frac{1}{2}\sum_{t=1}^T \mathcal{L}_{\ell_m^{(t)}}(h^{(t)}) + \mathcal{L}_{\ell_\mu^{(t)}}(h^{(t)}) - 1 \leq O\left((\ln(T) + \sqrt{m})T^{3/4}\right)$ as desired. $\qquad\square$

We now return to proving Theorem E.13.

*Proof of Theorem E.13.* By Lemma E.14, the learner's regret is $\text{Reg}_{\text{weak}}(h^{(1:T)}, \{\ell_m^{(t)} + \ell_\mu^{(t)}\}^{(1:T)}) \in O\left(m^{1/2}T^{3/4}\right)$. When $T = Cm^2/\varepsilon^4$, we thus have $\text{Reg}_{\text{weak}}(h^{(1:T)}, \{\max\{\ell_m^{(t)}, \ell_\mu^{(t)}\}\}^{(1:T)}) \leq T\varepsilon/8$. By construction, every $\max\{\ell_m^{(t)}, \ell_\mu^{(t)}\}$ is an $(\varepsilon/8)$ best-response to the cost $1 - \mathcal{L}_{D,(\cdot)}h^{(t)}$. By Lemma 3.4, there must exist a timestep where $h^{(t)}$ is $(\varepsilon/4)$-optimal. By Lemma 3.5, the predictors $h^{(t^*)}$ found by the algorithm are $(\varepsilon/2)$-optimal with probability at least $1 - \delta$. By Fact E.12, $h^{(t^*)}$ is a $(\mathcal{S}, \varepsilon, \lambda, m)$-mean-conditioned moment multicalibrated predictor. $\qquad\square$

---
**Algorithm 4** Conditional Multicalibration Algorithm (Theorem E.3)
---
1: Input: $\mathcal{S}, \mathcal{S}' \subseteq 2^{\mathcal{X}}, \varepsilon \in (0,1), k, \lambda, T, C \in \mathbb{Z}_+$, distributions $\{D'_S\}_{S' \in \mathcal{S}'}$, and agnostic learning oracle $\mathcal{A}$;
2: Initialize Prod iterate $h^{(1)} = [1/k, \ldots, 1/k]^{\mathcal{X}}$;
3: **for** $t = 1$ to $T$ **do**
4:     Let $S'^{(t)}, \ell^{(t)} = \mathcal{A}_{\varepsilon/8}(c_{\mathrm{adv}}^{(t)}, \mathcal{S}' \times \mathcal{G}_{\mathrm{mc}})$ where $c_{\mathrm{adv}}^{(t)}(S', \ell) := 1 - \mathcal{L}_{D_{S'}, \ell}(h^{(t)})$;
5:     Let $h^{(t+1)}(x) = \mathrm{Prod}(c_x^{(1:t)})$ where
$$c_x^{(t)}(\widehat{y}) := \frac{1[x \in S'^{(t)}]}{2\Pr_D(x \in S'^{(t)})}(1 + 1[h(x) \in v^{(t)}, x \in S^{(t)}] \cdot i^{(t)} \cdot \widehat{y}_{j^{(t)}});$$
6: **end for**
7: Take $C\ln(T/\delta)/\varepsilon^2$ samples $\mathbf{x}(S') \sim D_{S'}$ for all $S' \in \mathcal{S}'$, and let $t^* = \arg\min_{t \in [T]} \sum \ell^{(t)}(h^{(t)}, (x, y));$
8: Return the predictor $h^{(t^*)}$;

---
**Algorithm 5** Non-Deterministic Conditional Multicalibration Algorithm (Theorem E.4)
---
1: Input: $\mathcal{S}, \mathcal{S}' \subseteq 2^{\mathcal{X}}, \varepsilon \in (0,1), k, \lambda, T \in \mathbb{Z}_+$, and distributions $\{D'_S\}_{S' \in \mathcal{S}'}$;
2: Initialize Prod iterate $h^{(1)} = [1/k, \ldots, 1/k]^{\mathcal{X}}$ and ELP iterate $(S'^{(1)}, \ell^{(1)}) \sim \mathrm{Uniform}(\mathcal{S}' \times \mathcal{G}_{\mathrm{mc}})$;
3: **for** $t = 1$ to $T$ **do**
4:     Let $h^{(t+1)}(x) = \mathrm{Prod}(c_x^{(1:t)})$ where
$$c_x^{(t)}(\widehat{y}) := \frac{1[x \in S'^{(t)}]}{2\Pr_D(x \in S'^{(t)})}(1 + 1[h(x) \in v^{(t)}, x \in S^{(t)}] \cdot i^{(t)} \cdot \widehat{y}_{j^{(t)}});$$
5:     Sample $(x^{(t)}, y^{(t)}) \sim D_{S'^{(t)}}$ and let $S'^{(t+1)}, \ell^{(t+1)} := \mathrm{ELP}(c_{\mathrm{adv}}^{(1:t)})$ with
$$c_{\mathrm{adv}}^{(t)}(S', \ell_{i,j,S,v}) := \frac{1}{2}1[S' = S'^{(t)}] \cdot (1 + i \cdot 1[h(x^{(t)}) \in v, x^{(t)} \in S] \cdot (\delta_{y^{(t)}, j} - h(x^{(t)})_j));$$
6: **end for**
7: Return $h^*$, a uniform distribution over $h^{(1)}, \ldots, h^{(T)}$;

---
**Algorithm 6** Agnostic Deterministic Multicalibration Algorithm (Theorem E.9)
---
1: Input: $\mathcal{S} \subseteq 2^{\mathcal{X}}, \varepsilon \in (0,1), k, \lambda, T, C \in \mathbb{Z}_+$, agnostic learning oracle $\mathcal{A}$, and distribution $D$;
2: Initialize Hedge iterate $h^{(1)} = [1/k, \ldots, 1/k]^{\mathcal{X}}$;
3: **for** $t = 1$ to $T$ **do**
4:     Let $\ell_{i,j,S,v} = \ell^{(t)} = \mathcal{A}_{\varepsilon/8}(c_{\mathrm{adv}}^{(t)}, \mathcal{G}'_{\mathrm{ag}})$ where $c_{\mathrm{adv}}^{(t)}(\ell) := 1 - \mathcal{L}_{D, \ell}(h^{(t)})$;
5:     Let $h^{(t+1)}(x) = \mathrm{Hedge}(c_x^{(1:t)})$ with
$$c_x^{(t)}(\widehat{y}) := \frac{1}{2}(1 + \sum_{x' \in \mathcal{X}'} \Pr_D(x' \mid x) \cdot i^{(t)} \cdot 1[h(x) \in v^{(t)}, x' \in S^{(t)}] \cdot \widehat{y}_{j^{(t)}};$$
6: **end for**
7: Take $C\ln(T/\delta)/\varepsilon^2$ samples $\mathbf{x}$ from $D$, and let $t^* = \arg\min_{t \in [T]} \sum_{(x, x', y) \in \mathbf{x}} \ell^{(t)}(h^{(t)}, (x, x', y));$
8: Return the predictor $h^{(t^*)}$;

---

**Algorithm 7** Moment Multicalibration Algorithm (Theorem E.13)

---

1: Input: $\mathcal{S} \subseteq 2^{\mathcal{X}}, \varepsilon \in (0,1), k, \lambda, T, C \in \mathbb{Z}_+$, agnostic learning oracle $\mathcal{A}$, and distribution $D$;
2: Initialize Hedge iterate $h_m^{(1)} = 0.5^{\mathcal{X}}$ and Hedge iterate $h_\mu^{(1)} = 0.5^{\mathcal{X}}$;
3: **for** $t = 1$ to $T$ **do**
4:    Let $\ell_m^{(t)} = \mathcal{A}_{\varepsilon/8}(c_{\mathrm{adv}}{}^{(t)}, \mathcal{G}_{\mathrm{mc}}^m)$ and $\ell_\mu^{(t)} = \mathcal{A}_{\varepsilon/8}(c_{\mathrm{adv}}{}^{(t)}, \mathcal{G}_{\mathrm{mc}}^\mu)$ where $c_{\mathrm{adv}}{}^{(t)}(\ell) := 1 - \mathcal{L}_{D,\ell}(h^{(t)})$;
5:    For $x \in \mathcal{X}$, let $h_m^{(t+1)}(x) := \mathrm{Hedge}(c_{m,x}^{(1:t)})$ and $h_\mu^{(t+1)}(x) := \mathrm{Hedge}(c_{\mu,x}^{(1:t)})$ where

$$c_{m,x}^{(t)}(\widehat{y}) := \frac{1}{2}(1 + i_m^{(t)} \cdot 1[\widehat{y} \in v_m^{(t)}, x \in S_m^{(t)}]), \ c_{\mu,x}^{(t)}(\widehat{y}) := \frac{1}{2}(1 + i_\mu^{(t)} \cdot 1[\widehat{y} \in v_\mu^{(t)}, x \in S_\mu^{(t)}])$$

6: **end for**
7: Take $C \ln(T/\delta)/\varepsilon^2$ samples $\mathbf{x}$ from $D$ and let $t^* = \arg\min\limits_{t \in [T]} \sum\limits_{(x,y) \in \mathbf{x}} \ell_m^{(t)}(h^{(t)}, (x,y)) + \ell_\mu^{(t)}(h^{(t)}, (x,y))$;
8: Return the predictors $h_m^{(t^*)}, h_\mu^{(t^*)}$;

---

## E.4 Square-root Multicalibration Guarantees

The following Theorem E.15 is a stronger restatement of Theorem 4.4.

**Theorem E.15.** *Fix $\varepsilon > 0$, $\lambda, k \in \mathbb{Z}_+$ and sets of groups $\mathcal{S} \subseteq 2^{\mathcal{X}}$. Set $T = C \ln(k)/\varepsilon^2$ for some universal constant $C$. Algorithm 8, with probability at least $1 - \delta$, requires no more than $\widetilde{O}(\ln(k) \cdot (\ln(k |\mathcal{S}|/\varepsilon\delta) + k\ln(\lambda))/\varepsilon^4)$ samples[6] from $D$ to find a deterministic $k$-class predictor $h$ satisfying*

$$\left| \mathbb{E}_{(x,y) \sim D} [(h(x)_j - \delta_{y,j}) \cdot 1[h(x) \in v, x \in S]] \right| \leq \varepsilon\sqrt{\Pr(x \in S)},$$

*for all $S \in \mathcal{S}, v \in V_\lambda^k, j \in [k]$. That is, $h$ is a deterministic predictor that is $(\mathcal{S}, \sqrt{\Pr(x \in S)} \cdot \varepsilon, \lambda)$-multicalibrated, where the error tolerance $\sqrt{\Pr(x \in S)} \cdot \varepsilon$ depends on the group mass.*

Before proceeding to a proof, we first introduce some technical results which are simple variants of lemmas that the reader has seen previously in the manuscript. Note that, in the following multi-objective learning problem construction, we will allow negative objective values for simpler notation.

**Fact E.16.** *Let $D$ be a data distribution for some $k$-class prediction problem and fix $\varepsilon > 0, \lambda \in \mathbb{Z}_+$, and a set of groups $\mathcal{S} \subseteq 2^{\mathcal{X}}$. We define the following set of multicalibration losses:*

$$\mathcal{G}_{\mathrm{mc}}' := \left\{ \frac{1}{\sqrt{\Pr_D(x \in S)}} \cdot i \cdot (h(x)_j - \delta_{y,j}) \cdot 1[h(x) \in v, x \in S] \right\}_{i \in \pm\{1\}, j \in [k], S \in \mathcal{S}, v \in V_\lambda^k} \quad (12)$$

*Predictor $p \in \Delta(\mathcal{P})$ is a $\varepsilon$-optimal solution to the multi-objective learning problem $(\{D\}, \mathcal{G}_{\mathrm{mc}}, \mathcal{P})$ if and only if $p$ is $(\mathcal{S}, \varepsilon\sqrt{\Pr_D(x \in S)}, \lambda)$-multicalibrated for $D$.*

*Proof.* In the multi-objective learning problem $(\{D\}, \mathcal{G}_{\mathrm{mc}}', \mathcal{P})$, the multi-objective value of a predictor $p$, $\mathcal{L}^*(p) := \max_{\ell_{i,j,S,v} \in \mathcal{G}} \mathcal{L}_{D, \ell_{i,j,S,v}}(p)$, is exactly the magnitude of the predictor's multicalibration violation. Formally,

$$\mathcal{L}^*(p) = \max_{j \in [k], S \in \mathcal{S}, v \in V_\lambda^k} \frac{|\mathbb{E}_{(x,y) \sim D, h \sim p} [(h(x)_j - \delta_{y,j}) \cdot 1[h(x) \in v, x \in S]]|}{\sqrt{\Pr_D(x \in S)}}.$$

The optimal value is 0, as the objectives are symmetric around 0 and the Bayes clasifier achieves a loss of 0. Thus, $\mathcal{L}^*(p) - \min_{h^* \in \mathcal{P}} \mathcal{L}^*(h^*) \geq \varepsilon$ if and only if our multicalibration violation is at least $\varepsilon\sqrt{\Pr(x \in S)}$. $\qquad\square$

**Lemma E.17.** *Let $D$ be a data distribution for some $k$-class prediction problem, $\varepsilon \in (0, 0.6), \delta \in (0,1)$ and fix a set of groups $\mathcal{S} \subseteq 2^{\mathcal{X}}$ where, for all $S \in \mathcal{S}$, $\Pr_D(x \in S) \geq \varepsilon^2$. With only*

---

[6]Here, the tilde-O hides log-log factors.

$O(\ln(\mathcal{S}/\delta)/\varepsilon^4)$ *samples, one can find with probability at least* $1 - \delta$ *a vector* $v \in \mathbb{R}^{|\mathcal{S}|}$ *where, for all* $S \in \mathcal{S}$,

$$\left| v_S - \Pr_D(x \in S) \right| \leq \varepsilon \Pr_D(x \in S), \tag{13}$$

$$\left| \frac{1}{\sqrt{v_S}} - \frac{1}{\sqrt{\Pr_D(x \in S)}} \right| \leq \frac{\varepsilon}{\sqrt{\Pr_D(x \in S)}}, \tag{14}$$

$$\left| \sqrt{v_S} - \sqrt{\Pr_D(x \in S)} \right| \leq \varepsilon \sqrt{\Pr_D(x \in S)}. \tag{15}$$

*Proof.* Fix some sufficiently large universal constant $C$. Sample $N = C \ln(\mathcal{S}/\delta)/\varepsilon^4$ datapoints $X$ from $D$, and let $v_S = \frac{1}{|X|} \sum_{(x,y) \in X} 1[x \in S]$. Observe that $\mathbb{E}[v] = [\Pr_D(x \in S)]_{S \in \mathcal{S}}$. The multiplicative Chernoff bound says that, fixing an $S \in \mathcal{S}$,

$$\Pr(|v_S - \Pr_D(x \in S)| \geq \varepsilon \Pr_D(x \in S)) \leq 2\exp(-\varepsilon^2 N \Pr_D(x \in S)/3)$$

$$\leq 2\exp(-\varepsilon^4 N/3).$$

With our choice of $N$, taking a union bound over all $S \in \mathcal{S}$, with probability at least $1 - \delta$, we have our first claim. Taking a square-root of both sides, we also have that for all $S \in \mathcal{S}$, $\sqrt{(1-\varepsilon)\Pr_D(x \in S)} \leq \sqrt{v_S} \leq \sqrt{(1+\varepsilon)\Pr_D(x \in S)}$. The second claim then follows by observing that, for any $\varepsilon \in (0, 0.6)$, $1 - \frac{1}{\sqrt{1+\varepsilon}} \leq \varepsilon$, and $\frac{1}{\sqrt{1-\varepsilon}} - 1 \leq \varepsilon$. The third claim follows by observing that, for any $\varepsilon \in (0, 1)$, $\sqrt{1+\varepsilon} - 1 \leq \varepsilon$ and $1 - \sqrt{1-\varepsilon} \leq \varepsilon$. □

The following lemma is a modification of Theorem C.6 that states that the learner has a no-regret strategy on the calibration objectives given in Fact E.16.

**Lemma E.18.** *Consider the set of $k$-class predictors $\mathcal{P}$, a data distribution $D$, and any adversarial sequence of stochastic costs $\ell^{(1:T)} \in \mathcal{G}'_{\mathrm{mc}}$, where $\mathcal{G}'_{\mathrm{mc}}$ are the multicalibration objectives defined in Lemma E.16. Suppose you are given $v \in \mathbb{R}$ satisfying (14) and (13) for all $S \in \mathcal{S}$ and that, for all $S \in \mathcal{S}$, $\Pr_D(x \in S) \geq \varepsilon^2$. There is a no-regret algorithm that outputs (deterministic) predictors $h^{(1:T)} \in \mathcal{P}'$ such that $\mathrm{Reg}_{weak}(h^{(1:T)}, \{\mathcal{L}_{D,\ell^{(t)}}\}^{(1:T)}) \leq O(\sqrt{\ln(k)T} + \varepsilon T)$. Moreover, the algorithm does not need any samples from $D$.*

*Proof.* Consider the following algorithm. At each feature $x \in \mathcal{X}$, initialize a Prod algorithm that picks an action $h^{(t)}(x) \in \Delta(\mathcal{Y})$ at each timestep $t \in [T]$. Aggregating each algorithm's action yields our learner's overall action $h^{(t)} \in \mathcal{P}$. For each $x \in \mathcal{X}$, let $h^{(t+1)}(x)$ be the outcome of Prod at step $t+1$ after observing linear loss functions $f^{(\tau)}_{h^{(\tau)},x} : \mathbb{R}^k \to [0,1]$ for $\tau \in [t]$:

$$f^{(\tau)}_{h^{(\tau)},x}(z) := \frac{1}{2\sqrt{v_{S^{(\tau)}}}} \left( 1 + z_{j^{(\tau)}} \cdot i^{(\tau)} \cdot 1[h^{(\tau)}(x) \in v^{(\tau)}, x \in S^{(\tau)}] \right). \tag{16}$$

Prod gives $\sum_{t=1}^{T} f^{(t)}_{h^{(t)},x}(h^{(t)}(x)) - \min_{z^* \in \Delta(\mathcal{Y})} \sum_{t=1}^{T} f^{(t)}_{h^{(t)},x}(z^*) \leq C\sqrt{\ln(k)\sum_{t=1}^{T} v_{S^{(t)}}^{-1}}$ (Lemma A.4) for some universal constant $C$. Since this inequality holds for all $x \in \mathcal{X}$, applying the law of total expectation to Lemma A.4,

$$\sum_{x \in \mathcal{X}} C \Pr_D(x) \sqrt{\ln(k) \sum_{t=1}^{T} v_{S^{(t)}}^{-1}}$$

$$\geq \sum_{x \in \mathcal{X}} \sum_{t=1}^{T} \Pr_D(x) f^{(t)}_{h^{(t)},x}(h^{(t)}(x)) - \sum_{x \in \mathcal{X}} \min_{z^* \in \Delta(\mathcal{Y})} \sum_{t=1}^{T} \Pr_D(x) f^{(t)}_{h^{(t)},x}(z^*)$$

$$= \sum_{x \in \mathcal{X}} \sum_{t=1}^{T} \Pr_D(x) f^{(t)}_{h^{(t)},x}(h^{(t)}(x)) - \min_{h^* \in \mathcal{P}} \sum_{x \in \mathcal{X}} \sum_{t=1}^{T} \Pr_D(x) f^{(t)}_{h^{(t)},x}(h^*(x))$$

$$= \max_{h^* \in \mathcal{P}} \sum_{x \in \mathcal{X}} \sum_{t=1}^{T} \Pr_D(x) f^{(t)}_{h^{(t)},x}(h^{(t)}(x) - h^*(x)),$$

with the equality following because we allow arbitrary predictors. Expanding the definition of $f$,

$$\max_{h^* \in \mathcal{P}} \sum_{x \in \mathcal{X}} \sum_{t=1}^{T} \Pr_D(x) f_{h^{(t)},x}^{(t)} (h^{(t)}(x) - h^*(x))$$

$$= \max_{h^* \in \mathcal{P}} \sum_{x \in \mathcal{X}} \sum_{t=1}^{T} \Pr_D(x) \frac{1}{2\sqrt{v_{S^{(t)}}}} \left( (h^{(t)}(x) - h^*(x)) \cdot i^{(t)} \cdot \mathbb{1}[h^{(t)}(x) \in v^{(t)}, x \in S^{(t)}] \right)$$

$$\geq \sum_{t=1}^{T} \mathbb{E}_{x \sim D} \left[ \frac{1}{2\sqrt{v_{S^{(t)}}}} \left( (h^{(t)}(x) - \mathbb{E}_{y \sim D_{S^{(t)}}} [\delta_y \mid x]) \cdot i^{(t)} \cdot \mathbb{1}[h^{(t)}(x) \in v^{(t)}, x \in S^{(t)}] \right) \right]$$

$$= \sum_{t=1}^{T} \sqrt{\frac{\Pr_D(x \in S^{(t)})}{v_{S^{(t)}}}} \mathcal{L}_{D,\ell^{(t)}}(h^{(t)}).$$

By definition of $v$,

$$\sum_{t=1}^{T} \sqrt{\frac{\Pr_D(x \in S^{(t)})}{v_{S^{(t)}}}} \mathcal{L}_{D,\ell^{(t)}}(h^{(t)}) \geq \sum_{t=1}^{T} \sqrt{1-\varepsilon} \mathcal{L}_{D,\ell^{(t)}}(h^{(t)}) \geq \sum_{t=1}^{T} \mathcal{L}_{D,\ell^{(t)}}(h^{(t)}) - T\varepsilon.$$

We also know by construction of the losses in $\mathcal{G}'_{\mathrm{mc}}$ that the weak baseline $B_{\mathrm{weak}} = 0$. We can therefore bound

$$\mathrm{Reg}_{\mathrm{weak}}(h^{(1:T)}, \{\mathcal{L}_{D,\ell^{(t)}}\}^{(1:T)}) \leq T\varepsilon + \sum_{x \in \mathcal{X}} C \Pr_D(x) \sqrt{\ln(k) \sum_{t=1}^{T} \frac{1}{v_{S^{(t)}}}}$$

$$\leq T\varepsilon + \sum_{x \in \mathcal{X}} C \Pr_D(x) \sqrt{\ln(k) \sum_{t=1}^{T} (1+\varepsilon) \frac{1}{\Pr_D(x \in S^{(t)})}}$$

$$= T\varepsilon + C\sqrt{T \ln(k)(1+\varepsilon)} \sum_{x \in \mathcal{X}} \sqrt{\Pr_D(x)} \sqrt{\frac{1}{T} \sum_{t=1}^{T} \Pr_{D_{S^{(t)}}}(x)}.$$

To bound the last inequality, let $u \in \mathbb{R}^{\mathcal{X}}, v \in \mathbb{R}^{\mathcal{X}}$ be vectors defined as $u_x = \sqrt{\Pr_D(x)}$ and $v_x = \sqrt{\sum_{t=1}^{T} \Pr_{D_{S^{(t)}}}(x)}$. Then, by Cauchy-Schwarz inequality, we have that $u \cdot v \leq \|u\|_2 \|v\|_2$. In other words,

$$C\sqrt{T \ln(k)} \sum_{x \in \mathcal{X}} \sqrt{\Pr_D(x) \frac{1}{T} \sum_{t=1}^{T} \Pr_{D_{S^{(t)}}}(x)} \leq C\sqrt{T \ln(k)} \left( \sum_{x \in \mathcal{X}} \Pr_D(x) \right) \left( \sum_{x \in \mathcal{X}} \frac{1}{T} \sum_{t=1}^{T} \Pr_{D_{S^{(t)}}}(x) \right)$$

$$\leq C\sqrt{T \ln(k)}.$$

$\square$

The following lemma states that the adversary can efficiently best-respond to the objectives from Fact E.16.

**Lemma E.19.** *Consider a $k$-class predictor $h \in \mathcal{P}$, a data distribution $D$, and set of multicalibration objectives $\mathcal{G}'_{\mathrm{mc}}$. Suppose you are given $v \in \mathbb{R}$ satisfying (15) for all $S \in \mathcal{S}$. There is an algorithm that, taking only $\varepsilon^{-2}(\ln(|\mathcal{S}|/\delta) + k \ln(\lambda))$ samples for some universal constant $C$, returns a $\ell_{i,j,S,v} \in \mathcal{G}'_{\mathrm{mc}}$ such that, with probability at least $1 - \delta$, $\mathcal{L}_{D,\ell_{i,j,S,v}}(h) + 2\varepsilon \geq \max_{i^*,j^*,S^*,v^*} \mathcal{L}_{D,\ell_{i^*,j^*,S^*,v^*}}(h)$.*

*Proof.* Consider the following algorithm. Draw $N = C\varepsilon^{-2}(\ln(|\mathcal{S}|/\delta) + k \ln(\lambda))$ samples $(x,y)^{(1:N)}$ from $D$. For all $S \in \mathcal{S}$, initialize the empty buffer $X_S = \{\}$. Then, for every value of $r = 1, \ldots, N$ and for every group $S \in \mathcal{S}$, if $x^{(r)} \in S$, append $(x,y)^{(r)}$ to $X_S$. Let $\ell_{i,j,S,v} = \max_{i,j,S,v} \sqrt{\frac{v_S}{\Pr_D(x \in S)}} \frac{1}{|X_S|} \sum_{(x,y) \in X_S} \ell_{i,j,S,v}(h,(x,y))$ be the multicalibration objective

that minimizes the empirical risk on its respective buffer $X_S$. Note that finding $\ell_{i,j,S,v}$ does not require knowledge of $\Pr_D(x \in S)$, as the explicit factor of $\sqrt{1/\Pr_D(x \in S)}$ is cancelled out by $\ell_{i,j,S,v}$.

We first observe that $|X_S|$ is a binomial random variable with parameters $N$ and $\Pr(x \in S)$. By Chernoff's bound, with probability $1 - \delta |\mathcal{S}|^{-1}$, $|X_S| \geq \Pr(x \in S) \cdot C'\varepsilon^{-2}(\ln(|\mathcal{S}|/\delta) + k\ln(\lambda))$ for some universal constant $C'$, where we choose $C$ to be large enough so that $C' \geq \frac{C}{2}$. By union bound, with probability $1 - \delta$, for every $S \in \mathcal{S}$, $|X_S| \geq \Pr(x \in S)N/2$.

Condition on this event and fix a $\ell_{i,j,S,v} \in \mathcal{G}'_{\mathrm{mc}}$. We observe that each $(x, y) \in X_S$ is an unbiased sample from $D_S$, we have at least $\Pr(x \in S)N/2$ samples, and $\ell_{i,j,S,v} \in \pm\{1/\sqrt{\Pr(x \in S)}\}$. Thus, by Chernoff's bound, with probability at least $1 - \delta |\mathcal{G}'_{\mathrm{mc}}|^{-1}$,

$$
\left| |X_S|^{-1} \sum_{(x,y) \in X_S} \sqrt{\Pr(x \in S)}\ell_{i,j,S,v}(h, (x,y)) - \sqrt{\Pr(x \in S)}\mathcal{L}_{D_S, \ell_{i,j,S,v}}(h) \right|
$$
$$
\leq \varepsilon/(3\sqrt{\Pr(x \in S)}).
$$

Thus with probability at least $1 - \delta$, for all $\ell_{i,j,S,v} \in \mathcal{G}'_{\mathrm{mc}}$,

$$
\left| |X_S|^{-1} \sum_{(x,y) \in X_S} \Pr(x \in S)\ell_{i,j,S,v}(h, (x,y)) - \Pr(x \in S)\mathcal{L}_{D_S, \ell_{i,j,S,v}}(h) \right|
$$
$$
= \left| |X_S|^{-1} \sum_{(x,y) \in X_S} \Pr(x \in S)\ell_{i,j,S,v}(h, (x,y)) - \mathcal{L}_{D, \ell_{i,j,S,v}}(h) \right|
$$
$$
\leq \varepsilon.
$$

Taking a union bound over all $\ell \in \mathcal{G}'_{\mathrm{mc}}$, by uniform convergence, with probability at least $1 - 2\delta$, our returned $\ell_{i,j,S,v}$ is an $\varepsilon$ best-response to the cost function $\ell_{i,j,S,v} \mapsto -\sqrt{\frac{v_S}{\Pr_D(x \in S)}}\mathcal{L}_{D, \ell_{i,j,S,v}}(h^{(t)})$. By (15), $\max_{\ell_{i^*,j^*,S^*,v^*}} \mathcal{L}_{D, \ell_{i^*,j^*,S^*,v^*}}(h^{(t)}) - \mathcal{L}_{D, \ell_{i,j,S,v}}(h^{(t)}) \leq \varepsilon$. Thus, $\ell_{i,j,S,v}$ is an $2\varepsilon$ best-response to the cost function $\ell_{i,j,S,v} \mapsto -\mathcal{L}_{D, \ell_{i,j,S,v}}(h^{(t)})$. $\qquad\square$

*Proof of Theorem E.15.* First, we will assume without loss of generality that, for every group $S \in \mathcal{S}$, $\Pr_D(x \in S) \geq \varepsilon^2/32$. This is because for such $S$, our multicalibration constraint is trivially satisfied for tolerances of at least $\varepsilon/\sqrt{32}$. We can remove such $S$ from our group set $\mathcal{S}$ by testing if $\Pr_D(x \in S) \leq \varepsilon^2/32$; $O(\frac{\ln(|\mathcal{S}|/\delta)}{\varepsilon^4})$ samples suffices.

Next, we will sample $v$ according to Lemma E.17, which also takes at most $O(\frac{\ln(|\mathcal{S}|/\delta)}{\varepsilon^4})$ samples. Lemma E.18 then guarantees that, since $T \geq C\ln(k)/\varepsilon^2$, $\mathrm{Reg}_{\mathrm{weak}}(h^{(1:T)}, \{\mathcal{L}_{D, \ell^{(t)}}(\cdot)\}^{(1:T)}) \leq T\varepsilon/8$. By Lemma E.19, $\ell^{(t)}$ is an $(\varepsilon/8)$ best-response to the cost function $-\mathcal{L}_{D,(\cdot)}(h^{(t)})$ with probability at least $1 - \delta/T$ at each timestep $t$. By Lemma 3.4, since the learner has at most $T\varepsilon/8$ weak regret and the adversary is $\varepsilon/8$ best-responding, there is a timestep $t \in [T]$ where $h^{(t)}$ is $(\varepsilon/4)$-optimal for the problem $(\{D\}, \mathcal{G}'_{\mathrm{mc}}, \mathcal{P})$. By Lemma 3.5, the $h^{(t)}$ found by the algorithm is $(\varepsilon/2)$-optimal with probability at least $1 - \delta$. By Fact E.16, this is an $(\mathcal{S}, \varepsilon, \lambda)$-multicalibrated predictor. $\qquad\square$

# F   Other Fairness Notions

This general framework of approaching multi-objective learning problems with game dynamics can be extended beyond multicalibration. In this section, we use multi-objective learning to derive new guarantees for multi-distribution learning and multi-group learning.

---

**Algorithm 8** Conditional Multicalibration Algorithm (Theorem E.15)

---

1: Input: $\mathcal{S} \subseteq 2^{\mathcal{X}}$, $\varepsilon \in (0,1)$, $k, \lambda, T, C \in \mathbb{Z}_+$, and distribution $D$;
2: Initialize Prod iterate $h^{(1)} = [1/k, \dots, 1/k]^{\mathcal{X}}$;
3: Sample $C \ln(\mathcal{S}/\delta)/\varepsilon^4$ datapoints $X$ from $D$ and, for each $S \in \mathcal{S}$, remove $S$ from $\mathcal{S}$ if $\frac{1}{|X|} \sum_{(x,y) \in X} 1[x \in S] \leq \varepsilon^2/32$;
4: Sample $C \ln(\mathcal{S}/\delta)/\varepsilon^4$ datapoints $X$ from $D$, and let $v_S = \frac{1}{|X|} \sum_{(x,y) \in X} 1[x \in S]$ for all $S \in \mathcal{S}$;
5: **for** $t = 1$ to $T$ **do**
6:     Draw $N = C\varepsilon^{-2}(\ln(|\mathcal{S}| T/\delta) + k \ln(\lambda))$ samples $(x,y)_t^{(1:N)}$ from $D$;
7:     For all $S \in \mathcal{S}$, let $X_{t,S} = \{(x,y)_t^i \mid i \in [N], x_i \in S\}$;
8:     Let $c_{\mathrm{adv}}^{(t)}(S, \ell) := \sqrt{v_S} |X_{t,S}|^{-1} \sum_{(x,y) \in X_{t,S}} \ell(h^{(t)}, (x,y))$;
9:     Let $\ell^{(t)} = \arg\max_{\ell \in \mathcal{G}'_{\mathrm{mc}}} c_{\mathrm{adv}}^{(t)}(S, \ell)$;
10:    If $c_{\mathrm{adv}}^{(t)}(S^{(t)}, \ell^{(t)}) \leq \varepsilon/8$ terminate and return $h^{(t)}$;
11:    Let $h^{(t+1)}(x) = \mathrm{Prod}(c_x^{(1:t)})$ where
$$c_x^{(t)}(\widehat{y}) := \frac{1[x \in S^{(t)}]}{2\sqrt{v_{S^{(t)}}}}(1 + 1[h(x) \in v^{(t)}, x \in S^{(t)}] \cdot i^{(t)} \cdot \widehat{y}_{j^{(t)}});$$
12: **end for**
13: Return $p^*$, a uniform distribution over $h^{(1)}, \dots, h^{(T)}$;

---

**Competing in multi-distribution learning.** Usually, in agnostic multi-objective learning problems, the trade-off between objectives is arbitrated by the worst-off objectives $\max_{D^* \in \mathcal{D}, \ell^* \in \mathcal{G}} \mathcal{L}_{D^*, \ell^*}(\cdot)$. However, this approach to negotiating trade-offs may be suboptimal when some objectives are inherently more difficult. In those cases, we can take into account the difficulty of individual objectives by asking for a predictor where there is no objective for which a competitor $h \in \mathcal{H}^*$ performs significantly better.

**Definition F.1.** *For a multi-objective learning problem $(\mathcal{D}, \mathcal{G}, \mathcal{H})$, a solution that is $\varepsilon$-competitive with respect to a class $\mathcal{H}'$ is a hypothesis $p \in \Delta(\mathcal{H})$ satisfying,*

$$\mathcal{L}_{\mathcal{H}'}^*(p) - \min_{h^* \in \mathcal{H}} \mathcal{L}_{\mathcal{H}'}^*(h^*) \leq \varepsilon \text{ where } \mathcal{L}_{\mathcal{H}'}^*(h) := \max_{D \in \mathcal{D}} \max_{\ell \in \mathcal{G}} (\mathcal{L}_{D,\ell}(h) - \min_{h^* \in \mathcal{H}'} \mathcal{L}_{D,\ell}(h)).$$

Only a simple modification is needed to provide $\varepsilon$-competitive guarantees: amplify your original objectives $\mathcal{G}$ into a new objective set $\mathcal{G}' := \{\frac{1}{2}(1 + \ell(\cdot) - \ell(h')) \mid \ell \in \mathcal{G}, h' \in \mathcal{H}'\}$ and solve as usual. The following fact, which holds by definition, formalizes this reduction.

**Fact F.2.** *Consider a multi-objective learning problem $(\mathcal{D}, \mathcal{G}, \mathcal{H})$. For some choice of $\mathcal{H}'$, let $\mathcal{G}' := \{0.5 + 0.5(\ell(\cdot) - \ell(h')) \mid \ell \in \mathcal{G}, h' \in \mathcal{H}'\}$. Any solution $p$ that is $\varepsilon$-optimal for the multi-objective learning problem $(\mathcal{D}, \mathcal{G}', \mathcal{H})$, is also $2\varepsilon$-competitive (Definition F.1) w.r.t. $\mathcal{H}'$ for the original problem $(\mathcal{D}, \mathcal{G}, \mathcal{H})$.*

[19] showed that the sample complexity of finding an $\varepsilon$-optimal solution to a multi-objective learning problem $(\mathcal{D}, \mathcal{G}, \mathcal{H})$ is $O(\varepsilon^{-2}(\ln(|\mathcal{H}|) + |\mathcal{D}| \ln(|\mathcal{D}| |\mathcal{G}| /\delta)))$. Thus, Fact F.2 immediately implies Theorem 5.1, our sample complexity bound for finding an $\varepsilon$-competitive solution.

---
**Algorithm 9** Multi-Group Learning Algorithm (Theorem 5.3)

---

1: Input: $\mathcal{S} \subseteq 2^{\mathcal{X}}$, $\varepsilon \in (0,1)$, $\ell : \mathcal{Y} \times \mathcal{Y} \to [0,1]$, hypothesis class $\mathcal{H} : \mathcal{X} \to \mathcal{Y}$, and distribution $D$;
2: Initialize Hedge iterate $q^{(1)} = \mathrm{Uniform}(\mathcal{S} \times \mathcal{H})$ and $p^{(1)} = \mathrm{Uniform}(\mathcal{H})$;
3: **for** $t = 1$ to $T$ **do**
4:    Sample $(x_q^{(t)}, y_q^{(t)}) \sim D$ and $(x_p^{(t)}, y_p^{(t)}) \sim D$;
5:    Let $q^{(t+1)} := \mathrm{Hedge}(c_q^{(1:t)})$ and $p^{(t+1)} := \mathrm{Hedge}(c_p^{(1:t)})$ where

$$c_q^{(t)}(S,h) := \frac{1}{2} + \frac{1}{2} \underset{h' \sim p^{(t)}}{\mathbb{E}} \left[ \mathbb{1}[x_q^{(t)} \in S](\ell(h,(x_q^{(t)}, y_q^{(t)})) - \ell(h',(x_q^{(t)}, y_q^{(t)}))) \right]$$

$$c_p^{(t)}(h) := \frac{1}{2} + \frac{1}{2} \underset{S, h' \sim q^{(t)}}{\mathbb{E}} \left[ \mathbb{1}[x_p^{(t)} \in S](\ell(h,(x_p^{(t)}, y_p^{(t)})) - \ell(h',(x_p^{(t)}, y_p^{(t)}))) \right];$$

6: **end for**
7: Return: $p^*$, a uniform distribution over $p^{(1)}, \ldots, p^{(T)}$;

---

**Multi-group learning.** Consider the *multi-group learning* problem (Definition 5.2) where, rather than seeking simultaneously calibrated estimates of different subsets of the domain as in multicalibration, we seek to simultaneously minimize a general loss function on different subsets of the domain [37].

A (near) optimal sample complexity for multi-group learning of $O\left(\ln(|\mathcal{S}|\,|\mathcal{H}|)/\varepsilon^2\right)$ was attained by [39] using a reduction to sleeping experts. [39] also asked whether there exists a simpler optimal algorithm that does not rely on sleeping experts. We answer this affirmatively by designing an optimal algorithm that just runs two Hedge algorithms.

We can equate the multi-group learning problem (Definition 5.2) to finding an $\varepsilon$-competitive solution in a single-distribution multi-objective learning problem.

**Fact F.3.** *Fix $\varepsilon > 0$, a set of groups $\mathcal{S} \subseteq 2^{\mathcal{X}}$, a hypothesis class $\mathcal{H}$, data distribution $D$, and a loss $\ell : \mathcal{H} \times (\mathcal{X} \times \mathcal{Y}) \to [0,1]$. We define $\ell_S(h,(x,y)) := \ell(h,(x,y)) \cdot \mathbb{1}[x \in S]$. If a hypothesis $p \in \Delta(\mathcal{H})$ is $\varepsilon$-competitive with respect to the class $\mathcal{H}$ for the single-distribution multi-objective learning problem $(\{D\}, \{\ell_S\}_{S \in \mathcal{S}}, \mathcal{H})$, then $p$ is also an $\varepsilon$-optimal solution to the multi-group learning problem $(\mathcal{S}, \mathcal{H})$.*

The following—a formalization of Theorem 5.3—is a direct consequence of running no-regret no-regret dynamics, as in Lemma 3.3.

**Theorem F.4** (Multi-group learning). *Fix a set of groups $\mathcal{S} \subseteq 2^{\mathcal{X}}$, loss $\ell : \mathcal{Y} \times \mathcal{Y} \to [0,1]$, hypothesis class $\mathcal{H} : \mathcal{X} \to \mathcal{Y}$, and distribution $D$; For some universal constant $C$ and $T = C \ln(|\mathcal{S}|\,|\mathcal{H}|)/\varepsilon^2$, Algorithm 9 takes $2T = O\left(\ln(|\mathcal{S}|\,|\mathcal{H}|)/\varepsilon^2\right)$ samples from $D$ and returns an $\varepsilon$-optimal solution to the multi-group learning problem $(\mathcal{S}, \mathcal{H})$.*

## G   Empirical Results

### G.1   Experiment Setup

We conduct three sets of experiments to evaluate different batch multicalibration algorithms. The three sets of experiments we conduct correspond to three datasets: the UCI Adult Income dataset [26], a real-world dataset for predicting individuals' incomes based on the US Census, the UCI Bank Marketing dataset [32], a dataset for predicting whether an individual will subscribe to a bank's term deposit, and the Dry Bean Dataset [27], a dataset for predicting a dry bean's variety.

In every experiment, the performance of a multicalibration algorithm is measured based on the multicalibration violations of the average iterate and last iterate, where multicalibration violations are as defined in Definition 2.1. In every experiment, we empirically evaluate six multicalibration algorithms. Four algorithms are based on no-regret best-response dynamics, using an empirical risk minimizer as the adversary and implementing either Hedge [14] (Hedge-ERM), Prod [28] (Prod-ERM), Optimistic Hedge [36] (OptHedge-ERM), or Gradient Descent (GD-ERM) as the learner. Two algorithms are based on no-regret no-regret dynamics, using either Hedge (Hedge-Hedge) or Optimistic Hedge (OptHedge-OptHedge) as both the learner and adversary. We also note

that the most commonly used multicalibration algorithms are included in these comparisons. The original multicalibration algorithm of [21] is equivalent to GD-ERM. The revised, boosting-inspired, multicalibration algorithm of [25, 7] is equivalent to Hedge-ERM.

## G.2 Results

The results of these experiments are summarized in Table 3, which reports the multicalibration errors of each algorithm's final iterate and average iterate. In Appendix G, Figure 1 plots the evolution of training and testing multicalibration errors over the duration of the training process. We identify two key trends that are statistically significant and hold consistently in all three experiments.

| | | Error Measures | | |
|---|---|---|---|---|
| | | Train Error (Det) | Test Error (Det) | Test Error (Non-Det) |
| UCI Adult | Hedge-Hedge (NRNR) | 2.0e-2 $\pm$ 2.0e-3 | **3.0e-2** $\pm$ 3.0e-3 | **2.3e-4** $\pm$ 2.7e-5 |
| | OptHedge-OptHedge (NRNR) | 7.0e-3 $\pm$ 0.0 | **2.7e-2** $\pm$ 3.0e-3 | 2.6e-4 $\pm$ 2.8e-5 |
| | OptHedge-ERM (NRBR) | **0.0** $\pm$ 0.0 | 4.7e-2 $\pm$ 1.0e-3 | 4.8e-4 $\pm$ 9.0e-6 |
| | Hedge-ERM (NRBR) | **0.0** $\pm$ 0.0 | 6.4e-2 $\pm$ 1.0e-3 | 6.4e-4 $\pm$ 1.1e-5 |
| | Prod-ERM (NRBR) | **0.0** $\pm$ 0.0 | 5.3e-2 $\pm$ 4.0e-3 | 5.3e-4 $\pm$ 4.4e-5 |
| | GD-ERM (NRBR) | 5.3e-2 $\pm$ 1.1e-2 | 8.3e-2 $\pm$ 3.0e-3 | 9.5e-4 $\pm$ 6.5e-5 |
| UCI Bank | Hedge-Hedge (NRNR) | 2.4e-2 $\pm$ 1.0e-3 | 4.3e-2 $\pm$ 1.1e-2 | 5.3e-4 $\pm$ 1.2e-4 |
| | OptHedge-OptHedge (NRNR) | 1.3e-2 $\pm$ 1.0e-3 | **2.0e-2** $\pm$ 1.0e-3 | **2.1e-4** $\pm$ 5.0e-6 |
| | OptHedge-ERM (NRBR) | 2.0e-3 $\pm$ 1.0e-3 | **1.8e-2** $\pm$ 0.0 | 2.2e-4 $\pm$ 6.0e-6 |
| | Hedge-ERM (NRBR) | 2.0e-3 $\pm$ 0.0 | 5.2e-2 $\pm$ 1.0e-3 | 5.3e-4 $\pm$ 8.0e-6 |
| | Prod-ERM (NRBR) | **0.0** $\pm$ 0.0 | 4.6e-2 $\pm$ 3.0e-3 | 5.1e-4 $\pm$ 2.3e-5 |
| | GD-ERM (NRBR) | 8.0e-3 $\pm$ 1.0e-3 | 9.9e-2 $\pm$ 6.0e-3 | 1.1e-3 $\pm$ 7.1e-5 |
| Dry Bean | Hedge-Hedge (NRNR) | 3.2e-2 $\pm$ 5.0e-3 | **4.6e-2** $\pm$ 4.0e-3 | **2.4e-5** $\pm$ 1.0e-6 |
| | OptHedge-OptHedge (NRNR) | 1.9e-2 $\pm$ 1.0e-3 | 5.3e-2 $\pm$ 1.0e-3 | 2.7e-5 $\pm$ 1.0e-6 |
| | OptHedge-ERM (NRBR) | 1.3e-2 $\pm$ 0.0 | **5.2e-2** $\pm$ 2.0e-3 | 2.6e-5 $\pm$ 1.0e-6 |
| | Hedge-ERM (NRBR) | 1.4e-2 $\pm$ 0.0 | 5.5e-2 $\pm$ 1.0e-3 | 2.6e-5 $\pm$ 1.0e-6 |
| | Prod-ERM (NRBR) | 1.2e-2 $\pm$ 4.0e-3 | 6.5e-2 $\pm$ 1.6e-2 | 2.9e-5 $\pm$ 5.0e-6 |
| | GD-ERM (NRBR) | **6.0e-3** $\pm$ 0.0 | 7.6e-2 $\pm$ 1.0e-3 | 3.1e-5 $\pm$ 1.0e-6 |

Table 3: Average ($\pm$ standard error) of multicalibration violations on UCI Adult Dataset (20 seeds), UCI Bank Marketing Dataset (5 seeds), and the Dry Bean Dataset (5 seeds). *Train Error (Det)* and *Test Error (Det)* evaluate the last iterate (deterministic predictor) on training and test splits; *Test Error (Non-Det)* measures the average iterate (non-deterministic predictor) on the test split.

Figure 1 plots the evolution of training and testing multicalibration errors over the duration of the training process. These plots confirm that the relative performance of different multicalibration algorithms is fairly monotonic and regular, even across the duration of training and the learning rate schedule.

**The last iterates of no-regret no-regret dynamics are surprisingly multicalibrated.** On all datasets, the algorithms based on no-regret no-regret dynamics, namely Hedge-Hedge and OptHedge-OptHedge, consistently yield not only among the most multicalibrated randomized predictors (with their average iterate) but also the most multicalibrated deterministic predictors (with their last iterate). This is surprising because the last iterate of these algorithms is not guaranteed to be multicalibrated, and only enjoys a theoretical advantage over no-regret best-response algorithms in terms of average iterate guarantees. As corroborated by Figure 1, this trend does not appear to be an artifact of early stopping or learning rates, but may rather indicate that their more stable adversary updates provide regularization to these algorithms.

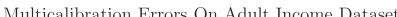

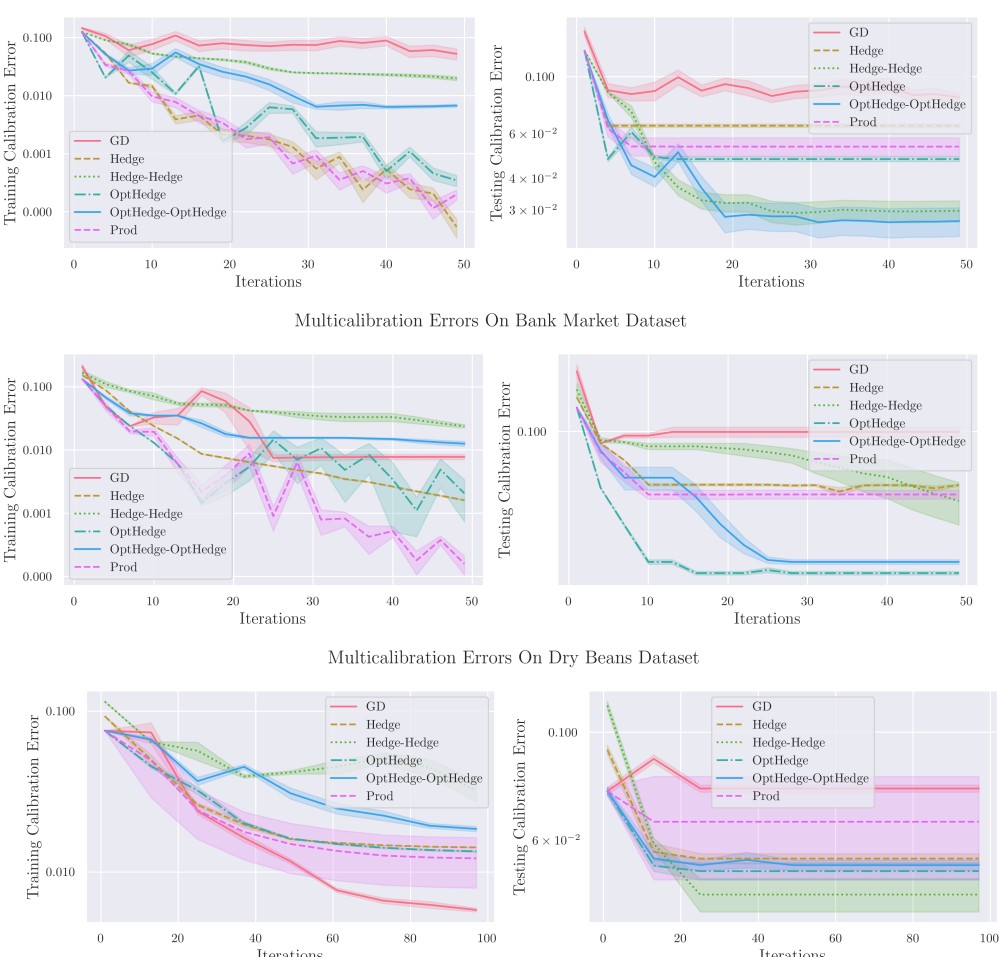

Figure 1: These plots depict the multicalibration violations (Definition 2.1) of various multicalibration algorithms on the UCI Adult Income dataset (top left), Bank Market dataset (top right), and Dry Bean dataset (bottom). The lines plot how much the current iterate violates the multicalibration condition on the training data (top plot) and testing data (bottom plot), with error bars denoting standard error. The iterates of the OptHedge-OptHedge algorithm, which implements no-regret vs no-regret dynamics using the Optimistic Hedge algorithm, are the most multicalibrated predictors.

**One's choice of no-regret algorithm matters.** On all datasets, we see that the best multicalibration results are consistently achieved by algorithms that instantiate the Optimistic Hedge no-regret algorithm. This is consistent with the theoretical results of [36], which show that Optimistic Hedge converges faster than a standard Hedge in games. We also see that the original multicalibration algorithm of [21], based on gradient descent, consistently attains the worst multicalibration errors, both in terms of average-iterate and last-iterate. This is consistent with gradient descent being a theoretically less effective no-regret algorithm, as it is unstable near the boundaries of a probability simplex.

Due to the superficial similarity between boosting and multicalibration, the field has already begun adopting multicalibration algorithms with Hedge's multiplicative updates rather than gradient descent's additive ones, as suggested by [25]. Our findings offer the first theoretical and empirical endorsement of this shift. Moreover, our results suggest that practitioners should further explore the use of Optimistic Hedge-based algorithms and algorithms based on no-regret no-regret dynamics—even when one is only interested in deterministic predictors.

### G.3 Experiment Details

The source code for these experiments is included in the repository `https://github.com/ericzhao28/multicalibration`. Model checkpoints for replicating our results can be found at `https://drive.google.com/drive/folders/1CVusrPZkB-15_55VVkoU3KrLXQGZzne4?usp=sharing`. All experiments were performed on a 2021 MacBook Pro, with a M1 Pro chip. The total compute time for replicating the experiments in this section, including hyperparameter tuning, is approximately 20 hours.

**UCI Adult Income Dataset.** The UCI Adult Income dataset [26] is a dataset for predicting individuals' incomes based on the US Census. Our experiments use the dataset's binary 'income' attribute as the target label and form 129 protected groups using eight of the dataset's labeled attributes: 'age', 'workclass', 'education', 'marital-status', 'occupation', 'relationship', 'race', and 'sex'. We perform random 80-20 train/test splits of the dataset, resulting in approximately 24000 training samples, 6000 test samples, and 130 groups. We discretize the label space into $0.1$-width bins ($\lambda = 10$). Experiments are repeated for 20 seeds, with multicalibration algorithms running for 50 iterations.

**UCI Bank Market Dataset.** The UCI Bank Marketing dataset [32] is a dataset for predicting individuals' subscriptions to a term deposit at a bank. Our experiments use the dataset's binary 'y' attribute as the target label and form 129 protected groups using eight of the dataset's labeled attributes: 'age', 'job', 'marital', 'education', 'default', 'housing', 'loan', and 'contact'. We round ages to the nearest age divisible by 5 since 'age' is a continuous attribute. We perform random 80-20 train/test splits of the dataset, resulting in approximately 36000 training samples, 9000 test samples, and 180 groups. We discretize the label space into $0.1$-width bins ($\lambda = 10$). Experiments are repeated for 5 seeds, with multicalibration algorithms running for 50 iterations.

**Dry Bean Dataset.** The Dry Bean Dataset [27] is a dataset for predicting a dry bean's variety using its physical attributes. Our experiments use the dataset's bean 'type' attribute (which takes 7 possible values) as the target label and form 80 protected groups using eight of the dataset's labeled attributes: 'perimeter', 'major-axis-length', 'minor-axis-length', 'aspect-ratio', 'eccentricity', 'convex-area', and 'equivalent-diameter'. We discretize all numerical/continuously-valued attributes by evenly dividing the range of possible values into 10 segments. We perform random 80-20 train/test splits of the dataset, resulting in approximately 11000 training samples, 2700 test samples, and 80 groups. We discretize the label space into $0.25$-width bins ($\lambda = 4$). Experiments are repeated for 5 seeds, with multicalibration algorithms run for 100 iterations.

**Hyperparameter tuning.** In each experiment, the learning rates of the algorithms are tuned on the training set using 10 seeds (Adult Income dataset), 5 seeds (Bank Market dataset), and 5 seeds (Dry Bean dataset). We sweep over the learning rate decay rates of $\eta \in [0.8, 0.85, 0.9, 0.95]$ for the learner and (if applicable) $\eta \in [0.9, 0.95, 0.98, 0.99]$ for the adversary, where the learning rate of the learner at the $t$th iteration is $\eta^t$ and the adversary is $100 \cdot \eta^t$. In the Dry Bean dataset, learning rates are universally doubled to $2\eta^t$ and the adversary is $200 \cdot \eta^t$. The selected learning rate decays are summarized below.

| Dataset | Hedge-Hedge | OptHedge-OptHedge | OptHedge-ERM | Hedge-ERM | Prod-ERM | GD-ERM |
|---|---|---|---|---|---|---|
| Adult Income | $\eta = (0.95, 0.9)$ | $\eta = (0.95, 0.9)$ | $\eta = 0.9$ | $\eta = 0.9$ | $\eta = 0.9$ | $\eta = 0.9$ |
| Bank Marketing | $\eta = (0.95, 0.95)$ | $\eta = (0.95, 0.95)$ | $\eta = 0.95$ | $\eta = 0.95$ | $\eta = 0.95$ | $\eta = 0.85$ |
| Dry Bean | $\eta = (0.99, 0.98)$ | $\eta = (0.95, 0.99)$ | $\eta = 0.95$ | $\eta = 0.95$ | $\eta = 0.95$ | $\eta = 0.95$ |

