# OpenReview forum: "A Unifying Perspective on Multi-Calibration: Game Dynamics for Multi-Objective Learning"
_NeurIPS.cc/2023/Conference — NeurIPS 2023 poster_

### Official Review · Reviewer_rNCE · 2023-06-30

**Soundness:** 2 fair
**Presentation:** 3 good
**Contribution:** 3 good
**Rating:** 4
**Confidence:** 3

**Summary:**

This paper presents a novel approach to multicalibration by leveraging game dynamics and no-regret learning algorithms. The central idea is that multicalibration can be modeled as a multi-objective learning problem where an adversary and a learner play against each other, guided by no-regret dynamics.

The authors propose three types of dynamics: no-regret no-regret (NRNR), no-regret best-response (NRBR), and best-response no-regret (BRNR), each serving different purposes. These dynamics are used to establish the multicalibration algorithms that match or improve the fastest known sample complexity rates for multicalibration, provide deterministic multicalibrated predictors, and offer online multicalibration.

This work also extends the application of these dynamics to other fairness notions, specifically multi-group learning and multi-distribution learning. For the multi-group learning problem, the authors devise an optimal multi-group learning algorithm that relies on NRNR dynamics and is simpler than the existing approaches.

The authors validate their theoretical claims with empirical results on a few standard datasetst. The results highlight the importance of effective no-regret algorithms for better multicalibration, with the Optimistic Hedge outperforming others in the no-regret no-regret dynamics.

Overall, this paper contributes to the field by providing a unified, game-theoretic framework for multicalibration that unites disparate existing results and offers more efficient solutions, both in theory and practice. Moreover, it highlights the broader applicability of game dynamics and no-regret algorithms to other fairness notions in machine learning.

**Strengths:**

The paper introduces a novel perspective by approaching multicalibration and other fairness notions using game dynamics and no-regret learning algorithms. It presents multicalibration as a multi-objective learning problem within a game-theoretic framework. However, while the application to multicalibration is new, the use of no-regret algorithms and game-theoretic models is well-established in other areas of calibration literature.

The paper offers a technically sound approach with rigorous mathematical derivations. The theorems are well-proved, and the proposed algorithms are clearly detailed. However, the empirical section could be expanded upon to strengthen the validation of the theoretical claims with real-world data.

The authors have done a reasonable job in articulating complex concepts and methodologies, with the paper being generally well-structured. The use of tables aids understanding, though more insights can be provided for the technical results.

The work has the potential to unify and extend prior work on multicalibration and other fairness notions. However, the impact of the work may depend heavily on how effectively these theoretical insights can be translated into practical applications. Furthermore, while the findings from the empirical results could guide the selection of no-regret algorithms in multicalibration tasks, the experimental validation is relatively limited and does not fully exploit the range of situations where these algorithms could be applied.

**Weaknesses:**

The paper provides a theoretically strong development of multicalibration algorithms using game dynamics. However, there seems to be a gap between the theoretical development and the empirical results. The authors could improve this aspect by designing more comprehensive experiments that test a wide range of scenarios to validate their theoretical claims. These could include different types of datasets, varying levels of complexity, and possibly real-world use cases.

The experimental evaluation is currently limited to only a few standard but simple dataset. This may not sufficiently test the robustness of the proposed algorithms under different conditions. More experiments with diverse datasets would provide a better understanding of the algorithms' performance and potential limitations. Additionally, it would be beneficial to compare the proposed methods with more baseline or state-of-the-art algorithms for multicalibration to understand the relative performance.

Also, the paper could be improved by providing more detailed descriptions of the proposed algorithms. While the authors do discuss the high-level ideas behind using game dynamics for multicalibration, it may be beneficial for readers to have more specific details about how these algorithms are implemented.

Although the authors discuss several previous works, it is not entirely clear how their contributions improve upon or differ from these existing methods. Besides theoretical comparison, a clearer empirical comparison would make the authors' contributions more evident.

The paper could benefit from a more explicit discussion of the limitations of the proposed methods.

**Questions:**

The paper provides an improved theoretical framework for multicalibration using game dynamics. However, it seems there's a disconnect between the theory and empirical results presented. Could the authors elaborate on why specific experiments were chosen and how they directly validate the theoretical developments?

The empirical evaluation mainly focuses on the standard datasets like UCI. How do the authors envision their proposed algorithms performing on other types of datasets, especially those with different characteristics, such as image datasets or high-dimensional problem? Could they provide any insights or theoretical guarantees on their algorithms' robustness across varied datasets?

The high-level descriptions of the proposed algorithms are appreciated, but having more concrete, step-by-step details of the algorithm implementations would be very helpful. Could the authors provide pseudocode or a more in-depth explanation of their proposed algorithms?

The paper could benefit from a more detailed comparison with previous works. How do the authors' contributions improve upon or differentiate from existing methods, specifically in terms of performance and applicability?

The paper presents a novel framework that generalizes previous works. How far can this generalization go? Are there any theoretical or practical constraints that limit its applications? Are there any lower bounds? Any insight into this would be beneficial for the understanding and application of the framework.







**Limitations:**

The authors do not provide a clear discussion of the limitations of their work or potential negative societal impacts, which is a critical part of any comprehensive research paper. This is an area where the paper could be improved. Here are a few suggestions for potential limitations and societal impact considerations:

Discuss the computational complexity and scalability of the proposed methods. This is crucial for real-world applications, particularly when dealing with large and complex datasets.

Given that the paper deals with fairness notions in learning, the authors could consider discussing the implications of misapplication of these methods. While the goal is to improve fairness, any shortcomings or misuse could unintentionally lead to bias or unfair outcomes.

---

> ### Author Rebuttal · Authors · 2023-08-10
>
> Thank you for your review. Below we respond to your questions.
>
> **On providing Pseudocode**\
> We already provide detailed step-by-step pseudocode for every algorithm in the Appendix: see Algorithms 2, 3, 4, 5, 6, 7. We also have fully released our source code for the experiments (see supplemental material).
> On including experiments comparing to state-of-the-art algorithms in multicalibration
> Our experiments already include comparisons to all existing and baseline multicalibration algorithms (see lines 1189-1191 of the appendix for further clarification). An important aspect of our framework is that it recovers all existing multicalibration algorithms. In the current presentation of Table 2, we refer to existing baseline algorithms by the name of the dynamics they take in our framework, e.g., the original multicalibration algorithm of HKRR18 is referred to as GD-ERM and the newer multicalibration algorithm that has become the community’s de facto standard (see e.g., KGZ19, DKRRY21) is referred to as Hedge-ERM. To clarify that these are indeed the baseline algorithms, we will add the appropriate citations on those row numbers in the final version of the paper.
>
> **On the choice of experiment datasets**\
> Our experiments, which evaluate on three datasets including both binary and multi-class problems, are among the most thorough in the multicalibration literature. Most prior work in this space is purely theoretical with no empirical evaluations. In cases where prior work in multicalibration has done empirical evaluations, those evaluations have only used toy simulated datasets (e.g. in GKSZ22). Indeed, our work includes the most extensive empirical evaluation of existing and introduced algorithms in multicalibration to date. This is in addition to the main contribution of our paper, which is to unify the theory of multicalibration.
>
> **On the lack of deep learning results**\
> Running deep learning experiments for high-dimensional image data sets is both beyond the scope of this work and of limited relevance to multicalibration. To the best of our knowledge multicalibration has never been successfully scaled to deep learning settings before (though we note multi-accuracy, a related but much easier problem, has). Moreover, UCI datasets present a more relevant empirical setting since they include demographic features that provide meaningful proxies for the definitions of groups in multicalibration.
>
> **On misapplication of multicalibration as a fairness technique**\
> Multicalibration has long been used as a method of algorithmic fairness and is an established line of work (see HKRR18). Overall, we are interested in improving the theory and design of algorithms for multicalibration that lead to better statistical and computational bounds. Note that in Section E.2 we do address some—to the best of our knowledge—previously overlooked shortcomings of multicalibration as a fairness technique and propose some solutions. However, a broad discussion of misapplication of statistical sample complexity bounds is not in the scope of this work.
>
> **References**\
> [HKRR18] U. Hebert-Johnson, M. P. Kim, O. Reingold, and G. N. Rothblum. Multicalibration: Calibration for the (computationally-identifiable) masses. In J. G. Dy and A. Krause, editors, Proceedings of the International Conference on Machine Learning (ICML), Proceedings of Machine Learning Research, pages 1944–1953. PMLR, 2018\
> [GKSZ22] P. Gopalan, M. P. Kim, M. Singhal, and S. Zhao. Low-degree multicalibration. In P.L. Loh and M. Raginsky, editors, Proceedings of the Conference on Learning Theory (COLT), pages 3193–3234. PMLR, 2022.

---

### Official Review · Reviewer_pd5T · 2023-07-07

**Soundness:** 3 good
**Presentation:** 2 fair
**Contribution:** 3 good
**Rating:** 6
**Confidence:** 4

**Summary:**

The paper provides a two-player dynamics framework that seeks to unify many strands of recent work on multicalibration and multiobjective optimization. With three possible setups considered: No regret against No regret, Best response against No regret, and Best response against Best response, efficient algorithms are provided to demonstrate that these generic setups apply in particular to (multi-)calibration-like constraints. The framework is then applied to several existing and new settings in the literature with the goal of improving some convergence bounds or, respectively, simplifying and streamlining the analysis of specific algorithms. Some experiments are given that show what happens when proposed dynamics are run with specific well-known no regret algorithms in the driver's seat.

**Strengths:**

To me, the main strength of this paper is the rather clean nature of the setup, which promises that by combining, in any combination, best-responding and no-regret agents, it is possible (though not without further work) to recover existing multigroup fairness results. Previous guarantees are (with a couple exceptions) improved somewhat marginally, or greater simplicity of the framework is claimed in some other cases, but overall it is the generality of this framework --- showcased by its adaptability and ability to recover various insights about calibration (not just regret bounds, but also questions of the simplicity of the output calibrated model, amount of randomness required, etc) --- that is the paper's main forte.

Additionally, the conditional multicalibration setup presented in the paper (and generalizing previous conditional results in the quantile setting) looks like an interesting addition to the literature.

**Weaknesses:**

No particular weaknesses, other than the relatively well-studied nature of, and relatively small gains in, some of the applications (but no big deal), as well as the somewhat terse and cramped presentation.

For instance, there are a few "intuitive-sounding" claims made about where NRNR, BRNR, BRBR may be applied to the greatest utility, but the actual applications are then scattered throughout the paper rather than recalled immediately (there is also Table 1, but I would appreciate a more intuitive dive into which dynamic was used where and why, at the point where these dynamics were actually introduced). For another example, I would be interested in further comments on potential alternative definitions of conditional multicalibration and a slightly expanded treatment of the comparison to the existing 1/sqrt(# people in group) type of guarantee.

Still, the paper is overall solid and well-done and I enjoyed reading it.

**Questions:**

See the above section.

---

> ### Author Rebuttal · Authors · 2023-08-10
>
> We thank you for your positive review. We appreciate your feedback on including a discussion about which dynamics are used where and lead to which improvements—we will make some revisions to further improve the flow. We also appreciate your suggestion to include additional comments on our sqrt-conditional multicalibration result (Corollary E.4).
>
> **On further details for conditional multicalibration**\
> We would like to highlight the following points regarding Corollary E.4 to clarify its significance, as we believe it is an example of a significant theoretical improvement that our framework yields as a consequence of  its versatility.
>
> Although existing multicalibration algorithms only guarantee an error bound that—for each group $G$—scales inversely with the group’s probability mass $\Pr(x \in G)$, we show **it is possible to obtain an error bound that scales with $1 / \sqrt{\Pr(x \in G)}$ rather than $1 / \Pr(x \in G)$**. We are the first to obtain this stronger guarantee (Corollary E.4). We achieve this without introducing any additional assumptions and with no increase in the sample complexity (compared to the sample complexity of algorithms that do not use adaptive data analysis) or only a cube-root increase in sample complexity (compared to algorithms that use adaptive data analysis). Even more interestingly, this result is obtained by simply replacing first-order regret bounds with second-order regret bounds in our game dynamics. Please let us know if you have further questions and comments about these results.
>
> Based on your feedback, we plan to add to the section some of these clarifying comments. We will also include refinement of Corollary E.4 which should be more readable, removes an extraneous $\sqrt{S’}$ factor from the sample complexity of Corollary E.4, and makes clear the corollary does not need to assume knowledge of $\Pr(x \in S)$ nor make any other assumptions.

---

> > ### Comment · Reviewer_pd5T · 2023-08-16
> > **Acknowledgment**
> >
> > Thank you for the reply. With the expectation that the flow of the paper will be improved --- especially with regards to the uses of and intuition on various combinations of no regret (NR) and best responding (BR) agents --- I will keep my current score.

---

### Official Review · Reviewer_ymik · 2023-07-14

**Soundness:** 4 excellent
**Presentation:** 4 excellent
**Contribution:** 3 good
**Rating:** 6
**Confidence:** 3

**Summary:**

This work exploits connections to game dynamics to propose a unifying algorithmic framework to address the multicalibration problem which has been recently used for tackling fairness concerns in machine learning. More precisely, based on the classic game dynamics approach used in learning problems, it is shown that multicalibration results can be seen as learning dynamics for two-player zero-sum games relying on no-regret algorithms or best response dynamics. Using this unifying framework, the paper recovers some guarantees for existing multicalibration algorithms with simplified proofs and results. Multi-objective learning guarantees are shown and new guarantees are also established for several multicalibration settings including in particular an exponential reduction in the complexity of k-class multicalibration over prior work. Experiments were conducted to evaluate the empirical performance of multicalibration algorithms on some real-world datasets.


**Strengths:**

-  The paper proposes a unifying approach using game dynamics which recovers prior results with simpler proofs and establishes novel guarantees improving over prior work in multiple settings.
- This paper is well-written, the presentation is very clear.
- While I am not familiar with the calibration/multicalibration literature and I did not go through the long appendix supporting the main part in details, the results are sound, quite rigorously exposed and the proofs which leverage for instance prior online learning results seem solid to the best of my knowledge.
- Several additional experimental results are also provided in the appendix to support the theoretical findings.

**Weaknesses:**

Regarding novelty, the related work section mentions in l. 55-56 that ‘no work has established a broad connection between no-regret learning and multicalibration’. No-regret learning has been priorly used for calibrated forecasting as acknowledged by the paper. While it is mentioned that multicalibration ‘has very different challenges than calibrated forecasting’, the discussion regarding these is reduced to a single line l. 57-58 in the related work section. Given that one of the main contributions of this work is the unifying framework based on the connection with game dynamics, I would expect a more detailed discussion to further clarify the novelty with respect to prior work and calibrated forecasting regarding this particular aspect. For instance, some proofs such as the one of Theorem 3.8 follow similar lines to the proof of calibration as also mentioned in l. 246-247. Moreover, it seems that this connection to game dynamics was known for multi-objective learning and multicalibration is linked to it via the simple facts 2.5 and 2.6.

**Questions:**

- The game dynamics approach to multi-objective learning is a ‘common approach’ as mentioned in l. 138. The paper connects multicalibration to multi-objective learning in Facts 2.5 and 2.6 which leads to a connection between multicalibration and the game dynamics approach. Are Facts 2.5 and 2.6 novel results? Is there a particular reason for using ‘Facts’ for these results instead of ‘Lemma’ for instance? Is this related to the simplicity of the proofs in Appendix B once the losses are defined as in the facts?


- How do you justify the equality in l. 230-231?

- Section 3.1 is about multicalibration but section 3 is about multi-objective learning. Section 4 is 'Multicalibration with Game Dynamics'. Would it be more appropriate to move section 3.1 to section 4?

**Minor questions and suggestions:**

- Why is the baseline of (2) not also used in (1) so that the formulations are unified?

- The notations max and min are used for minimization over a hypothesis class or maximization over a class of loss functions and a set of data distributions. How are these guaranteed to be reached without further assumptions? Are the classes supposed to be finite or are the more general notations sup and inf more appropriate? For instance, cardinalities are used in Lemma 3.5 while the preliminaries do not seem to specify the nature of the classes.

- Minor suggestion: in Definition 2.2., you could keep the same notation as in Definition 2.1 and only change the dependence on $t$ for $D$ and $p$ to ease the reading (with the same order for the quantities and a single indicator function).

 - l. 82: $\Delta(\mathcal{Y})^{\mathcal{X}}$ with parenthesis.


**Limitations:**

Limitations of the results do not seem to be clearly discussed throughout the paper to the best of my knowledge.

---

> ### Author Rebuttal · Authors · 2023-08-10
>
> Thank you for your review. Below, we address your comments and questions.
>
> **On game dynamics being a common approach for multi-objective learning.**\
> While it appears intuitive that game dynamics must have a role to play in multicalibration, given that no-regret learning plays a role in calibrated forecasting, it has taken 5 years and 50+ papers in this space for our work to be the first to make this connection in a broad and unified way.
>
> The nontrivial underlying challenge is not the mere reduction to min-max optimization—which is why we refer to Facts 2.5 and 2.6 as facts and not lemmas/theorems—but rather in solving the resulting equilibrium computation problem in a way that connects to practical algorithms. The needs of multicalibration (such as determinism, large and complex predictor space, etc.) differ significantly in this regard from earlier applications of general-purpose no-regret algorithms and game dynamics. Most notably, the minimizing player’s action set is the set of all predictors, which scales exponentially in the domain size $|X|$! We must therefore construct a novel and highly nontrivial online learning strategy for the minimizing player with a regret that is independent of the size of the action set and $|X|$. Moreover, this no-regret algorithm must use non-randomized predictors against an adaptive adversary (Theorem 3.7). Another example is that we need to obtain a deterministic solution from the game dynamics despite not having convexity, which required us to introduce a novel form of no-regret/best-response game dynamics (Lemma 3.4) that, to the best of our knowledge, has not been studied previously.
>
> The key to our simple proofs are not Facts 2.5 and 2.6 but rather Theorems 3.7 and Lemma 3.4, and other lemmas in Section 3, that allow us to use different dynamics to address various specific needs and settings in multicalibration. None of these theoretical tools come from (or were even needed) in calibrated forecasting. Moreover, in section 3, we highlight how alternative game dynamics and benchmarks (such as notions of weak regret and deterministic solutions) can lead to equilibrium notions that are more in line with the needs of multicalibration. It is against this backdrop of alternative game dynamics that we introduce Theorem 3.7 that is the key to many of our results.
> We agree that Theorem 3.8 is inspired by a similar lemma in calibrated forecasting, as we mentioned in the paper. But Theorem 3.8 goes further, providing a connection between online multicalibration and the calibrated forecasting literature that is new to our work. Indeed, we highlight that Theorem 3.8 is primarily used by our online multicalibration results.
>
> **Equality on Line 230-231**
>
> This equality is because we can choose $h^*$ so that $h^*(x) = z^*$. Intuitively, this equality arises because we are optimizing over all well-defined predictors $h: X\rightarrow Y$.
>
> **The baseline of (2) vs baseline of (1)**\
> The baseline of (2) is much larger than the baseline of (1). In fact, the baseline of (2) would be trivial to attain in an offline batch setting, whereas the baseline of (1) is impossible to attain in an online setting.
>
> **On related work.**\
> We appreciate your feedback and will amend the related works to emphasize that the technical tools our results rely on differ significantly from those explored in the calibrated forecasting literature. Below, we highlight some clarifying examples.
> * Non-deterministic predictors are necessary in calibrated forecasting, whereas multicalibration is usually defined for deterministic predictors. We needed to design novel game dynamics that produce deterministic solutions since, as we note above, we could not rely on convexity to perform online-to-batch reductions. These determinism/non-determinism concerns do not arise at all in calibrated forecasting.
> * The concept of calibrated forecasting does not require one to reason about a covariate distribution. In contrast, multicalibration is a form of supervised learning and is—at its asymptotic limit—exactly the task of learning a Bayes-optimal classifier. As a result, calibrated forecasting requires solving only a simple online learning problem over 2 actions—predicting 0 or predicting 1—while multicalibration requires online learning on an action set that grows exponentially with the size of one's domain $2^{|X|}$.
> * Calibrated forecasting reduces to an online learning problem where sublinear regret is achievable [Fos99, Har22]. Multicalibration reduces to an online learning problem where sublinear regret is not always achievable. As a result, we had to introduce notions such as weak regret (Line 150).

---

> > ### Comment · Reviewer_ymik · 2023-08-14
> > **Acknowledgement**
> >
> > I thank the authors for their clarifications regarding related work and concerning the challenges raised by multicalibration compared to calibrated forecasting. Most of my questions have been answered. I maintain my positive score.

---

### Official Review · Reviewer_Mwev · 2023-07-18

**Soundness:** 4 excellent
**Presentation:** 3 good
**Contribution:** 3 good
**Rating:** 7
**Confidence:** 1

**Summary:**

The authors proposed a unified framework for multicalibration learning by exploiting its connection to the game dynamics in multi-objective learning. Strong theoretical guarantees were given and its extension to address group fairness was discussed.

**Strengths:**

1. The analysis of the game dynamics in multi-objective learning was novel and strong.
2. Connection between multicalibration learning and game dynamics in multi-objective learning was well exploited, unified framework was given.
2. The author gave a clear presentation of the key ideas of the work despite substantial material.

**Weaknesses:**

1. The experiment parts seem to compare with algorithms within the proposed framework. Is there any comparison with existing baseline algorithms?
2. How does the multi-objective learning discussed in the paper related to the Pareto optimal one?
3. Given different learner choices shown in the empirical section, is there any learner who fits the proposed framework better?

**Questions:**

see weakness.

---

> ### Author Rebuttal · Authors · 2023-08-10
>
> We thank you for your positive review. Below, we address your comments and questions.
>
> **On experiment comparisons to baselines existing algorithms.**\
> Our experiments already include comparisons to existing and baseline multicalibration algorithms (see lines 1189-1191 of the appendix for further clarification). An important aspect of our framework is that it recovers **all existing multicalibration algorithms**. In the current presentation of Table 2, we refer to existing baseline algorithms by the name of the dynamics they take in our framework, e.g., the original multicalibration algorithm of HKRR18 is referred to as GD-ERM and the newer multicalibration algorithm that has become the community’s de facto standard (see e.g., KGZ19, DKRRY21) is referred to as Hedge-ERM. To clarify that these are indeed the baseline algorithms, we will add the appropriate citations on those row numbers in the final version of the paper.
> We note that our experiments show that using optimistic Hedge and no-regret/no-regret dynamics (as a concrete proposal for an algorithm in our framework) provides better empirical performance than both of these existing baselines.
>
> **Is there a learner which empirically does better?**\
> As discussed above, all of the algorithms discussed in the empirical section are derived from our framework, reflecting the generality of the framework. In terms of which specific algorithms are the most empirically successful, we find that playing optimistic hedge against optimistic hedge (Opt-Hedge Opt-Hedge NRNR) is consistently the most effective algorithm. We emphasize that this choice of algorithm had not been considered in the past, and it is through the unifying feature of our framework that its importance has come to light.
>
> **On the relationship of multiobjective learning to pareto optimality.**\
> The multi-objective learning discussed in the paper is concerned with attaining a min-max value rather than finding a strategy guaranteed to be on the Pareto front. We did not need to additionally consider the stronger condition of Pareto optimality in multiobjective optimization because the connection between multi-objective learning and multicalibration goes through the min-max value alone. We therefore do not include Pareto optimality in our definition of multiobjective learning. This is also consistent with recent works in multi-objective learning such as [HJZ22].
>
> **References**\
> [HKRR18] U. Hebert-Johnson, M. P. Kim, O. Reingold, and G. N. Rothblum. Multicalibration: Calibration for the (computationally-identifiable) masses. In J. G. Dy and A. Krause, editors, Proceedings of the International Conference on Machine Learning (ICML), Proceedings of Machine Learning Research, pages 1944–1953. PMLR, 2018\
> [HJZ22] N. Haghtalab, M. I. Jordan, and E. Zhao. On-demand sampling: Learning optimally from multiple distributions. In S. Bengio, H. M. Wallach, H. Larochelle, K. Grauman, N. Cesa Bianchi, and R. Garnett, editors, Advances in Neural Information Processing Systems, 2022\
> [KGZ19] M. P. Kim, A. Ghorbani, and J. Y. Zou. Multiaccuracy: Black-box post-processing for fairness in classification. In V. Conitzer, G. K. Hadfield, and S. Vallor, editors, Proceedings of the AAAI Conference on Artificial Intelligence (AAAI), pages 247–254. ACM, 2019.\
> [DKRRY21] C. Dwork, M. P. Kim, O. Reingold, G. N. Rothblum, and G. Yona. Outcome indistinguishability. In Proceedings of the Annual ACM SIGACT Symposium on Theory of Computing (STOC), pages 1095–1108. ACM, 2021.

---

### Official Review · Reviewer_vjmg · 2023-07-26

**Soundness:** 3 good
**Presentation:** 3 good
**Contribution:** 3 good
**Rating:** 5
**Confidence:** 2

**Summary:**

This paper provides a unifying framework for the algorithm design and performance analysis of multicalibrated predictors.
In this paper, the multicalibraion problems is placed in the setting of multi-objective learning.
Under this interpretation, approaches based on game dynamics is proposed and analyzed.
It is shown that this approach yields improved performance guarantees.

**Strengths:**

- Bounds are improved for a variety of problems.
- Experimental results support the effectiveness of the proposed method.

**Weaknesses:**

- The approach based on game dynamics does not seem very surprising as it is a common approach to transform min-max optimization into a problem of finding an equilibrium solution by interpreting it as a zero-sum game.
- I have concerns about whether the definition is consistent with that of existing studies. (please refer to "Questions")


Minor comments:
- When citing existing results in Table 1, etc., it would be better to indicate the theorem number or the relevant section. I had a hard time checking the corresponding part.

**Questions:**

- In previous studies (e.g., [17,21,34]), multicalibrated predictors appear to be defined in terms of conditional expectation.
On the other hand, in Definition 2.1 of this paper, it appears to be defined in some sense by joint probabilities.
I believe this means that the errors bounded in 2.1 are smaller than those adopted as definitions in previous studies.
This raises concerns about whether comparisons with existing research bounds are valid.
I would appreciate an answer to this concern to see if there is a problem.
- Is there lower bounds that can be compared to the results obtained?

**Limitations:**

I have no concerns about the limitations and potential negative societal impact.

---

> ### Author Rebuttal · Authors · 2023-08-10
>
> Thank you for your review. Below we address your comments and questions.
>
> **Is the game dynamics approach surprising?**\
> We agree that writing min-max optimization problems as a zero-sum game is a common first step in many analyses. But it is the next steps that matter, and for multicalibration it has not been obvious what those next steps might be.  Indeed, no prior multicalibration work has succeeded in using a game-theoretic framing as a unifying principle for multicalibration despite over 5 years (and 50+ papers) on the topic.
>
> The nontrivial underlying challenge is not the mere reduction to min-max optimization—which is why we refer to Facts 2.5 and 2.6 as facts and not lemmas/theorems—but rather in solving the resulting equilibrium computation problem in a way that connects to practical algorithms. The needs of multicalibration (such as determinism, large and complex predictor space, etc.) differ significantly in this regard from earlier applications of general-purpose no-regret algorithms and game dynamics. Most notably, the minimizing player’s action set is the set of all predictors, which scales exponentially in the domain size $|X|$! We must therefore construct a novel and highly nontrivial online learning strategy for the minimizing player with a regret that is independent of the size of the action set and $|X|$. Moreover, this no-regret algorithm must use non-randomized predictors against an adaptive adversary (Theorem 3.7). Another example is that we need to obtain a deterministic solution from the game dynamics despite not having convexity, which required us to introduce a novel form of no-regret/best-response game dynamics (Lemma 3.4) that, to the best of our knowledge, has not been studied previously.
>
> Overall we think that it is quite surprising that *every* known multicalibration algorithm and guarantee can be cleanly recovered—and improved upon—with game-theoretic learning dynamics using our framework. The generality of our framework and strength of Theorems 3.7 and Lemma 3.4 are further evidence that the progress made by our work is surprising, novel, and has the potential to significantly change the landscape of research in multicalibration. We will make sure that the final version of our paper emphasizes this generality from the outset and more clearly places in relief the challenges of applying learning dynamics to multicalibration.
>
> **Defining multicalibration with conditional expectations versus joint probabilities.**\
> Thank you for raising this question. Due to the inconsistent definitions of multicalibration employed in the multicalibration community, this has been an ongoing point of confusion.
> We define multicalibration using joint probabilities in the same way as other recent literature (see GHKRS23, DLLT23), where the error tolerance for a group $G$ is inversely proportional to the group’s probability mass $\Pr(x \in G)$. Some early multicalibration works (such as HKRR18) define multicalibration with a conditional expectation—where the error tolerance for a group is independent of the group’s probability mass—**but make an assumption that all groups have a probability mass of at least some constant fraction** (see, e.g., Theorem 3.7 in HKRR18 which states $|S| \geq \gamma N$ ). That assumption means that their definition is actually weaker than our more direct joint-probability definition that is now the norm in the multicalibration community. Please let us know if this clarifies your question and we will add a footnote clarifying this in the final version as well.
>
> We also want to bring to your attention that one of our results, Corollary E.4, shows that a guarantee even stronger than those of prior works is possible when the error tolerance for a group $G$ scales with $1 / \sqrt{\Pr(x \in G)}$ rather than $1 / \Pr(x \in G)$. We are able to obtain this novel guarantee by plugging in second-order regret bounds into our game dynamics framework.
>
> **Lower bounds**\
> We are not aware of any (nontrivial) lower bounds for multicalibration. We want to emphasize that our work has improved several upper bounds in multicalibration (see Table 1), suggesting the community’s known results are likely not tight.
>
> [HKRR18] U. Hebert-Johnson, M. P. Kim, O. Reingold, and G. N. Rothblum. Multicalibration: Calibration for the (computationally-identifiable) masses. In J. G. Dy and A. Krause, editors, Proceedings of the International Conference on Machine Learning (ICML), Proceedings of Machine Learning Research, pages 1944–1953. PMLR, 2018\
> [GHKR23] Globus-Harris I, Harrison D, Kearns M, Roth A, Sorrell J. Multicalibration as boosting for regression. arXiv preprint arXiv:2301.13767. 2023 Jan 31.\
> [DLLT23] C. Dwork, D. Lee, H. Lin, and P. Tankala. New insights into multicalibration, 2023.

---

> > ### Comment · Reviewer_vjmg · 2023-08-16
> >
> > Thank you very much for your thoughtful reply.
> > All my concerns have been addressed and I have no additional questions.
> > I think I understand the discussion about the definition of multicalibration thanks to the responses.
> > I would like to determine the final score after reading the opinions of other reviewers.

---

### Decision · Program_Chairs · 2023-09-21

**Decision:**

Accept (poster)

**Comment:**

By exploiting connections between multicalibration and multiobjective optimization, the paper unifies and in several cases improves upon existing multicalibration algorithms and guarantees through the game dynamics perspective. The contributions are strong and insightful. Most reviewers are positive with the contributions except one who did not participate in the discussion.  Hence, I recommend to accept the paper.